# Non-genetic neuromodulation with graphene optoelectronic actuators for disease models, stem cell maturation, and biohybrid robotics

Elena Molokanova [1,2] ✉, Teng Zhou [2,3], Pragna Vasupal [2,3], Volodymyr P. Cherkas [4,5], Prashant Narute [6], Mariana S. A. Ferraz[7,8], Michael Reiss [3], Angels Almenar-Queralt[7], Georgia Chaldaiopoulou [7], Janaina Sena de Souza [7], Honieh Hemati [2,3], Francisco Downey[1,3], Omowuyi O. Olajide [3], Carolina Thörn Perez [9], Francesca Puppo[7,10], Pinar Mesci[7,12], Samuel L. Pfaff [9], Dmitry Kireev [6], Alysson R. Muotri [7,10,11] ✉ & Alex Savchenko [1] ✉

Light can serve as a tunable trigger for neurobioengineering technologies, enabling probing, control, and enhancement of brain function with unmatched spatiotemporal precision. Yet, these technologies often require genetic or structural alterations of neurons, disrupting their natural activity. Here, we introduce the Graphene-Mediated Optical Stimulation (GraMOS) platform, which leverages graphene's optoelectronic properties and its ability to efficiently convert light into electricity. Using GraMOS in longitudinal studies, we found that repeated optical stimulation enhances the maturation of hiPSC-derived neurons and brain organoids, underscoring GraMOS's potential for regenerative medicine and neurodevelopmental studies. To explore its potential for disease modeling, we applied short-term GraMOS to Alzheimer's stem cell models, uncovering disease-associated alterations in neuronal activity. Finally, we demonstrated a proof-of-concept for neuroengineering applications by directing robotic movements with GraMOS-triggered signals from graphene-interfaced brain organoids. By enabling precise, non-invasive neural control across timescales from milliseconds to months, GraMOS opens new avenues in neurodevelopment, disease treatment, and robotics.

The ability to modulate neural activity at will is essential for fundamental research and therapeutic applications, including understanding brain function, disease modeling, neuroengineering, and developmental and regenerative neuroscience. Light is an exceptional actuating stimulus for neuromodulation due to its high spatial and temporal precision, non-invasiveness, tunability, and compatibility with various techniques for monitoring neural activity. However, neurons are not inherently light-sensitive and, therefore cannot respond to light on their own. To be modulated by light, they must either be genetically or structurally modified with internally inserted light-sensitive modules[1-4] or integrated with external optical actuators[5-8].

Genetic modification of neurons to enable light-controlled neuromodulation (optogenetics) involves expressing exogenous light-

sensitive ion channels[1,2]. Although optogenetics has advanced significantly over the past two decades, it still faces inherent limitations. These include the potential alteration of normal neuronal function or possible unintended side effects resulting from inserting opsin genes into the genome[9].

Structural modifications of neurons aimed at achieving built-in light sensitivity involve the insertion of synthetic photoisomerizable azobenzene-containing domains into the cell membrane of neurons[3,4]. This approach requires UV (~320-400 nm) and blue light (~400–500 nm) to manipulate the membrane thickness (and, therefore, its capacitance) by converting azobenzenes from a more stable elongated *trans* form to a less stable bent *cis* form and back. The limitations of this method include (a) the necessity for internal structural modifications; (b) compromised neuronal activity even in the absence of light (e.g., at rest, Ziapin2-stained neurons are less excitable, less spontaneously active, and more refractory to voltage changes[3]); (c) poor compatibility with long-term studies due to compromised integrity of the cell membrane; and (d) inevitable side effects from the high-intensity light (4 kW/cm² for UV and 15 kW/cm² for blue light) required for the repeated isomerization of azobenzene in OptoDArG[4].

However, preserving the structural and genetic integrity of neurons is essential for accurately deciphering brain activity and for the long-term neuromodulation needed in regenerative medicine and for treating neurological disorders. This is because, when neurons are altered to gain light sensitivity, the subsequent studies provide insights only into these modified neurons, undermining the goal of understanding the natural state of neurons and their networks. Our solution to this challenge is the use of optical actuators positioned outside the cells. These can stimulate neurons in their natural state without necessitating any modifications, thereby preserving their integrity and natural activity patterns.

In this work, we introduce graphene-based optoelectronic actuators that leverage graphene's unique physicochemical properties[10-15] for ultrafast, reversible, and efficient graphene-mediated optical stimulation (GraMOS) of intact neurons through a non-Faradaic capacitive mechanism. While graphene has traditionally served a passive role in neuroscience, our approach redefines it as an active neuromodulation actuator, greatly expanding its functional scope. To realize our solution, we fabricated graphene-based actuators in both freestanding and supported formats and demonstrated their reliable, precise, and repeatable optical neuromodulation in 2D and 3D space. We used acute GraMOS to assess the functional properties of Alzheimer's hiPSC-derived cell models, and applied GraMOS for extended periods to enhance neuronal maturation, underscoring its utility across disease modeling, neurodevelopmental studies, and therapeutic applications. GraMOS also facilitated a neuroengineering demonstration in which light-responsive graphene-interfaced brain organoids controlled an external robotic system. In summary, this work presents graphene-based optoelectronic actuators for ultrafast, reversible, and efficient neuromodulation, establishing graphene as an active interface for precise, non-Faradaic stimulation of neurons in both 2D and 3D systems, with broad applications spanning disease modeling, neurodevelopment, and bio-integrated robotics.

## Results
### Mechanism of GraMOS
Graphene possesses a range of tunable optoelectronic properties[10,11] that render it exceptionally well-suited for light-controlled neuromodulation applications. These include: (a) strong light-matter interaction, (b) efficient broadband light absorption despite its one-atom thickness[12,13], (c) ultrafast optical response, (d) high electrical and thermal conductivities, and (e) the capacity to effectively convert light into electricity via a hot-carrier multiplication process on a femtosecond timescale[14-16]. In graphene, electrons are delocalized across the

sp²-hybridized carbon lattice due to the π-conjugated system formed by overlapping $p_z$ orbitals, and this electron π-cloud extends above and below the graphene plane. In multilayer graphene and reduced graphene oxide (rGO), holes localize within the carbon lattice in specific regions such as defects and strained areas, while electrons largely inhabit the π-cloud above and below the lattice. External electric fields can further enhance this spatial charge separation (Fig. 1a).

Graphene, being a zero-bandgap semimetal with high carrier mobility, low density of states near the Dirac point, and delocalized electron π-cloud, is extremely sensitive to its electrostatic environment. When light is turned on, photogenerated electrons in graphene induce a rapid displacement of nearby ions near the graphene-electrolyte interface, forming an electric double layer. These processes modulate the local electrostatic environment of graphene, creating a quasi-static local photogating field, that results in a persistent shift of the Dirac point and Fermi level.

The absorbance of a single photon by graphene leads to the generation of multiple electrons through a phenomenon known as multiple carrier generation[14-19]. Initially, these photogenerated electrons possess high kinetic energy (referred to as 'hot' electrons) but rapidly undergo electron-electron scattering, phonon-assisted relaxation, or impact ionization[20]. Electron-electron interactions in graphene are substantially stronger than electron-optical phonon coupling[21,22], which enables a single absorbed photon to generate multiple electron-hole pairs. This process of carrier multiplication ensures that, instead of transferring energy to the graphene lattice (which would lead to heating), the 'hot' electrons preferentially transfer energy to other electrons instead of phonons, keeping the graphene lattice 'cold'. Additionally, any residual lattice energy can dissipate rapidly, typically within microseconds, due to the high thermal conductivity of graphene. These properties make graphene-based optoelectronic systems exceptionally effective.

A neuron near the graphene surface becomes optically activated when photogenerated 'hot' electrons from the π-cloud outside the graphene lattice displace cations at the interface between graphene and the neuronal cell membrane (Fig. 1a). This displacement alters the cell membrane potential through capacitive coupling[23] between the cell membrane and the graphene surface, which leads to depolarization, activation of voltage-gated ion channels, and generation of action potentials. The femtosecond lifetime of photogenerated electrons[16,24] prevents charge accumulation at the interface, and the diffusion of these short-lived photogenerated electrons beyond the illuminated area is minimal. As a result, the spatiotemporal resolution of GraMOS is strictly defined by the dimensions of the illuminated area.

### Characterization of GraMOS materials through analytical techniques and live-cell studies
To produce GraMOS actuators, we utilized chemically converted graphene (rGO) (Supplementary Fig. 1a) because rGO can offer cost-effective production and ease of processing, long-term electrochemical stability, reduced carrier recombination time, low impedance and high charge delivery parameters, and increased light absorption due to scattering[25,26]. We synthesized rGO using green chemistry methods[27,28] and thoroughly characterized it through UV-Vis spectroscopy, X-ray diffraction (XRD), Raman spectroscopy, atomic force microscopy (AFM), amplitude mode-Kelvin probe force microscopy (AM-KPFM), scanning electron microscopy (SEM), and Fourier-Transform infrared (FTIR) spectroscopy. This analytical characterization conclusively established a successful reduction of graphene oxide (GO) to rGO marked by a near-total removal of functional groups from the basal plane of GO, restoration of the conjugated π-electron system, and enhancement of the material's electronic properties (Supplementary Discussion, Supplementary Fig. 1b-f).

To visualize the resulting rGO flakes and quantify their dimensions, we used SEM (Fig. 1b) and AFM (Supplementary Fig. 2). We found

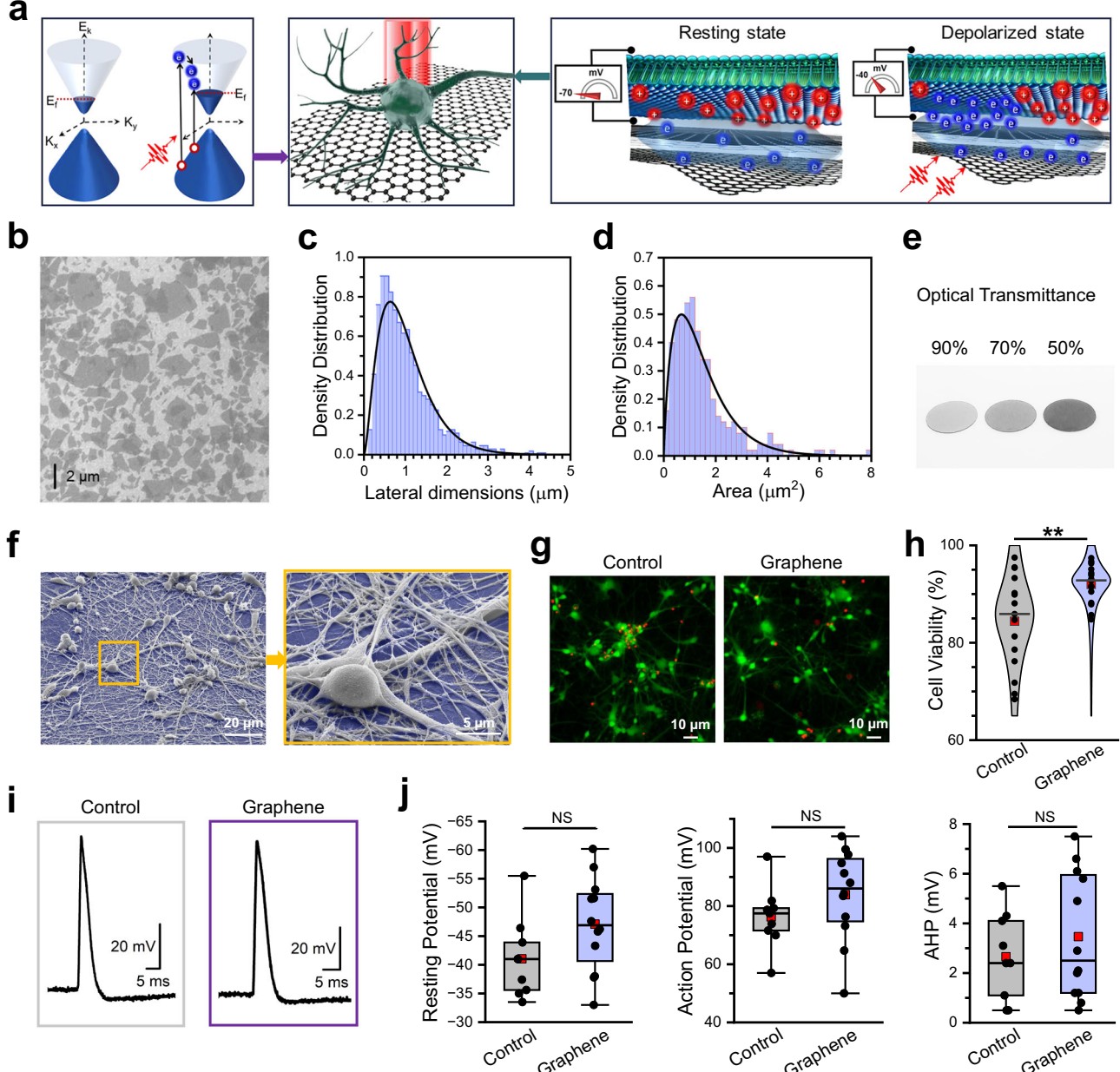

**Fig. 1 | Mechanism and neuronal biocompatibility of GraMOS. a** Mechanism of action of GraMOS. Left: Dirac cones for multilayer graphene materials exhibiting a small bandgap. $K$ = momentum; $E$ = energy; $E_f$ = Fermi energy (red dotted line). Light generates high-energy "hot" electrons (blue spheres) which, in turn, can produce additional electrons via impact ionization, leading to carrier multiplication. Holes are depicted as red spheres. Center: cartoon of a light-activated neuron on a graphene interface. Graphene lattice (one hexagonal unit cell is ~0.0524 nm²) and neuron (10,000 nm) are shown at different scales for illustration. Right: zoom-in of neuronal membrane at rest and under light illumination. Photogenerated electrons from the electron cloud outside the graphene lattice can dynamically induce capacitive effects at the cell-electrolyte-graphene interface, leading to membrane depolarization. **b** Representative SEM image of rGO flakes on the ITO substrate selected from 8 images of 4 samples. **c** Histogram of rGO flake lateral dimensions ($n$ = 1000). **d** Histogram of rGO flake areas ($n$ = 250). **e** Representative image of rGO flakes spray-coated on glass coverslips, producing G-substrates with ~90%, 70%, and 50% optical transmittance (left to right). **f** Representative SEM image of hiPSC-derived neurons on G-coverslips (false-colored blue), selected from 97 images of 15 samples. **g** Representative fluorescent images showing hiPSC-derived neurons on control coverslips (left) and G-coverslips (right). Neurons were labeled with EthD-1 (dead cells, red fluorescence) and calcein-AM (live cells, green fluorescence). **h** Summary of viability experiments using hiPSC-derived neurons on control coverslips and G-coverslips (three replicates, five random fields of view each). Data are presented as violin plots: center lines = median, red squares = mean, and individual data points are shown as circles (**$P$ = 0.00924, one-way ANOVA). **i** Representative action potentials from hiPSC-derived neurons on control coverslips and G-coverslips. **j** Selected electrophysiological properties of hiPSC-derived neurons on control ($n$ = 9) and G-coated ($n$ = 12) coverslips. AHP = Afterhyperpolarization. Data are presented as box plots: center lines = median, red squares = mean, box = upper/lower quartiles, whiskers = 5th–95th percentiles, and individual points are shown as circles.

that rGO flakes ($n > 250$) had average lateral dimensions of $1.05 \pm 0.04\,\mu m$ (Fig. 1c), a height up to $2.9 \pm 0.3\,nm$ (Supplementary Fig. 2b-c, Supplementary Fig. 4a, 4c-d), and a surface area of $1.46 \pm 0.3\,\mu m^2$ (Fig. 1d). Further adjustment of the lateral dimensions

and height of rGO flakes toward larger or smaller sizes was achievable via additional vortexing and brief centrifugation at low speed. To fabricate graphene-coated glass coverslips (G-coverslips) with optical transmittance in the range of 50% to 90% (Fig. 1e), we employed either

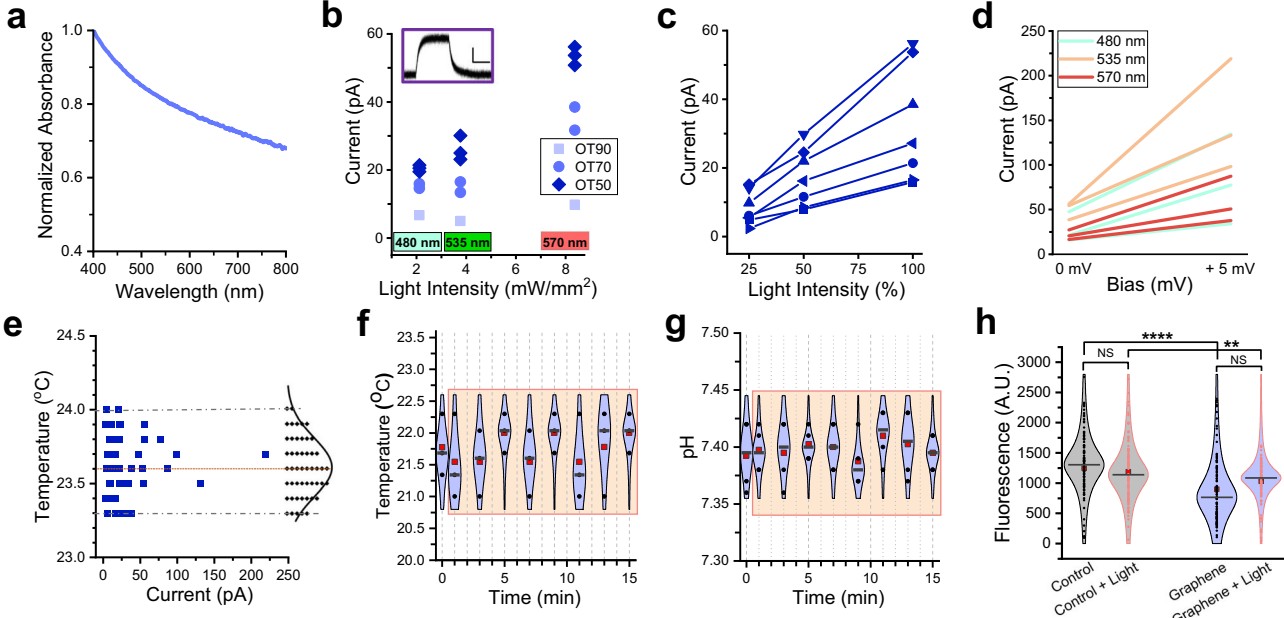

**Fig. 2 | Photophysical characterization of G-interfaces. a** Visible-range absorption spectra of rGO, showing broadband absorption with decreasing efficiency at longer wavelengths. **b** Photocurrents acquired from G-coverslips (the optical transmittance (OT) of 90% (squares), 70% (circles), and 50% (diamonds)) when illuminated by 480 nm light (2.1 mW/mm²), 535 nm light (3.7 mW/mm²), and 575 nm light (8.3 mW/mm²). The holding potential = 0 mV. Insert: an example of a photocurrent trace (bars: 20 pA/500 ms). **c** Photocurrents generated under different light intensities. Measurements from the same G-coverslips are connected by lines. **d** Photocurrents acquired at 0 mV and +5 mV holding potentials. The 0-mV and +5 mV values from the same G-coverslips are connected by lines. **e** Temperature measurements on the surface of G-coverslips as a function of photocurrent amplitudes (n = 50). The average surface temperature of G-coverslips when illuminated by light was 23.6 ± 0.03 °C vs 23.7 ± 0.04 °C in the dark as detected using a non-contact digital IR thermometer Mestek 800 C. **f** Local temperature

measurements near G-coverslips during 15-minute light exposure (535 nm, 3.7 mW/mm²; n = 3). The beige box indicates the light exposure period. Data are presented as violin plots: center line = median, red square = mean, and individual data points are shown as circles. **g** pH values of an electrolyte solution containing submerged 70-OT G-coverslips during 15 min light exposure (535 nm, 3.7 mW/mm²; n = 4). The beige box indicates the light exposure period. Data are presented as violin plots: center line = median, red square = mean, and individual data points are shown as circles. **h** CM-H₂DCFD-based quantification of reactive oxygen species (ROS) levels in control and G-interfaced 2-month-old WT83 brain cortical organoids in the dark and after light stimulation (2 Hz, 1 h, 1.9 mW/mm²) for 7 consecutive days. n = 100 cells per condition. Data are presented as violin plots: center line = median, red square = mean, and individual data points are shown as circles (**P = 0.0091, ****P = 4.3648 × 10⁻⁵; Two-way ANOVA).

drop-casting of rGO dispersions at concentrations ranging from 0.1 to 0.5 mg/mL, or fine-mist spray-coating of a 0.5 mg/mL rGO dispersion applied in 5–25 discrete deposition steps.

To evaluate biocompatibility at the largest neuron-graphene interface area, we cultured 2D human induced pluripotent stem cell (hiPSC)-derived neurons on G-coverslips for 3 weeks. While pristine graphene exhibits minimal interaction with cells due to its inert basal plane, rGO contains some residual functional groups and structural defects, particularly at its edges, which can increase its reactivity. This enhanced reactivity may lead to beneficial, neutral, or detrimental effects, necessitating a comprehensive assessment of its impact on biological systems. In this study, we found that hiPSC-derived neurons thrived on G-coverslips and formed complex neuronal networks, as evidenced by brightfield microscopy and scanning electron microscopy (SEM) images (Fig. 1f, Supplementary Fig. 3). To quantify biocompatibility of our graphene materials, we conducted cell viability assays using hiPSC-derived neurons cultured for 6 weeks on either G-coverslips or non-coated (control) coverslips and confirmed that G-interfaced neurons were at least as healthy as control neurons (Fig. 1g–h). Finally, we found no statistically significant differences in resting membrane potential, action potential, or afterhyperpolarization amplitudes between control and G-interfaced hiPSC-derived neurons (Fig. 1i–j), further confirming the absence of detrimental effects of the G-interfaces on overall health and electrical properties of neurons[29–32]. These findings align with numerous studies that have utilized graphene materials in various cellular scaffolds[33–37] and shown that nanoscale surface roughness, surface functional

groups, and electrical conductivity play key roles in promoting strong cell adhesion and ensuring high cell viability on graphene surfaces[38,39]. It is important to note that some groups have reported that graphene materials produced in their labs were toxic to their cells. However, it is crucial to consider that (a) the cellular toxicity of any given graphene sample can be influenced by a variety of factors, including the presence of contaminants, the lateral dimensions of graphene materials and their concentrations, etc[40], and (b) it is generally easier to harm cells than to preserve their physiological integrity. Therefore, conclusions regarding the adverse effects of graphene materials described in some studies should not be overgeneralized.

## Optoelectronic properties of graphene interfaces

To quantify the optoelectronic properties of G-interfaces, we took advantage of their broadband absorption spectra (Fig. 2a) and exposed them to different light signals varying properties while recording light-triggered currents at a distance of approximately 5 μm using the patch-clamp method. We established that the amplitude of light-triggered currents depended on light intensity and wavelength, and the optical transmittance of G-interfaces (Fig. 2b). As expected, increasing the number of graphene layers and the intensity of a specific light wavelength resulted in a corresponding increase in photocurrent amplitude (Fig. 2c), although a decrease in absorption at longer light wavelengths, combined with variations in light intensity, could obscure this outcome. Note that in G-coverslips, the absorption is dictated by the rGO coating, since the underlying glass is optically transparent. As a result, the optical absorption characteristics of the G-coverslips (Fig. 1e) closely align with

the intrinsic absorption spectrum of rGO shown in Fig. 2a, and differences in optical transmittance arise solely from variations in the thickness of the rGO coating. To illustrate this point, we show absorption spectra for G-coverslips of different optical transmittance (Supplementary Fig. 4). Photocurrents were detectable at a holding potential of 0 mV and increased at +5 mV in the patch pipette (Fig. 2d), indicating that these currents are driven by the flux of photogenerated negatively charged carriers. To validate this finding and analyze the potential gradient across rGO flakes, we employed AM-KPFM. These experiments (Supplementary Fig. 5) confirmed that the average potential differences between rGO flakes and the $SiO_2/Si$ substrate ranged from approximately −40 mV to −65 mV. The observed negative potential gradient suggests that rGO flakes carry a negative charge due to an excess of electrons, leading to localized negative charge accumulation. These results align with the findings from FTIR and Raman spectroscopy (Supplementary Fig. 1d, f), which indicate the presence of an electron-donating dopant−nitrogen in this case−further supporting the conclusion that the rGO flakes are negatively charged.

Finally, we evaluated the photoresponse of rGO flakes by fabricating an electrolyte-gated rGO-based field-effect transistor (FET) device on an interdigitated electrode (IDE) design (Supplementary Fig. 6a, b). The transfer characteristics displayed a V-shaped curve typical of graphene-based devices (Supplementary Fig. 6c). Upon exposure to low-intensity UV light ($1.8 \text{ mW/cm}^2$), we observed a shift in the charge neutrality point to the right, by nearly 100 mV, indicating light-induced doping and emphasizing the significant impact of UV light on the electronic properties of rGO driven by photo-induced charge carrier dynamics. Furthermore, the process is dynamic (Supplementary Fig. 6d): in this instance, the transistor was sampled at an operational point, measuring the drain-source current over time while the laser was directed onto the transistor for fixed periods (~15 s), resulting in an immediate change in current, as anticipated by the model described above.

Importantly, we found no correlation between surface temperature and photocurrent amplitude (Fig. 2e), confirming that these currents were not thermally induced. To verify that there are no localized temperature changes on the surface of G-coverslips during light exposure, we employed a patch-clamp-based temperature measurement approach, leveraging the strong temperature dependence of electrolyte conductivity[41,42] (Fig. 2f) Next, we established that continuous 15-min light exposure (452 nm, $4.1 \text{ mW/mm}^2$) of G-coverslips immersed in an electrolyte solution did not cause significant changes in the solution's pH (Fig. 2g), indicating that electrochemical processes play a minimal, if any, role in photocurrent generation at G-interfaces. Lastly, to rule out the contribution of Faradaic processes−which involve charge transfer to species in the electrolyte, leading to redox reactions and the generation of reactive oxygen species (ROS)−we evaluated ROS levels under our experimental conditions. We conducted ROS assays across four groups: control and G-interfaced 2 month-old WT83 brain cortical organoids, each with or without prolonged light stimulation (2 Hz, 1 h per day for 7 consecutive days, $1.9 \text{ mW/mm}^2$). Our results showed that light exposure of G-interfaces did not increase ROS levels. In fact, the mere presence of G-interfaces significantly reduced ROS levels compared to control organoids, both with and without light exposure (Fig. 2h). The reduced oxidative stress observed in G-interfaced cells may result from several physicochemical and biological properties of graphene materials, including their inherent ROS-scavenging capabilities, which allow them to chemically neutralize reactive oxygen species, and high electrical conductivity, which can facilitate more efficient electron transfer during redox reactions, thereby preventing ROS accumulation[43,44].

## GraMOS triggers action potentials in neurons

Combining these cell-friendly, photo-responsive graphene materials with neurons creates an exceptional interface that not only supports neuronal viability and function, but, most importantly, can actively modulate neuronal activity by light. To demonstrate how G-interfaces can enable GraMOS, we cultured primary cortical neurons on G-coverslips and conducted patch-clamp studies with and without GraMOS-triggering light signals (Fig. 3a).

In voltage-clamp experiments, we demonstrated that light illumination produced inward ion currents, with the amplitude of these currents scaling with light intensity (Fig. 3b). The rise and decay times of these currents were in the millisecond range, and their amplitude remained at steady-state levels during continuous light illumination. These results further confirm that photothermal effects[41] do not play a role in GraMOS (Fig. 2e, f), as anticipated from the inefficiency of electron-phonon coupling in graphene and its exceptional thermal conductivity. This conclusion is further supported by the absence of statistically significant changes in the surface temperature of G-substrates during prolonged light exposure, as measured using a thermocouple probe or a patch pipette[33].

In current-clamp experiments, brief light pulses reliably triggered single action potentials, demonstrating precise temporal control of neuronal firing. In contrast, prolonged light exposure induced a train of action potentials (Fig. 3c). These results highlight the versatility of our optical stimulation approach in modulating neuronal firing patterns.

## GraMOS-empowered all-optical studies of neuronal activity

Calcium imaging enables millisecond-scale, single-cell dynamic monitoring of neural network activity across multiple cells. Integrating optical stimulation with optical monitoring provides a powerful tool to studying functional connectivity, synaptic transmission, and plasticity, making it especially useful for identifying dysfunctional neural circuits in neurological disorder models.

GraMOS is ideally suited for use alongside optical monitoring of neuronal activity (Fig. 3d), because graphene exhibits (a) high optical transparency, and (b) broadband light absorption due to the graphene's electronic band structure which allows it to absorb photons across a wide range of wavelengths, from ultraviolet to infrared[10,11]. Figure 3e presents Fluo-4-labeled primary cortical neuronal cultures interfaced with graphene, highlighting that the exceptional optical transparency of G-coverslips enables high-quality optical imaging without interference.

In GraMOS-empowered all-optical assays, two distinct light signals are required: $L_S$ to photoactivate G-interfaces and $L_e$ to excite fluorescent indicators. To prevent optical crosstalk between these two signals, we took advantage of graphene's broad light absorption and the differing light intensity thresholds necessary for GraMOS initiation versus fluorescent indicator excitation (Fig. 3f). If the light wavelength for GraMOS is selected outside the absorption spectrum of fluorescent indicators, $L_S$ does not impact fluorophores, and does not interfere with optical monitoring. Conversely, the excitation light for fluorophores cannot initiate GraMOS if its intensity remains below the activation threshold required for GraMOS. For instance, in GraMOS-empowered all-optical Fluo-4-based calcium imaging, we can use the following combination of light signals: (1) Fluo-4 excitation light at a subthreshold (low) intensity (e.g., 488 nm, $<0.5 \text{ mW/mm}^2$), and (2) light of any wavelength outside the fluorophore excitation spectrum (e.g., 638 nm, $2 \text{ -}10 \text{ mW/mm}^2$). Since the intensity and duration of $L_S$ and $L_e$ must be independently controlled, we engineered an LED-based light source PhotonMaker (Nanotools Bioscience) comprising seven independent LED light sources that span the visible light spectrum. Each channel can be individually modulated in both intensity and duration with microsecond precision, while also supporting simultaneous multi-wavelength operation when required.

We demonstrated that GraMOS triggered by wide-field light illumination can simultaneously activate multiple cells, and their

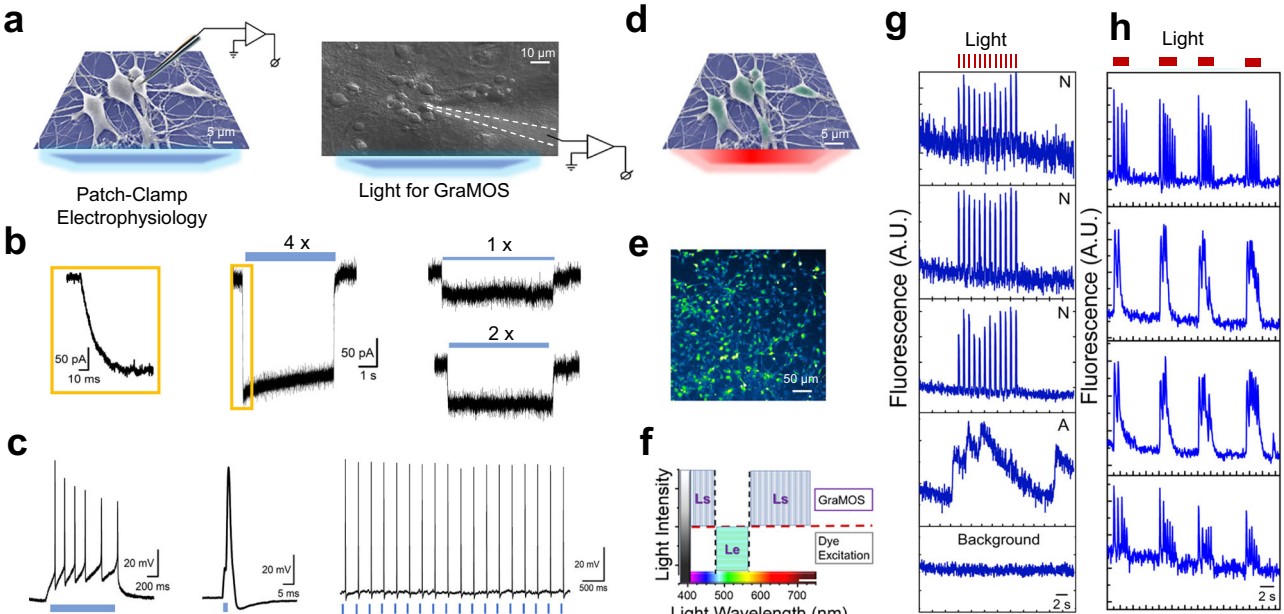

**Fig. 3 | GraMOS-enabled activation of neurons on G-interfaces. a** Experimental scheme (left) and a representative brightfield image of a neuron on a G-substrate during patch-clamp experiments (right) ($n > 50$ experiments). **b** Currents triggered by 5-s light illumination (452 nm; 1.9 mW/mm², 4.1 mW/mm², and 8 mW/mm² for 1x, 2x, and 4x light intensities, respectively) in voltage-clamped G-interfaced neurons ($V_h$ = -70 mV). The insert on the left (yellow box) shows a zoom-in of the raising phase of light-triggered currents in neurons. **c** Light-triggered action potentials in G-interfaced neurons (452 nm; 4.1 mW/mm²; 1-s duration for the left trace, and 2-ms duration for the center and right traces). **d** Experimental scheme for GraMOS-empowered all-optical calcium imaging under wide-field illumination. **e** Representative images of Fluo-4-labeled neurons on G-substrates ($n > 50$

experiments). **f** Light parameters for GraMOS-based assays minimize optical crosstalk: wavelengths of stimulation light ($L_S$) lie outside the fluorophore absorption range, while the intensity of fluorophore excitation light ($L_e$) remains below the GraMOS activation threshold. **g** Representative calcium transients from several regions of interest in the same field of view of G-interfaced cells (neurons – N; astrocyte – A) and Fluo-4 signal outside cells (background – the bottom trace) in response to pulsed wide-field light illumination (638 nm; 3.9 mW/mm², 5 ms duration). **h** Representative bursting calcium transients triggered by prolonged wide-field light illumination (638 nm; 3.9 mW/mm²; 2-s duration) in G-interfaced primary cortical neurons.

activity patterns are governed by the duration of light signals: brief 2-ms pulses trigger single action potentials (Fig. 3g), and 1-s pulses initiate bursting activity containing action potential trains (Fig. 3h). Beyond stimulating already active neurons, GraMOS also recruited many that were not spontaneously active, revealing its capacity to induce activity even in some quiescent cells. For example, in a 3-month-old cell model comprising hiPSC-derived neurons, the percentage of spontaneously active neurons was low ($10.5 \pm 2.4$ %), whereas GraMOS triggered calcium transients in $68.2 \pm 3.3$ % of cells in the field of view (Fig. 3g, Supplementary Movie 1). Since the insufficient maturity of hiPSC-derived neurons often contributes to their lack of spontaneous activity, GraMOS can help overcome this limitation by providing optical stimulation to evoke activity in these immature neurons, even before they can fire on their own. The GraMOS's ability to activate these dormant neurons can also be invaluable in studies of neurodevelopmental or neurodegenerative diseases, when certain populations of neurons may be hypoactive, dysfunctional, or disconnected.

To visualize the propagation of action potentials along neuronal networks in response to a single spatially-constrained light pulse, we used a Dragonfly 600 Confocal microscope with a MicroPoint module (Oxford Instruments). In this setup, the focused light spot measured approximately 2 μm, allowing precise optical stimulation of individual neurons and the evaluation of neuronal connectivity through observed activation patterns within networks (Fig. 4a). Using 2D hiPSC-derived neurons on G-coverslips or within our graphene-coated 96-well plates, we demonstrated that a single light pulse can directly activate one neuron, gradually triggering network-wide activation as excitation signals propagate in all directions (Fig. 4b, Supplementary Movie 2).

## GraMOS for phenotyping neurological disease models

To explore how GraMOS-empowered all-optical studies can aid in evaluating neuronal connectivity and circuit integrity in health and disease, we used GraMOS for early-stage functional phenotyping of hiPSC-derived neurons modeling familial Alzheimer's disease (FAD). Earlier initiation of studies provides more informative insights into FAD progression. GraMOS enables functional phenotyping at very early disease stages by eliminating waiting for detectable spontaneous neuronal activity. Additionally, since external stimuli drive organized cognitive activity in the brain, GraMOS can replicate these processes, further emphasizing its necessity.

In these pilot experiments, we used G-interfaced hiPSC-derived neurons harboring the APP M233L mutation, a pathogenic FAD-linked mutation, and WTAD neurons, wild-type iPSC-derived neurons from non-affected individuals[45]. M233L neurons tend to display a combination of affected network activity, disrupted calcium homeostasis, and synaptic dysfunction which can lead to cognitive deficits commonly observed in Alzheimer's disease. Being 3 weeks old, neither WTAD nor M233L hiPSC-derived neurons exhibited spontaneous calcium transient spiking activity (Fig. 4c). By utilizing spatially constrained GraMOS, we optically activated these neurons and demonstrated that M233L neurons exhibit increased excitability in response to activating stimuli. This is reflected in the higher ratio of GraMOS-activated M233L neurons compared to WTAD neurons (Fig. 4d). Further, we found that the amplitude and kinetics of calcium transients were affected in M233L neurons (Fig. 4e), indicating disrupted calcium homeostasis, which can lead to impaired synaptic plasticity and activity-dependent signaling. Spearman correlation coefficient matrices reveal that while WTAD neuronal networks are comprised of several distinct neuronal clusters connected with each

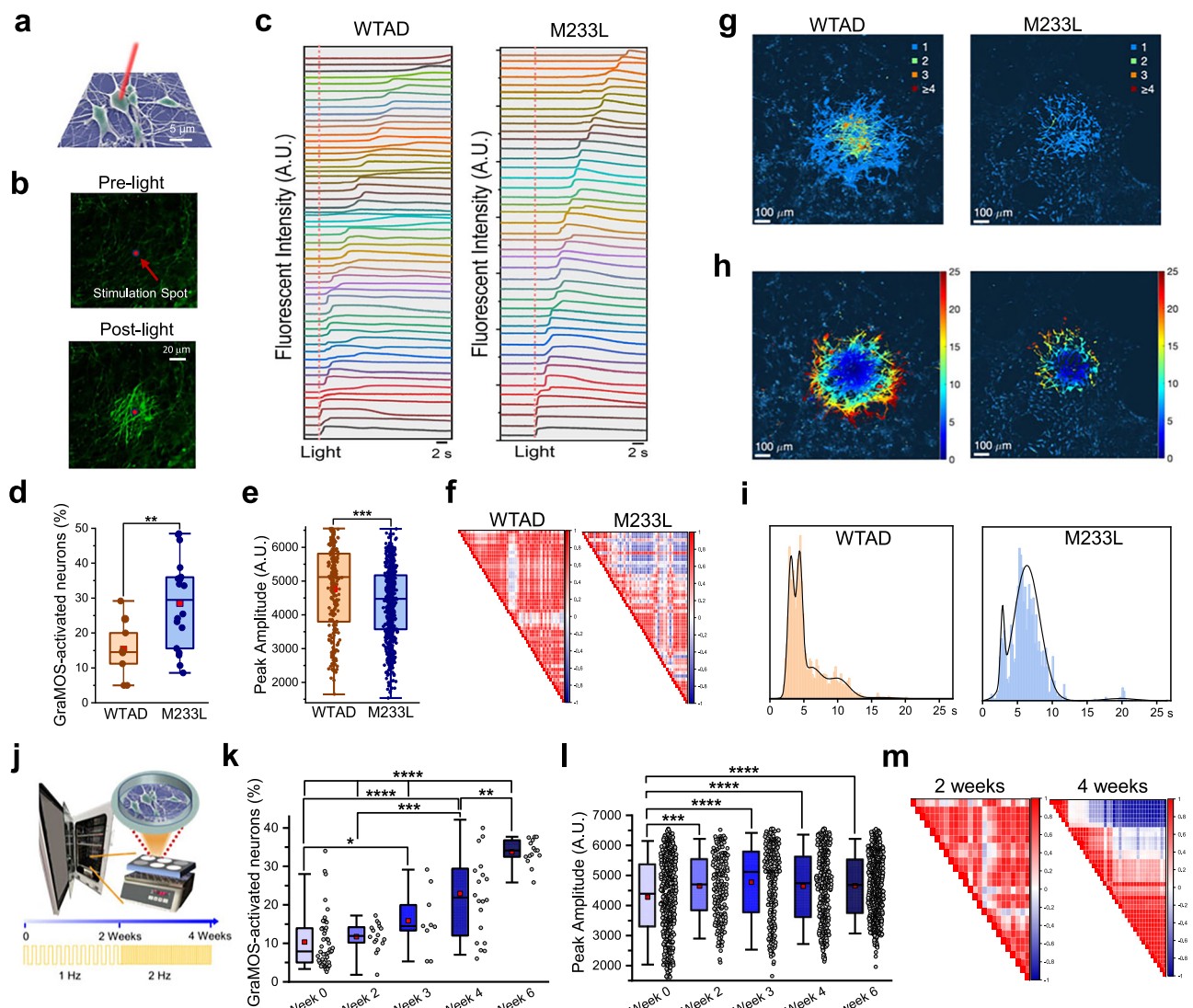

**Fig. 4 | GraMOS enables functional phenotyping and promotes maturation of G-interfaced hiPSC-derived neurons. a** Scheme for GraMOS-empowered all-optical studies using single-cell single-pulse light. **b** Videoframes of G-interfaced Fluo-4-labeled neurons at rest and following activation of a central neuron by a single light pulse (561 nm, 5.2 mW/mm²). **c** Calcium transients across WTAD and M233L networks triggered by a single light pulse. **d** Ratio of GraMOS-activated neurons in WTAD and M233L models (n = 9 and 18 G-coverslips, respectively; **$P$ = 0.0098, one-way ANOVA). **e** Calcium transient amplitudes in G-interfaced WTAD and M233L neurons (n = 251 and 678, respectively; ***$P$ = 0.0004, one-way ANOVA) activated by a single light pulse. Box plots in (**d**) and (**e**): center line = median, red squares = mean, box = upper/lower quartiles, whiskers = 5th–95th percentiles, and individual points = circles. **f** Spearman correlation matrix for GraMOS-initiated neuronal activity in G-interfaced WTAD and M233L neurons. **g** Spatial maps showing the number of spikes in GraMOS-activated WTAD and M233L neuronal networks. **h** Spatio-temporal maps depicting the activation times (in seconds) after a light pulse in WTAD and M233L neurons (****$P$ = 1.0063 × 10⁻²³⁵,

two-sided Wilcoxon rank sum). **i** Histogram of activation times for WTAD and M233L models. **j** Scheme for GraMOS-driven maturation of G-interfaced hiPSC-neurons. **k** Ratio of GraMOS-activated WTAD neurons at different timepoints during light training (n = 47, 16, 9, 20, 15 for Weeks 0, 2, 3, 4, 6) (*$P$ = 0.0422 (Weeks 0 vs 3); ***$P$ = 0.0006 (Weeks 2 vs 4); ****$P$ = 8.119 × 10⁻⁷ (Weeks 0 vs 4); $P$ for Week 6 vs Week 0–6.6059 × 10⁻¹⁸, Week 2–8.2329 × 10⁻⁷, Week 3–1.3661 × 10⁻⁷, Week 4–0.0011; Two-way ANOVA). **l** Amplitudes of GraMOS-triggered calcium transients in WTAD neurons at different timepoints during light training (n = 542, 202, 245, 364, 675 neurons for Weeks 0, 2, 3, 4, 6) (****$P$ = 7.0376 × 10⁻⁷ (Weeks 0 vs 3); ***$P$ = 3.1209 × 10⁻⁴ (Weeks 0 vs 2); ****$P$ = 5.154 × 10⁻⁵ (Weeks 0 vs 4); $P$ = 2.5673 × 10⁻⁸ (Weeks 0 vs 6); Two-way ANOVA). Box plots in (**k**) and (**l**): center line = median, red squares = mean, box = upper/lower quartiles, whiskers = 5th–95th percentiles, and individual points = circles. **m** Spearman correlation matrix for GraMOS-initiated neuronal activity in WTAD models at different timepoints during light training. A.U. arbitrary units.

other, M233L networks are firing randomly (Fig. 4f). Finally, we analyzed the neuronal network connectivity in WTAD and M233L cell models by computing the activation times individually for each pixel and depicting the regions where more than one spike timing occurs in spatial maps (Fig. 4g–j). Only WTAD neuronal networks showed spatial sites in which three or more spike timings were consistently observed, indicating well-organized networks with the greater electrical connectivity within the network. Note that although the analysis was performed pixel-wise, the spatial sites showing multiple activations

coincided with the locations of entire neurons. Using the timing of the GraMOS-evoked spikes, we also generated activation maps for WTAD and M233L (Fig. 4h, Supplementary Movie 3), and found that WTAD networks show approximately three times the number of pixels where calcium spikes were observed. This difference further confirms that M233L networks have fewer electrically connected neurons. Histograms of the activation times (Fig. 4i) follow approximately five Gaussian components ($t_{1-5}$ = 3.096 s, 4.382 s, 6.082 s, 10.111 s, 16.518 s) in WTAD neuronal networks versus approximately three Gaussian

components ($t_{1-3}$ = 2.853 s, 6.361 s, and 19.764 s) in M233L neuronal networks, suggesting more complex propagation patterns in WTAD. Finally, the standard deviation of activation times in M233L networks was 2 to 3 times larger than WTAD, indicating a lesser degree of coherence in the network.

## GraMOS-driven accelerated maturation of hiPSC-derived neurons

hiPSC-derived neurons typically fail to exhibit mature, adult-like properties, much less an aged phenotype. Instead, they resemble neurons in a fetal or neonatal-equivalent maturational state. This limitation is particularly problematic for predictive neurological disease modeling, as the inability to accurately recapitulate adult neuronal phenotypes significantly restricts their utility in studying age-related neurological disorders and drug discovery efforts[46]. Long-term cell stimulation can enhance hiPSC-derived neuron properties, but existing technologies have limitations for stem cell research. GraMOS, however, is exceptionally cell-friendly and suitable for both short-term and chronic studies. Repeated use of GraMOS over time promotes activity-dependent processes that enhance maturation and drive aging of hiPSC-derived neurons, providing them with 'activity-driven education' in a dish.

To explore GraMOS's utility for promoting the maturation of hiPSC-derived neurons, we conducted pilot experiments by placing 2-week-old WTAD hiPSC-derived neurons cultured on G-coverslips atop of a light illumination module (the LightKick system (Nanotools Bioscience)) inside a cell culture incubator, and subjected them to light illumination patterns with increasing frequencies for 4 weeks (Fig. 4j). At this early stage of development, WTAD hiPSC-derived neurons—whether cultured on control or G-coverslips—did not exhibit spontaneous activity, and only a small fraction of neurons interfaced with graphene responded to optical stimulation, while neurons on control coverslips showed no light-evoked response as usual. Over the course of this GraMOS-based training, we determined that the number of GraMOS-activated neurons and the peak amplitudes of GraMOS-triggered calcium transients increased during light training (Fig. 4k, l), indicating an increase in responsiveness to external stimuli and improved calcium homeostasis. Importantly, Spearman coefficient correlation matrices show that the neuronal network activity became more clustered and organized over the course of light training (Fig. 4m), reflecting the emergence of functional neuronal networks and suggesting stronger and more refined synaptic connections.

## GraMOS in graphene-interfaced brain cortical organoids

hiPSC-derived brain cortical organoids replicate the 3D development[47], facilitating studies on neural connectivity, disease modeling, personalized medicine, drug discovery, neurotoxicity, and brain-computer interfaces, and offering insights beyond traditional 2D cultures. Currently, brain organoids are cultured while being disconnected from the external environment, which impedes their maturation and limits their potential for studying real-time neural dynamics and interactions. To overcome this, we integrated light-activatable G-interfaces into the organoids, enabling them to 'see' light and transforming them into light-responsive systems.

To achieve this, we developed a protocol in which rGO flakes at concentrations 0.01–0.1 mg/mL were introduced during the early stages of organoid generation (Fig. 5a, b). Given that the dimensions and concentrations of rGO flakes, along with potential edge reactivity, may influence their biocompatibility, we characterized G-interfaced brain organoids by performing the cell viability assays and immunochemistry studies. We determined that graphene materials inside organoids did not affect their health (Fig. 5c, d) and that G-interfaced organoids developed normally and maintained a similar organization to control organoids (Fig. 5e, f), with the exception of a decreased VGLUT1/GAD65/67 ratio observed in G-interfaced organoids (Fig. 5f,

right). It is well established that the excitation-inhibition (E/I) ratio typically decreases during neuronal maturation[47–49], due to the earlier differentiation and integration of excitatory glutamatergic neurons compared to the slower maturation and delayed circuit incorporation of inhibitory GABAergic interneurons. Therefore, the observed reduction in the VGLUT1/GAD65/67 ratio suggests that graphene may subtly influence the maturation dynamics of excitatory and inhibitory neuronal subtypes, potentially contributing to an earlier stabilization of the E/I balance.

To test whether neurons inside G-interfaced brain organoids can be stimulated via GraMOS, we conducted calcium imaging assays (Fig. 5g) using a Dragonfly 600 Laser Confocal System with a MicroPoint module (Oxford Instruments). Using a single light pulse to optically stimulate a single neuron within an organoid, we successfully triggered network-wide neuronal activity. As illustrated in Fig. 5h, electrical signaling propagated from one neuron to another, activating numerous neurons multiple times on its way. The success rate of single-cell optical stimulation inside G-interfaced organoids was approximately 20%, which we attribute to the limited number of neurons directly interfaced with graphene, resulting in a decreased probability of a small light spot encountering a G-interfaced neuron. Future technical optimizations could involve increasing graphene concentrations during organoid generation or using a broader light beam. Note that calcium imaging is less temporally precise than electrophysiological recordings due to the slower dynamics of calcium transients and the indirect detection of electrical activity through calcium influx. In hiPSC-derived neurons, this temporal precision is further affected by the immaturity of calcium handling systems as well as slower calcium influx and buffering kinetics.

To directly assess GraMOS effects on the electrical activity across entire G-interfaced brain organoids, we used a Maestro Pro™ microelectrode array (MEA) system equipped with a Lumos module for wide-field optical stimulation (Axion Biosystems) (Fig. 6a). Unlike calcium imaging, which captures activity from individual cells, a MEA system enables simultaneous, non-invasive recording of population-level neuronal spiking activity. We tested two types of graphene-organoid interfacing: (1) an external configuration, where G-interfaces are applied to the organoid surface (Fig. 6b, left), and (2) an internal configuration, where G-interfaces are randomly embedded within the organoids (Fig. 6b, right). While the external configuration is more suitable for acute experiments, the internal configuration supports long-term studies by allowing organoids to remain free-floating and maintain their spherical architecture. Distinct GraMOS effects in these configurations could result from the differences in the surface areas and positions of G-interfaces.

For external interfacing, we generated and cultured brain organoids for 3 months according to our validated semi-guided protocol[47,50], transferred them onto G-coverslips for 3–4 weeks, and then placed organoids attached to these coverslips into a well of a MEA plate by flipping a coverslip so that organoids were positioned directly on electrodes with a G-coverslip on top. Using this external G-interfacing, we found that 5-ms light pulses can trigger a dramatic increase in the average electrical activity of G-interfaced organoids, but not in control organoids (Fig. 6c, d), followed by a return to pre-stimulation electrical activity levels after the light stimulation ends. The histogram of lag times between light stimulus and spike origination showed a trimodal distribution (Fig. 6e), indicating three distinct response groups: short-latency (direct stimulation and monosynaptic connections), intermediate-latency (polysynaptic pathways or indirect activation), and long-latency (network reverberation, inhibitory rebound, or delayed integration from distant circuits). The upper limit of fold changes in firing frequencies per active electrode in externally G-interfaced brain organoids was 16.2 (Fig. 6f), with the histogram of these changes also displaying a trimodal distribution (Fig. 6g). To

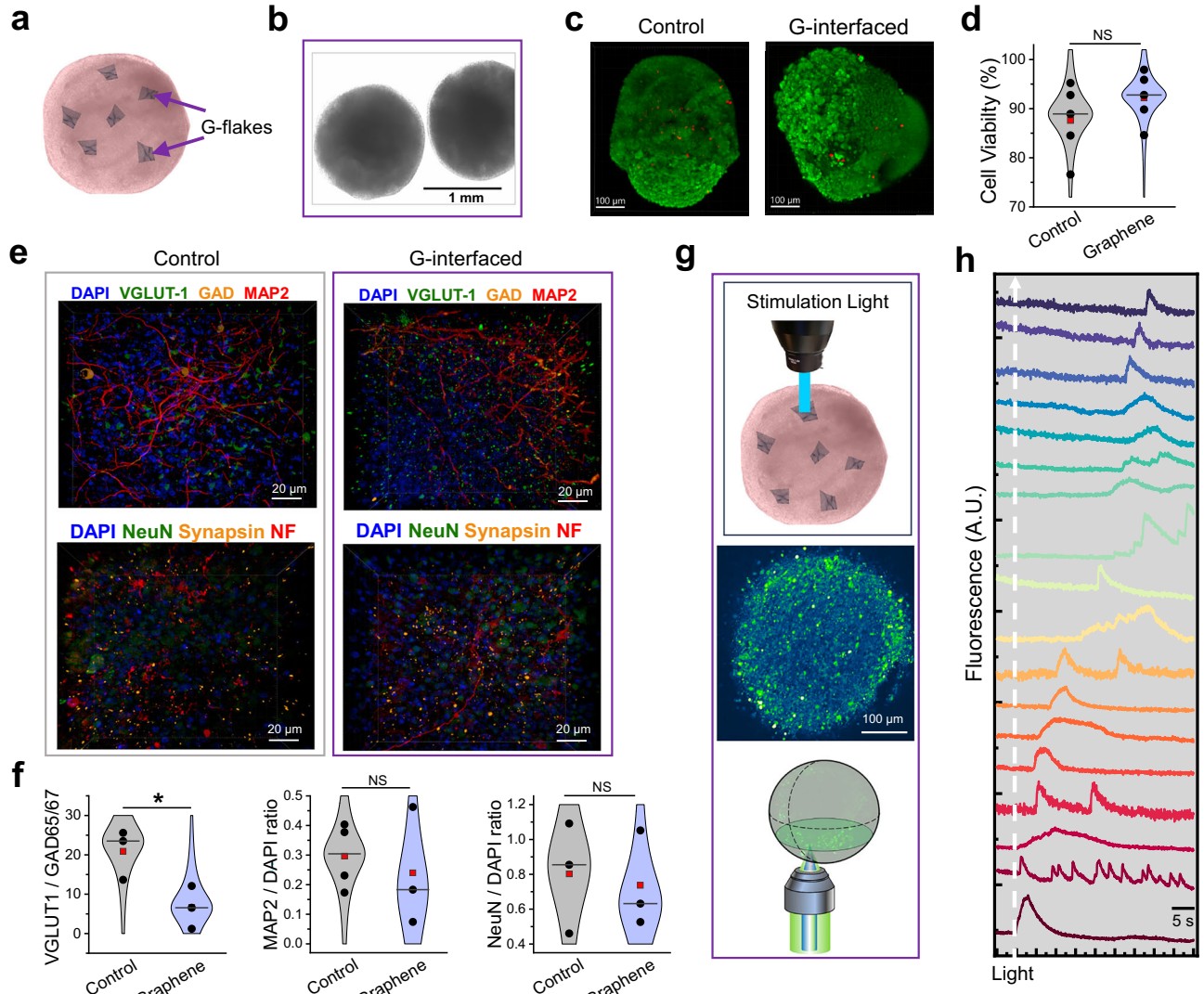

**Fig. 5 | GraMOS in hiPSC-derived brain cortical organoids. a** Schematic of a hiPSC-derived brain organoid interfaced with internal G-flakes. **b** Bright-field image of brain organoids with G-flakes inside. **c** Control and G-interfaced brain organoids labeled with EthD-1 (dead cells, red fluorescence) and calcein-AM (live cells, green fluorescence). **d** Summary of cell viability experiments performed in control and G-interfaced brain organoids (*n* = 5 per group). Data are presented as violin plots: center line = median, red square = mean, and individual data points are shown as symbols. Data were analyzed using one-way ANOVA; no statistically significant differences were observed. **e** Representative immunostaining images of 10-week-old control and G-interfaced brain organoids labeled with DAPI and antibodies for characterization of neuronal morphology and composition of neuronal networks.

**f** Population analysis of specific markers in control and G-interfaced brain organoids. Data are presented as violin plots: center line = median, red square = mean, and individual data points are shown as symbols (*n* = 3 per group; *P* = 0.0416, one-way ANOVA). **g** Experimental scheme of GraMOS-empowered all-optical calcium imaging on a confocal microscope using pulsed spatially-limited light illumination (top and bottom) with a representative confocal plane inside a Fluo-4-labeled G-interfaced brain organoid (center). **h** Calcium transients triggered by spatially confined 561 nm light pulses (5.2 mW/mm², 1 ms duration), with excitation propagating across the neuronal network and appearing in selected neurons within the field of view. The white dashed line shows the timing of the pulsed light signal. A.U. - arbitrary units.

evaluate changes in the complexity of electrical responses caused by light stimulation in externally G-interfaced brain organoids, we calculated the Lempel-Ziv Complexity (LZC) measure and found that it increased by 1.90 ± 0.03 (*n* = 32), indicating enhanced network engagement, dynamic information processing, and a shift toward a more physiologically relevant activity state.

Brain organoids that were internally interfaced with randomly dispersed rGO flakes (Fig. 5b, right) during their generation (Fig. 5) also exhibited increased electrical activity in response to different light patterns (Fig. 6h). However, the average mean firing rate increase was smaller than in externally G-interfaced organoids, likely due to the reduced surface area available for neuron-G interactions in the internal configuration. The lag time histogram

(Fig. 6j) showed a broad distribution with a single peak, as expected when multiple G-interfaced activatable neurons are distributed throughout the organoid, leading to broad response latencies influenced by synaptic distances and network dynamics. The upper limit of fold changes in firing frequencies per active electrode in internally G-interfaced brain organoids was 39.3 which is significantly higher than in externally G-interfaced organoids (Fig. 6k), with the histogram of these changes also displaying a very broad distribution (Fig. 6l), which could be explained by signaling amplification from multiple activation spots. The increase in the LZC in internally G-interfaced brain organoids in response to light was also significantly greater (5.11 ± 0.44 (*n* = 32)) than in externally G-interfaced organoids. When multiple activation spots are

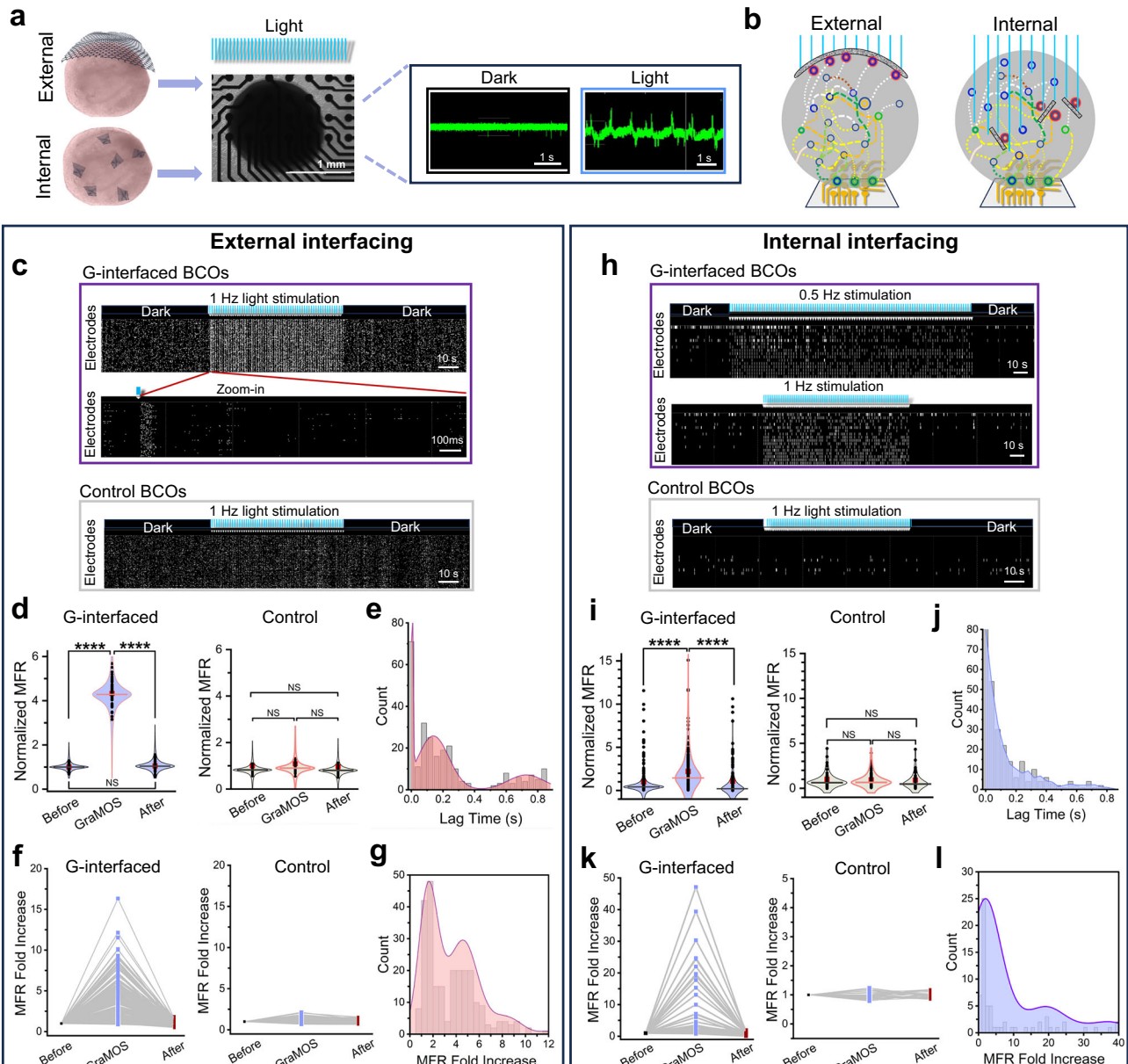

**Fig. 6 | GraMOS-evoked electrical activity in G-interfaced hiPSC-derived brain cortical organoids on microelectrode arrays. a** Schematic of GraMOS-empowered MEA experiments from G-interfaced brain cortical organoids (BCOs) (left), and representative examples of spontaneous and GraMOS-evoked electrical activity (right). **b** GraMOS during external (**c**–**g**) and internal (**h**–**l**) interfacing of rGO flakes with brain organoids. **c** Raster plots showing GraMOS-evoked electrical activity in brain organoids with external G-interfaces (top and central panels) and control brain organoids (bottom panel). A representative zoom-in raster plot shows electrical activity after a single light pulse (**c**, center). **d** Normalized mean firing rates (MFR) in externally G-interfaced and control organoids before, during, and after GraMOS (approximately 10 G-interfaced and 10 control organoids on 64 electrodes each). Data are shown as violin plots: center lines = median, red squares = mean, and individual data points = circles ($P = 1.1052 \times 10^{-85}$ for "before" vs "GraMOS"; $P = 5.3556 \times 10^{-82}$ for "GraMOS" vs "after"; two-way ANOVA). **e** Lag times of GraMOS-evoked neuronal spikes in externally G-interfaced brain organoids. **f** MFR fold

increase per active electrode in externally G-interfaced and control organoids before, during, and after GraMOS ($n > 100$ per group). **g** Population histogram of MFR fold increase in externally G-interfaced brain organoids ($n > 100$ per group). **h** Raster plots showing GraMOS-evoked electrical activity in brain organoids with internal G-interfaces (top and central panels) and control brain organoids (bottom panel). **i** Normalized MFR in internally G-interfaced and control organoids before, during, and after GraMOS (approximately 50 G-interfaced organoids on 253, 153, 135 electrodes, respectively; approximately 20 control organoids on 54, 54, 34 electrodes). Data are shown as violin plots: center line = median, red mean = mean, and individual data points = circles ($P = 1.0183 \times 10^{-7}$ for "before" vs "GraMOS"; $P = 3.2932 \times 10^{-5}$ for "GraMOS" vs "after"; two-way ANOVA). **j** Lag times of GraMOS-evoked neuronal spikes in internally G-interfaced brain organoids. **k** MFR fold increase per active electrode in internally G-interfaced and control brain organoids before, during, and after GraMOS ($n > 100$ per group). **l** Population histogram of MFR fold increase in internally G-interfaced brain organoids.

distributed within the organoid, light stimulation can engage deeper and more diverse circuits, trigger reverberating activity across multiple layers, and promote nonlinear interactions between distant parts of the network. This provides strong evidence for the critical role of 3D stimulation strategies in reproducing brain-like activity patterns in organoids.

To assess how long-term GraMOS can accelerate the development of brain cortical organoids, we exposed G-interfaced 4-month-old organoids to 2-Hz pulsed light training for 2 weeks, then monitored their electrical activity—alongside non-stimulated controls—for three additional weeks (Fig. 7a). Light-trained organoids exhibited higher mean firing and bursting rates at the start of the monitoring period and

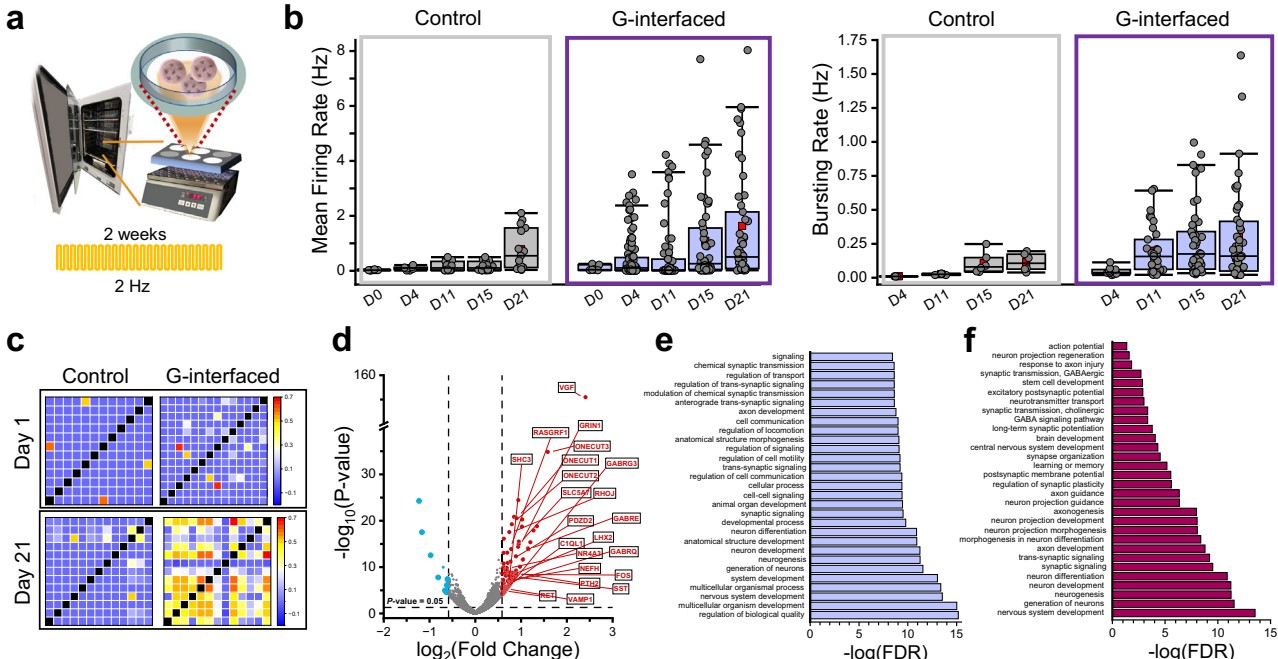

**Fig. 7 | GraMOS-driven maturation of G-interfaced brain cortical organoids via long-term optical stimulation. a** Schematic for GraMOS-enabled activity-dependent enhancement of maturation of G-interfaced brain cortical organoids. **b** Mean firing rates (left) and bursting rates (right) per electrode in G-interfaced (blue bars) and control (gray bars) brain organoids at different time points after long-term GraMOS (50 G-interfaced and 20 control organoids). Data presented as box plots: center line = median, red squares = mean, box = upper/lower quartiles, whiskers = 5th–95th percentiles, and individual points = circles. **c** Representative STTC matrices for MEA recordings from G-interfaced and control brain organoids on Day 1 and Day 21 after light training. **d** Volcano plot of DEGs between GraMOS-trained G-interfaced and control brain organoids. Cutoffs are at +/− 0.6 log₂ fold change and P-values < 0.05. Labeled DEGs are linked to brain developmental processes. P-values were adjusted for multiple comparisons using the Benjamini−Hochberg false discovery rate (FDR) correction method. **e** Top 30 GO analysis results by FDR. **f** Top 30 GO analysis Neuronal metrics by FDR. Enrichment score was calculated as the -log₁₀(FDR).

demonstrated faster gains in activity (Fig. 7b). Since mature networks typically show increased connectivity and synchrony, we analyzed these parameters as well. To quantify the degree of synchrony between neuronal spike trains while avoiding the issues of spike rate dependency, we calculated the Spike Time Tiling Coefficient (STTC) for all active electrodes. Unlike Pearson or cross-correlation methods, STTC provides a firing-rate-independent measure of synchrony. STTC matrices for Day 1 and Day 21 revealed a notable increase in synchronous spiking in light-trained G-interfaced organoids, but not in controls (Fig. 7c).

To identify gene expression changes due to long-term neuronal stimulation, we performed bulk RNA sequencing using 5-month-old light-trained G-interfaced and control brain organoids after 3 months of light training (Fig. 7d). Long-term stimulation of G-interfaced brain organoids affected 423 genes. DEGs upregulated in light-trained G-interfaced brain organoids are linked to neuronal differentiation, synaptic plasticity, neurogenesis, neuronal migration, and axonal development (*VGF, ONECUT1-3, RHOJ, FOS, RET*). More than half of top upregulated genes were related to increased synaptic transmission and neurotransmitter-gated receptors (*GRIN1*, a glutamatergic NMDA receptor subunit; *GABRE, GABRQ, GABRG3*, subunits of GABA_A receptors; *GLRA3*, a glycine receptor subunit; *VAMP1*, a SNARE protein involved in synaptic vesicle exocytosis and neurotransmitter release).

To determine which pathways were affected in the most significant way during light training of G-interfaced brain cortical organoids, we performed the Gene Ontology (GO) analysis. Among the upregulated pathways are synaptic signaling and transmission, neurogenesis, neuronal development and differentiation, axon development, multicellular organism development, signal transduction, and cell-to-cell signaling. There were significantly fewer downregulated genes, and the extent of downregulation was significantly smaller.

These results demonstrate that long-term GraMOS-supported neuronal activation can dramatically enhance neurodevelopment in G-interfaced brain organoids (Fig. 7e, f).

## Robotic control by GraMOS-activated G-interfaced brain organoids

Human brain cortical organoids hold significant promise for the development of future brain-computer interface systems. Achieving this goal requires: (a) enabling brain organoids to perceive and respond to environmental cues—such as detecting and reacting to light—and (b) integrating their neural activity with robotic actuators in a functional feedback loop.

To explore this possibility, we developed a closed-loop platform composed of a quadruped bionic robot (PuppyPi4, LewanSoul), the Maestro Pro™ MEA system with an integrated Lumos optical stimulation module (Axion Biosystems), and a computing unit running custom control software. This platform was orchestrated by Cogniborg, a custom Python-based software suite incorporating a Python API with custom C ++ and OpenCV libraries for efficient signal analysis and robotic control. Communications between the Maestro Pro™ system and the robot were facilitated via high-speed Wi-Fi protocol (up to 1.8 Gbit/s).

In the experimental setup, the PuppyPi4 robot was tasked with navigating a test course containing clearly detectable vertical obstacles. The robot was equipped with a time-of-flight (ToF) LiDAR sensor to monitor its surroundings. When an obstacle was detected at a distance of ~25 cm, the robot's Raspberry Pi unit transmitted an alert signal to the Maestro Pro™ system. This triggered a predefined optical stimulation protocol (475 nm, 3.9 mW/mm², 1 Hz, 10-ms pulses) applied to G-interfaced brain cortical organoids cultured on a 48-well MEA plate.

Electrophysiological signals from the organoids were recorded and filtered using AxIS Navigator (Axion Biosystems) with a 200 Hz high-pass and 3000 Hz low-pass Butterworth filters. Spike detection was performed by the Cogniborg software using a thresholding algorithm set at three standard deviations above baseline noise. In experimental runs, G-interfaced organoids exhibited rapid and pronounced increases in neural activity in response to light stimulation, including elevated firing rates and activation across multiple electrodes. When the mean firing rate—averaged over at least eight active electrodes (≥50% of all electrodes in a well)—increased by at least twofold compared to baseline, the Cogniborg software translated this activity into a command for the robot to perform an avoidance maneuver. After the light stimulation ceased, organoid activity returned to baseline levels. Accordingly, the robot resumed forward navigation.

To validate platform functionality, we conducted ten independent experimental trials using prerecorded organoid activity datasets collected under this stimulation protocol. In all ten trials, light-evoked activation of G-interfaced organoids was successfully detected and processed within a low-latency window, resulting in the successful execution of the avoidance behavior in 100% of cases. Supplementary Movie 4 illustrates a representative experiment using a split-screen format to display organoid MEA activity alongside the robot's motion. As a negative control, ten trials were repeated using datasets from control organoids without incorporated G-interfaces (i.e., lacking light sensitivity), none of which resulted in avoidance behavior—thereby supporting the necessity of light-evoked G-interfaced organoid activity for triggering robotic actions. In addition to evaluating behavioral outcomes, we quantified the temporal dynamics of platform response time. The total combined time for response detection, command generation, signal transition to the robot, and command execution did not exceed 50 ms.

This work presents a close-loop functional interface between brain organoids and robotic actuators mediated by optical stimulation and electrophysiological monitoring. These findings offer a proof of concept for developing neuro-biohybrid systems that may eventually enable coupling of neural networks with robotic agents in adaptively responsive settings.

## Discussion

Light-controlled neuromodulation is essential for advancing fundamental neuroscience research, developing innovative therapeutic strategies for brain disorders, and creating more adaptive, environmentally responsive brain-computer neuroengineering systems. Here, we present a nanotechnological platform that leverages the unique physicochemical properties of graphene, including its ability to efficiently convert light into electricity[14,15,17], to facilitate remote, safe, non-invasive light-controlled neuromodulation without any genetic or structural modifications. GraMOS actuators can function in either freestanding or supported configurations. Freestanding GraMOS actuators are among the most sought-after nanomaterials for neuroscience[51] due to their versatility, scalability, and remote-control capabilities. However, like all dispersible freestanding actuators, their spatial placement is constrained by the stochastic nature of their deposition. In contrast, supported GraMOS actuators provide enhanced spatial control, as they can be fabricated into various shapes using established techniques such as lithography, inkjet printing, and electrospinning.

In short-term studies, we demonstrated that GraMOS could trigger and modulate neuronal spiking activity in 2D neuronal cell models and 3D brain cortical organoids at single-cell and network levels. We showed that, due to the high optical transparency of graphene, GraMOS can be combined with optical monitoring methods, supporting non-genetic all-optical interrogation of neuronal networks. To showcase this capability, we utilized GraMOS to functionally phenotype 2D hiPSC-derived neurons on 2D G-substrates, allowing the characterization of human cell models of Alzheimer's' disease at very early stages of neurodevelopment when neurons do not yet exhibit spontaneous activity.

Unlike traditional 2D cell models, brain cortical organoids possess a 3D architecture that enables more advanced interfacing with GraMOS structures. We explored two interfacing configurations: external, where a large G-interface is placed on top of a fully-formed brain organoid, and internal, with numerous small G-interfaces dispersed within a developing organoid from the start. Optical stimulation efficiency was higher in the external configuration, likely due to the greater surface area of G-interfaces, making it more suitable for future neuroengineering applications. However, the internal configuration allows for the activation of neurons throughout the organoid, creating more intricate activation patterns.

Long-term studies are usually more demanding than short-term ones, as neurons must endure the structural and process-related demands required for effective neuromodulation. Nonetheless, long-term neuromodulation is vital, as neurodevelopment, memory formation, and disease treatment occur over extended periods. To showcase GraMOS performance in long-term studies, we successfully employed GraMOS to accelerate the maturation of 2D hiPSC-derived neurons by exposing them to repeated optical stimulation over a 4-week period to leverage activity-dependent maturation processes. Brain organoids with internally dispersed rGO flakes present a more challenging target for GraMOS-enabled accelerated maturation because the interface area between the randomly dispersed graphene materials and neurons inside organoids is considerably smaller than that in 2D G-interfaced hiPSC-derived neuronal cell cultures. Nevertheless, by optically stimulating G-interfaced developing brain organoids for 4 weeks, we successfully increased electrical activity levels and enhanced the complexity of neuronal networks. Due to a dynamic interplay of numerous factors affecting the maturation, in the future, a careful selection of light stimulation parameters tailored to different developmental stages over several months will be essential to achieve adult-like maturation levels in hiPSC-derived neurons and to do so faster than it naturally occurs within a human being.

With its excellent biocompatibility, structural stability, and operational safety in both short-term and long-term studies, GraMOS offers significant advantages over existing optical actuators for neuromodulation applications. The majority of these actuators operate via a photothermal optocapacitive mechanism[52], including gold nanoparticles[5], amorphous silicon mesostructures[6], and silicon nanowire-templated 3D fuzzy graphene with random out-of-plane grown graphene flakes (NT-3DFG)[7]. These actuators convert very-high-intensity light ($3–240\ kW/cm^2$) into localized heat spots on the cell membrane, causing a rapid temperature jump of kilokelvins per second, resulting in the temperature-dependent changes in membrane capacitance[41]. This triggers capacitive currents[53,54] across the membrane, leading to membrane depolarization and action potential generation[55]. Generating high-temperature gradients in neurons for the sole purpose of triggering action potentials does not appear to be a viable long-term strategy, because (a) elevated temperatures can impact the expression profiles of various proteins[56], changing surviving neurons, and (b) a tolerable temperature increase in neurons is less than ~1.5 °C, because prolonged exposure to high temperatures may lead to cellular stress, protein denaturation, and potential irreversible damage to neurons.

There are no thermal effects during GraMOS due to strong electron-electron and weak electron-phonon coupling in graphene[14,15]. Instead, GraMOS operates via non-Faradaic capacitive effects[57] by charging and discharging an electrical double layer near the graphene-electrolyte-neuron interfaces without involving photoelectrochemical reactions. While electrochemical effects cannot be entirely dismissed, their contribution seems to be minimal, as the pH values of cell culture

media with immersed GraMOS interfaces remained steady during 30 min of light exposure[33].

Critically, light intensities required for GraMOS are ~100–1000 times lower than light intensities needed for triggering thermal optocapacitive effects in cells[41,58]. Unlike an optically transparent GraMOS platform, a photothermal graphene-based NW-3DFG platform absorbs almost all incident light due to light trapping by the densely packed out-of-plane graphene flakes grown in all directions on silicon nanowires, leading to significantly enhanced broadband absorption by NT-3DFG[59] and their high photothermal efficiency. NW-3DFG requires extremely high-intensity light to operate[7], when the laser powers of ~3 kW/cm² (635 nm laser with a 20 µm spot size) were used to produce local temperature increases from $1.86 \pm 0.03$ to $6.74 \pm 0.07$ K. While these laser powers are one to two orders of magnitude lower than required for optical stimulation using gold nanoparticles and silicon-based nanomaterials[5,6,60–62], they are still three orders of magnitude higher than the light intensity needed for GraMOS. Another nanotechnology platform, coaxial p-type/intrinsic/n-type silicon nanowires with atomic gold on their surfaces[8], employs a somewhat safer Faradaic electrochemical mechanism, yet it still demands similarly high light intensities. Recent studies introduced nano-scale optoelectrodes that combined zinc porphyrin nanostructures with either $TiO_2$ coating or single gold atom centers[57,63]. These optoelectrodes were shown to activate neurons via non-Faradaic or Faradaic mechanisms, respectively, using relatively low light intensities.

Technologically, the GraMOS platform offers several key advantages for efficient neuromodulation: (a) it operates without requiring genetic or structural modification of cells; (b) its capacitive operating mechanism ensures safe physiological stimulation across diverse cell types; and (c) due to graphene's broadband absorption[10,11], it can be activated by a wide range of light wavelengths, allowing for multiparametric neuromodulation and seamless integration with other optical technologies.

Our study represents a significant step toward unlocking the potential of graphene materials in neuroscience, nanotechnology, and neuroengineering[64], paving the way for innovative interfacing strategies with increasingly complex neuronal tissues, all the way to the brain. The ability to modulate activity and enhance the maturation of brain cortical organoids opens unique opportunities to use them as predictive models for developing therapies targeting neurodegenerative and neurodevelopmental disorders, where disrupted neuronal connectivity may impair the capacity to perceive and respond to external signals. Moreover, this platform can be adapted to tackle challenges in tissue engineering by enabling noninvasive high-precision control of stimulation in engineered tissues. Finally, integrating biological neural networks with robotic systems may offer an alternative means of exploring how biological adaptability and learning mechanisms could complement machine-based computation, with possible implications for future developments in artificial intelligence and machine learning.

## Methods
### Fabrication of graphene interfaces
An aqueous dispersion of chemically converted graphene (reduced graphene oxide (rGO)) was synthesized using green chemistry methods[27,28]. Specifically, it was synthesized from graphene oxide (Graphenea Inc.) by chemical reduction using L-ascorbic acid. First, 50 mL of aqueous GO dispersion (0.4 mg/mL) was sonicated for 1 h at room temperature. To enable the colloidal stability of aqueous GO dispersions, the pH was adjusted to ~10 using 25% ammonia solution. GO dispersions were transferred into a glass beaker and agitated with a stir bar (225 RPM) following the addition of 32 mg of ascorbic acid. The GO-to-rGO reduction process was monitored by UV-Vis spectroscopy using a NanoDrop 2000 spectrophotometer (Thermo Fisher

Scientific). After 24 h, an rGO aqueous dispersion was aliquoted and washed four times through a repeated process of centrifugation, decanting, and resuspension in deionized (DI) water to the initial volume. Poly(vinylpyrrolidinone) (PVP) (0.05% wt/vol) was added to the final rGO dispersion, and the pH was adjusted to 10 using ammonia. All chemicals were from Sigma Aldrich.

To prepare dispersible free-standing G-interfaces, a stock of rGO aqueous dispersion (1 mg/mL) was sonicated for 30 min, centrifuged for 15 min, dispersed in cell culture media up to desired concentrations, and sonicated again for 10 min. The structural parameters of rGO flakes were estimated using optical microscopy (Olympus IX71 microscope) and subsequent image analysis (ImageJ software) by determining the dimensions of region of interests (ROIs) corresponding to single flakes, and their optical transparency as compared to ROIs located just outside flakes.

To fabricate substrate-based G-interfaces, glass coverslips (12 mm or 5 mm in diameter, VWR) were thoroughly cleaned using the 5% Triton X-100 solution for 1 h, washed in DI water and ethanol, placed into KOH/hydrogen peroxide (1:1) for 1 h, washed in DI water again, and finally dried in 60 °C oven for 10 min. Glass coverslips were then coated using an rGO dispersion at concentrations ranging from 0.1 to 0.5 mg/mL by either drop casting or multistep fine-mist spray coating, dried for 1 h at 200 °C, and placed in a cell culture hood for overnight sterilization under UV light. In experiments with hiPSC-derived neurons, we predominantly used G-substrates with optical transmittance in the 70–80% range. This corresponds to ~$10 \pm 2$ graphene layers, based on the established fact that a single layer of graphene has an optical transmittance of 97.7%[10], and that optical absorption scales linearly with the number of layers in multilayer graphene[65].

### Characterization of graphene materials
Raman spectroscopy was performed using an NTEGRA Spectra instrument (NT-MD) with 532 nm excitation through a 100× objective. Raman spectra were acquired from five different locations across six samples using 0.5 neutral density (ND) filter and an acquisition time of 30 s.

Fourier-transform infrared (FT-IR) spectra were recorded using an IRTracer-100 (Shimadzu) within the spectral range of 400–4000 cm⁻¹. For the measurements, 5 µL aliquots of aqueous rGO dispersions were deposited on the objective, and each sample was measured three times.

The X-ray diffraction (XRD) patterns of rGO samples were recorded using an X-ray diffractometer X'Pert Pro (PANalytaca) operating at 40 mA, 40 kV with Cu Kα radiation source of wavelength 1.5406 Å. The measurements were conducted over a 2θ range of 5°–50° with a step size of 0.005° and a scan speed of 0.3 °/min. For sample preparation, rGO solution was drop-casted onto a Si wafer, followed by drying at 100 °C for 10 min and subsequent annealing at 270 °C for 4 h to remove all water molecules.

UV-Vis spectra of GO and rGO dispersions were acquired using a NanoDrop 2000 UV-Vis spectrophotometer (Thermo Fisher Scientific), and UV-Vis spectra of G-substrates were acquired using a Lambda 1050 spectrophotometer (Perkin Elmer) equipped with a 150 mm InGaAs integrating sphere. Measurements were performed over a wavelength range of 400–800 nm, using a clear glass coverslip as the reference.

The rGO-based field-effect transistor (FET) device was fabricated on an interdigitated electrode (IDE) array. The IDE, with an inter-electrode spacing of channel length of 50 µm and a width of 350 µm, was fabricated on a Si/SiO₂ substrate using UV lithography, metal deposition (Au/Cr (100 nm/ 5 nm)) by an e-beam evaporator, and lift-off techniques (Supplementary Fig. 6b). The IDEs served as the source and drain electrodes for the FET device. An rGO dispersion (0.8 g/L) was deposited onto the IDE via drop-casting, followed by drying at 70 °C for 10 min and annealing at 175 °C for 30 min to ensure a

conductive thin film. Electrical date were acquired using a Keysight B2902B source meter (Keysight Technologies). A poly-dimethylsiloxane (PDMS) liquid bridge, with a well at the center, was filled with 0.1 M phosphate-buffered saline (PBS) solution. A gate potential ($V_{GS}$) was applied using an Ag/AgCl pellet reference electrode immersed in the solution, and the gate voltage was swept between -1 V and +1 V, while drain-source voltage ($V_{DS}$) was constant 0.2 V. Time dependent photocurrent was measured at $V_{GS}$ of 0.1 V and $V_{DS}$ of 0.2 V. The device was exposed to 365-nm light (uvBeast) with a radiant intensity of 1.8 mW/cm². To isolate the UV exposure to Ag/AgCl electrode the device area and Ag/AgCl electrode was shielded by shielding material, except rGO channel area.

To characterize the morphology of rGO flakes, atomic force microscopy (AFM) and scanning electron microscopy (SEM) were employed. For the measurement, rGO solution (0.2 g/L) was spin-coated onto indium tin oxide (ITO)-coated glass substrates at 2000 rpm for 30 s and baked at 100 °C for 10 min, followed by annealing at 270 °C for 3 h. AFM was conducted in tapping mode using an Asylum MFP-3D system with an Optus 160AC-NA AFM tip (Nanoandmore) operating at a resonance frequency of 300 kHz. The AFM data were processed using Gwyddion software to extract surface features, including the height profile and average thickness of the rGO flakes. SEM imaging was performed using an FEI Magellan 400 system under high vacuum conditions with an accelerating voltage of 20 kV. The lateral dimensions and surface areas of rGO flakes were determined from SEM images analyzed using Fiji, an open-source distribution of ImageJ with built-in plugins for image processing.

Amplitude mode-Kelvin Probe Force Microscopy (AM-KPFM): The rGO solution (0.2 g/L) was spin-coated onto Si substrate at 2000 rpm for 30 s and baked at 100 °C for 10 min, followed by an annealing at 270 °C for 3 h. AM-KPFM measurements were conducted using a Cypher ES system (Asylum Research, Oxford Instruments) in tapping mode. The conductive microcantilever was used with a spring constant of 2 N/m and a resonance frequency of 70 kHz (AC240TM-R3, Oxford Instruments). During the measurements, a voltage of 3 V was applied to the AFM tip. The KPFM data were processed using Gwyddion software to analyze surface potential variations between the rGO flakes, and Si substrate.

## Cell Culture

Primary neuronal cultures were prepared following standard protocols with some modifications[66]. Cerebrocortical and hippocampal cultures were derived from E17 Sprague Dawley rats or PO-P1 C57BL/6 J mice. Following enzymatic treatment (papain, 200 U/ml; 30 min, 37 °C) and mechanical dissociation of brain tissues in minimum essential medium (MEM, Invitrogen), cells were plated on poly-L-lysine-coated glass coverslips in DMEM with Ham's F12 and heat-inactivated iron-supplemented calf serum (HyClone) at a ratio of 8:1:1. Cells were grown to 50% confluence at 37 °C in a humidified 5% $CO_2$/95% air atmosphere in Neurobasal-A media (ThermoFisher Scientific Cat10888022) supplemented with Glutamax (ThermoFisher Scientific Cat35050061), Pen/Strep (ThermoFisher Scientific Cat10378016), and B27 supplement (ThermoFisher Scientific Cat17504044). Neuronal recordings were made 17−28 days in culture.

Human induced pluripotent stem cells (hiPSCs) were generated by four-factor reprogramming and differentiated to neural progenitor cells (NPCs)[67,68]. NPC cultures were placed on plastic dishes coated with 20 μg/ml poly-L-ornithine overnight (Sigma) followed by 5 μg/ml laminin (Sigma) for at least 2 h inside incubator or directly on graphene-coated (G-coated) glass coverslips that were further coated with 100 μg/ml poly-L-ornithine followed by 10 μg/ml laminin. NPCs were maintained in NPC base media containing DMEM-F12+Glutamax (Life Technologies) supplemented with N2 (Life Technologies), B27 (Life Technologies), Pen/Strep and 20 ng/ml fibroblast growth factor (FGF) (Millipore). Media was changed every 2–3 days. Once NPCs

reached confluency, FGF was withdrawn from the media and maintained for 5 weeks changing media twice per week. After 3 week differentiation neurons were FACS-purified using a cell surface-antigen signature: differentiated neurons were detached using a 1:1 mixture Accutase/Accumax (Innovative Cell Technologies), and stained with CD184, CD44, and CD24 antibodies (BD Biosciences) as described[67]. Neurons negatively stained for CD184 and CD44 and positively stained for CD24 were selected and plated on poly-ornithine/laminin treated G- coverslips in NPC base media supplemented with 0.5 mM dbCAMP (Sigma), 20 ng/mL BDNF and 20 ng/mL GDNF (Peprotech).

Brain cortical organoids were generated according to previously published protocols[47]. Briefly, hiPSCs were maintained with mTeSR on Matrigel-coated tissue culture dishes. iPSCs were dissociated into single cells using Accutase and plated in 6-well plates in mTeSR with 5 μM ROCK inhibitor (Y-27632) and SMAD inhibitors (SB431542 (10 μM), Dorsomorphin (1 μM) (all chemicals are from Sigma)). At this point, graphene dispersion (10−100 μg/mL) was added to wells for generation of graphene-interfaced organoids. Plates were continuously shaken with an orbital platform (95 rpm) from this point on. Media was changed daily through day 6. From days 7–11, media was changed every other day with M1 media [Neurobasal Medium (Life Technologies, 21103049) + 1% GlutaMAX (ThermoFisher Scientific, 35050061) + 1% Gem21 NeuroPlex (Gemini Bio, 400-160-010) + 1% N2 NeuroPlex (Gemini Bio, 400-163-005) + 1% NEAA (ThermoFisher Scientific, 11140050) + 1% PS (Thermo-Fisher Scientific, 15140122) + 10 μM SB + 1 μM Dorso] and embryoid bodies were split between new wells approximately once during this period to prevent overcrowding. From days 12–18, media was changed every day with M2 media [Neurobasal Medium + 1% GlutaMAX + 1% Gem21 NeuroPlex + 1% NEAA + 1% PS] + 20 ng/mL FGF2 (Peprotech, 100−25). From days 19−25, media was changed every other day with M2 + 20 ng/mL FGF2 + 20 ng/mL EGF (PeproTech, AF-100-15). From days 26−29, half media changes were performed with M3 media [M2 medium + 10 ng/mL BDNF (PeproTech, 450-02) + 10 ng/mL GDNF (PeproTech, 450−10) + 10 ng/mL NT-3 (PeproTech, 450-03) + 200 μM L-ascorbic acid (Sigma-Aldrich, A4403) + 1 mM dibutyryl-cAMP (Stem-Cell Technologies, 100−0244)] + 20 ng/mL FGF2. On days 30−35, media was changed every 3−4 days with M3 media. On days 36−42, media was changed every 3−4 days with M2.5 media [M3 medium with half the concertation of factors].

Independent brain organoids were treated as biological replicates, as each organoid represents a separately developed neural system with inherent biological variability.

G-substrates (3−5 mm PDMS circles or 5 mm glass coverslips) were coated with 100 μl/ml poly-L-ornithine and 10 μg/ml laminin, and then placed inside a glass-bottom 96-well plate (Greiner). Organoids (2-3 per coverslip) at 6 weeks of age were plated on G-substrates and maintained in Media 2 for 1−2 months. Media were changed twice a week. Before imaging, Media 2 was replaced with BrainPhys media (StemCell Technologies) supplemented with 1% N2, 2% Gem21, and 1% pen/strep. All cell cultures were routinely tested for mycoplasma using the MycoAlert Mycoplasma Detection Kit (Lonza). Only negative samples were used in the study.

Allocation of all samples (hiPSC-derived neurons and hiPSC-derived brain cortical organoids) into different experimental groups was always random.

## Light-driven maturation of hiPSC-derived neurons and brain organoids

hiPSC-derived neurons were plated onto G-coverslips and placed on top of the LightKick illumination module (Nanotools Bioscience) inside a cell culture incubator for 4 weeks with media changes every 3 days. All neurons were exposed to wide-field light stimulation patterns for 4 weeks (intensity: 1.9 mW/mm², frequency: 1 Hz for the first 2 weeks, and 2 Hz for the next 2 weeks), and these patterns were

dictated by a control unit located outside an incubator. Control hiPSC-derived neurons were never exposed to light.

Six-well cell culture plates with G-interfaced brain cortical organoids on a shaker were exposed to light pulses (intensity: 1.9 mW/mm², frequency: 2 Hz) from the LightKick illumination module placed on top. Control brain cortical organoids were never exposed to light.

### Scanning Electron Microscopy (SEM) of hiPSC-derived neurons on graphene

To prepare samples for SEM, neurons were first washed with 0.1 M phosphate buffer (pH 7.4), fixed with 4% formaldehyde solution for 2 h at room temperature, and washed with the same buffer three times for 5 min each. Following dehydration with graded series of alcohol (35% ethanol−10 min, 50% ethanol−10 min, 75% ethanol−10 min, 95% ethanol−2 changes in 10 min, 100% ethanol−3 changes in 15 min), all samples were freeze-dried in a vacuum chamber, and coated with sputtered iridium by an Emitech Sputter Coater (K575X) 8 s at 85 mA. SEM images of 2D neuronal cultures were acquired using the XL30 ESEM-FEG (FEI) at the working distance of 10 mm while using the 10-kV accelerating voltage. SEM images of brain cortical organoids were acquired using a Zeiss Sigma 500 scanning electron microscope (a 3-kV accelerating voltage, a 10-mm working distance).

### Cell Viability

The cell viability was quantified using the LIVE/DEAD® Viability/Cytotoxicity Kit (Thermo Fisher Scientific) containing membrane-permeable calcein-AM (for detection of enzymatically active live cells, green fluorescence) and membrane impermeable Ethidium homodimer-1 (EthD-1) (for detection of dead cells with compromised membrane, red fluorescence). We used the LIVE/DEAD® Viability/Cytotoxicity Kit instead of an MTT assay because this Kit non-destructive method employs fluorescent dyes, enabling real-time visualization of live and dead cells. In contrast, the MTT assay requires cell lysis and measures metabolic activity, which is an indirect indicator of viability and may fail to distinguish between live but metabolically inactive cells and truly dead cells, especially when conductive cell substrates are present.

To compare the cell viability in 2D neuronal cell cultures on graphene and control glass coverslips, hiPSC-derived neurons were incubated with 2 μM calcein AM, 4 μM EthD-1, and 2 μM Hoechst for 30 min at 37 °C and 5% $CO_2$. After samples were washed with Hank's Balanced Salt Solution (HBSS) 3 times, live-cell imaging was performed using an Olympus IX71 fluorescent microscope, a 20x lens, and a standard filter set (fluorescein, rhodamine, and DAPI filters for calcein, EthD-1, and Hoechst, respectively). The images (three replicates and three images per condition) were analyzed using Fiji image analysis software by a scientist blinded to experimental conditions. The cell viability was presented as the percentage of the number of calcein-positive cells divided by the total number of cells.

To assess the cell viability in 3D in control and G-interfaced brain cortical organoids, calcein AM (2 μM) and EthD-1 (4 μM) were added to wells with organoids and incubated for 30 min at 37 °C and 5% $CO_2$. Following the triple wash in M2, organoids were imaged using a Dragonfly 600 Laser Confocal System (Oxford Instruments) and a 10x lens by acquiring either 3D Z-series of brain organoids or five representative focal planes from each organoid. All images were analyzed using Imaris Image Analysis Software v10.1.1 (Oxford Instruments) by a scientist blinded to experimental conditions. The cell viability was presented as the percentage of the number of calcein-positive cells divided by the sum of calcein-positive and EthD-1-positive cells.

To ensure accurate and unbiased quantification when using the LIVE/DEAD® Viability/Cytotoxicity Kit, scientists blinded to the experimental conditions analyzed the fluorescent images.

### Electrophysiology

Whole-cell current-clamp recordings were performed using the previously validated experimental protocol[66]. Briefly, coverslips with cells were placed in an experimental chamber (RC-25-F, Warner Instruments) filled with an extracellular solution consisting of 150 mM NaCl, 5.4 mM KCl, 1.8 mM $CaCl_2$, 1 mM $MgCl_2$, 1 mM Na-pyruvate, 15 mM glucose, and 10 mM Hepes (pH 7.4). Patch pipettes with a final tip resistance of 3 to 6 MΩ were filled with a solution consisting of 150 mM KCl, 5 mM NaCl, 5 mM MgATP, 10 mM Hepes, 5 mM EGTA (pH 7.2). All recordings were acquired using a Digidata 1322 interface, an Axopatch 200B amplifier, and pClamp software (Molecular Devices). The data were digitally sampled at 10 kHz and filtered using an eight-pole Bessel analog low-pass filter at 2-kHz cutoff frequency.

Whole-cell voltage-clamp recordings were performed at room temperature using standard recording protocols. For these experiments, we used patch pipettes with fire-polished tips with a tip resistance of 2.2−4.5 MΩ. Neurons were optically activated via graphene substrates using a Melles Griot 30 mW 488-nm ion laser. All electrophysiological data were acquired using a Digidata 1322 interface, an Axopatch 700B patch-clamp amplifier, and pClamp software (Molecular Devices), digitally sampled at 20 kHz. In a subset of electrophysiological experiments, neurons were optically activated using 100 mW 561-nm laser equipped with the variable density filter, installed on an inverted Laser Scanning Microscope Zeiss LSM 780.

Alternatively, we utilized the PhotonMaker 7-LED light system (Nanotools Bioscience), capable of providing illumination at 398 nm, 452 nm, 500 nm, 515 nm, 601 nm, and 638 nm, with independent control over the intensity and temporal patterns of each individual light channel. In some experiments, the electrical activity was acquired using HEKA EPC10/2 patch-clamp amplifier in different recording modes with sampling rate 20 kHz and was filtered using low-pass 2.9 kHz digital filter.

### Temperature measurement

Temperature values before, during, and after light illumination of graphene interfaces were measured using a thermocouple (#5TC-TT-K-30-36, Omega) and monitored using a non-contact Digital Laser Infrared Thermometer Mestek 800 C. Alternatively, the temperature values were directly determined using a Digital Laser Infrared Thermometer Mestek 800 C.

Additionally, we took advantage of the fact that local temperature changes can be evaluated by monitoring the resistance of a patch pipette[41,42] as previously described[33]. Briefly, a recording chamber had the following solution: 111 mM NaCl, 3 mM KCl, 25 mM $NaHCO_3$, 1.1 mM $KH_2PO_4$, 11 mM Glucose, 2 mM $CaCl_2$ and 3 mM $MgSO_4$. Patch pipettes were filled with a 4 mM NaCl, 128 mM KGlu, 10 mM HEPES, 1 mM Glucose, 0.0001 mM $CaCl_2$ and 5 mM ATP aqueous solution, and G-coverslips were placed into a recording chamber. To detect the local temperature, we positioned a patch pipette ( ~5 MΩ) as close as possible to a G-coated coverslip, applied current pulses (10 mV/MΩ) using an Axopatch 200B amplifier (Molecular Devices), and measured the pipette resistance while continuously illuminating G-coverslips (535 nm, 3.7 mW/mm²). To construct a calibration curve for converting pipette resistance to temperature, the solution in the chamber was preheated to 50 °C, and then the solution temperature and pipette resistance were then monitored simultaneously as the solution cooled to room temperature. A linear calibration curve was fitted to determine $E_a$, the electrolyte's activation energy, from the slope of the resulting Arrhenius plot. Pipette resistance was converted to temperature values using the equation $T_i = [1/T_0 - R/E_a \times \ln(R_0/R_i)]^{-1}$, where $R$ is the gas constant, $T_0$ is the room temperature, $T_i$ is the temperature at a given time, $R_0$ is the resistance at $T_0$, and $R_i$ is the resistance at $T_i$.

## pH monitoring

G-coverslips were placed in an MS-512 recording chamber (ALA Scientific Instruments) and immersed into 100 µl of solution composed of 124 mM NaCl, 3 mM KCl, 3 mM CaCl$_2$, 1.5 mM MgCl$_2$, 26 mM NaHCO$_3$, and 1 mM NaH$_2$PO$_4$ (pH 7.2). This low-capacity electrolytic buffer is known to be suitable to monitor small pH changes in neuronal slices[69]. We continuously illuminated G-coverslips for 10-min using 535 nm light (3.7 mW/mm$^2$), and measured pH every 2 min using the Orion Micro Automatic Temperature Compensation Probe (928007MD, Thermo Fisher Scientific), which has a minimum sample size of 10 µl and a precision of pH 0.02.

## Reactive oxygen species assay

50 µg of CM-H$_2$DCFDA (5-(and-6)-chloromethyl-2′,7′-dichlorodihydrofluorescein diacetate, Thermo Fisher Scientific) is reconstituted in 173 µL of biological grade DMSO to make a 0.5 mM stock concentration. 3 µL of the stock solution was then added to 300 µL of phenol red-free cell culture media, to achieve a final working concentration of ~5 µM. We evaluated ROS levels in four experimental groups: control and G-interfaced 2-month-old WT83 brain cortical organoids, each with or without prolonged light stimulation (2 Hz, 1 h per day for 7 consecutive days, 1.9 mW/mm$^2$). Measurements were taken immediately after the final (7th) light stimulation session. After loading the CM-H2DCFDA, the plate is returned to incubation (37 °C, 5% CO$_2$, 90% humidity) for 1 h. The sample was then imaged using a Dragonfly 600 Laser Confocal System under incubation (37 °C, 5% CO$_2$), where the fluorescence was measured at an excitation/emission wavelength of 485/530 nm. To ensure accurate and unbiased quantification, a scientist blinded to the experimental conditions analyzed the fluorescent images.

## All-optical GraMOS-empowered assays

To perform calcium imaging assays, cells were incubated with a fluorescent calcium-sensitive indicator Fluo-4 (4 µM) and probenecid (1 mM) (both from Thermo Fisher Scientific) for 45 min at 37 °C, and were washed three times in HBSS prior to imaging. To avoid optical crosstalk between light signals for GraMOS (Ls) and light signals to excite calcium-sensitive fluorescent dyes (Le) (Fig. 3f), a light wavelength for GraMOS is selected outside the absorption spectrum of fluorescent dyes, and an intensity of fluorophore excitation light was selected below the threshold required for GraMOS. For example, for GraMOS-empowered all-optical calcium assays with Fluo-4, we can use two light wavelengths at different light intensities: (1) Fluo-4 excitation light at a subthreshold (low) intensity (488 nm, <0.5 mW/mm$^2$) and (2) a light of any wavelength outside fluorophore excitation spectrum (e.g., 605 nm, 2–10 mW/mm$^2$).

Wide-field all-optical calcium imaging experiments were conducted using a Zeiss Axiovert microscope equipped with a 20x lens and an Orca-Flash4.0 CMOS camera (Hamamatsu). Illumination was provided by a 7-LED light source PhotonMaker (Nanotools Bioscience) which could simultaneously generate two independently controlled signals, each with different wavelengths (638 nm for Ls and 452 nM for Le), intensities (80–100% for Ls and 10–15% for Le), and temporal illumination patterns. Raw movies were acquired at 30–50 frames per second (fps), and analyzed using Fiji image analysis software and customized plugins.

Alternatively, calcium imaging was performed using a Dragonfly 600 Laser Confocal System equipped with a 20x lens and an Andor Zyla 5.5 CMOS camera (Oxford Instruments). We excited Fluo-4 using a 488-nm laser, and initiated GraMOS using a 561-nm dye laser excitation. Laser intensity used for GraMOS was attenuated as needed using the neutral density filters. Raw image sequences were acquired at 50–100 fps, and analysis of calcium transients was performed using Fiji image analysis software using our customized plugins. In brain cortical organoids, the neuronal responsiveness to GraMOS was calculated as the ratio of neurons responding to a single-cell laser pulse to the total number of tested neurons targeted by the pulse.

## Neuronal network connectivity analysis

**Data preprocessing.** We preprocessed the raw videos using a truncated singular-value decomposition (SVD) approach to retain the most dominant modes and suppressing noise. We chose the number of modes based upon the Eigenspectrum. Pixels that did not show significant calcium fluctuation over time (as determined by the difference between maximum and minimum fluorescence) were excluded in further analyses to reduce computational load.

**Calculation of spike times.** For each retained pixel, we smoothed the fluorescence trace with a moving average filter and then applied a Hilbert transform to compute the instantaneous phase. Local minima of the phase signal with sufficient prominence correspond to rising calcium amplitude increases and were designated as spike events. The global minimum in each phase-trace represents the stimulus-evoked spike.

**Calculation of activation maps.** We assigned an activation time T(x,y) to each pixel, defined by the onset of its stimulus-evoked spike. Pixels that did not exhibit a clear event were assigned NaN, yielding a two-dimensional activation-time array.

## MEA recordings and analysis

After CytoView™ 6-well or 48-well MEA plates (Axion Biosystems) were coated with 0.07% PEI (Sigma-Aldrich) and 0.4 mg/mL laminin (Life Technologies), brain cortical organoids were added to the wells with M2 media and incubated for 7−14 days at 37 °C with media changes every 4−5 days. MEA recordings were performed at 37 °C with 5% CO$_2$ using a Maestro Pro™ MEA system and AxIS Software Spontaneous Neural Configuration (Axion Biosystems) with a customized script for band-pass filter (0.1 Hz and 5 kHz cutoff frequencies). Spikes were detected with AxIS software using an adaptive threshold crossing set to 5.5 times the standard deviation of the estimated noise for each electrode (channel). Light stimulation was enabled by a multi-well light delivery device Lumos controlled by the AxIS Navigator software (Axion Biosystems). Raw data were filtered in AxIS on-line using a 200 Hz Butterworth high-pass filter and a 3000 Hz Butterworth low-pass filter. Spikes were detected in AxIS on-line using peak detection with an adaptive threshold of 5.5 SDs from noise levels. To avoid the detection of overlapping spikes, detection was prevented for 2.16 ms after each peak.

For MEA analysis, the electrodes that detected at least 5 spikes/min were classified as active electrodes using Axion Biosystems' Neural Metrics Tool. Bursts were identified in the data recorded from each individual electrode using an inter-spike interval (ISI) threshold requiring a minimum number of 5 spikes with a maximum ISI of 100 ms. A minimum of 10 spikes under the same ISI with a minimum of 25% active electrodes were required for network bursts in the well. The synchrony index was calculated using a cross-correlogram synchrony window of 20 ms. Raw signals (sampled at 20 kHz) were imported into MATLAB (Mathworks). The LFP component was extracted by first low pass filtering the raw data (frequency cutoff of 500 Hz using 4th order Butterworth filter), and down-sampled to 1 kHz. For spike sorting, we first conducted the Principal Component Analysis (PCA) from the shape spikes for each electrode. After that, we apply a clusterization using K-means algorithm with Gap criterion for the first 5 principal components, and maximum number of clusters equal to 10. The analysis was done in Matlab, with a custom code, and using the function 'evalclusters'. To quantify the degree of synchrony between neuronal spike trains, we calculated the STTC (Spike Time Tiling Coefficient) values between active electrodes in any given MEA plate per performing analysis in Matlab / Python using a custom code.

To assess the correlation between stimulation time and the observed network activity response, we calculated the lag time between the first peak (above a threshold) occurring after each stimulation event. The lag time is defined as follows: Lag time (s) = 1st peak time (s)−previous stimulation time (s).

We evaluated the Lempel Ziv complexity (LZC) for each electrode by performing calculations in Matlab. The LZC is a measurement of the randomness of a series by measuring the number of distinct substrings and their occurrence rate. The LZC has been used to analyze biological signals as a complexity parameter, serving as a biomarker, classification and characterizing different neural states. The value of the normalized complexity is given by $LZC = \frac{c(n)}{n} * \log_2(n)$ where n is the length of the data, and c(n) is the raw complexity, which is the number of sub-strings.

## Whole brain organoid immunofluorescence staining and imaging

Organoids were rinsed with D-PBS and fixed in 4% paraformaldehyde (ThermoFisher Scientific, J19943K2) at 4 °C overnight (~16 h). After fixing, the organoids were washed three times with washing buffer, then incubated in permeabilization solution (PBST) on an orbital shaker at 150 rpm and 37 °C for 48 h. Next, the brain organoids were washed with washing buffer three times and placed in blocking buffer with gentle shaking at 4 °C overnight, and incubated with primary antibodies diluted with blocking solution at room temperature with gentle shaking for 3–4 days. Then, the brain organoids were washed three times and incubated with secondary antibodies diluted with solution in room temperature with gentle shaking for 3–4 days. One day before the secondary antibody incubation was completed, DAPI was added to the incubation solution. Then, the brain organoids were washed three times with washing buffer (four times if staining for vesicle proteins) and incubated in pre-warmed RapiClear (SunJin Lab, RC152002) overnight at room temperature with horizontal mixing. Lastly, all cleared immunostained organoids were transferred into a glass-bottom plate for imaging. Primary antibodies used were: FOXG1 (Millipore, MABD79; 1:500), DCX (Abcam, ab18723; 1:200), MAP2 (Abcam, ab5392; 1:1000), ZO-1 (Invitrogen, 33-9100; 1:100), S-OPSIN (Invitrogen, OSR00219W; 1:500), RCVRN (Millipore, AB5585; 1:2000), CRX (RD Systems, AF7085; 1:100), RHO (Abcam, ab5417; 1:200), VGLUT1 (Synaptic Systems, 135311; 1:100), and GAD65 + 67 (Abcam, ab11070; 1:200). Secondary antibodies (Invitrogen; 1:500) used were: Goat anti-Rabbit Alexa Flour 488 (A11034), Donkey anti-Mouse Alex Flour 555 (A31553), Goat anti-Rabbit Alexa Flour 647 (A21244), Donkey anti-Mouse Alexa Flour (A21202), Donkey anti-Mouse Alexa Flour 647 (A31571), Donkey anti-Sheep Alexa Fluor 647 (A21448), and Goat anti-Chicken Alexa Flour 647 (A21449).

Cleared whole organoids were imaged with a 40X glycerol objective and 63x oil objective on a Dragonfly 600 Laser Confocal System (Oxford Instruments) to match the refraction index of the RapiClear solution. Since organoids were fully cleared, we were able to capture the neural processes projects throughout the organoid with high resolution using the Z-stack features. The tile and stitch features were also utilized to help capture the organoids wholistically. The acquired image data were analyzed using Imaris Image Analysis Software v10.1.1 (Oxford Instruments). Deconvolution and background subtractions were applied to every channel to further differentiate positive signals and backgrounds through Imaris. Vesicle proteins and Nucleus were identified using the "surface" creation feature. Additionally, the machine learning feature in "surface" generation was used to increase the detection accuracy, where Imaris was trained and validated to recognize positive signals and avoid potential artifacts. Positive signals close to each other, such as two touching DAPI stains, were also recognized by enabling the "split touching" feature. To quantify neural processes such as axons and dendrites, we utilized the "filament" creation tool in Imaris to trace the processes. The nuclei are clear in our MAP2 staining and are used as starting points for the filament tracings. Since the number of DAPI-stained nuclei served as a good indicator for the size of the organoid, we normalized other data such as NeuN+ count and MAP2 length against the volume of DAPI positive stain. Vesicle proteins such as VGLUT1 and GAD65/67 were quantified using Imaris by quantifying spheres of a specific seed size (3.5 μm for DAPI+ nuclei and 0.2 μm for VGLUT+ and GAD65 + 67+ puncta) and voxels of a specific threshold (15.5 for DAPI+ nuclei and 25 for VGLUT+ and GAD65/67+ puncta). To eliminate the effect of backgrounds and artifacts, we only counted the vesicle proteins that co-localized with neuronal processes and then normalized the vesicle protein counts against the neural processes' length for further analysis. To ensure accurate and unbiased quantification of immunostaining, a scientist blinded to the experimental conditions analyzed the fluorescent images.

## RNA sequencing and analysis

Approximately 4-month-old hiPSC-derived brain organoids (n = 20 per condition) were transferred to conical tubes, rinsed three times with D-PBS to remove residual media and debris, and snap-frozen in liquid nitrogen to preserve RNA integrity. The frozen samples were then stored at −80 °C until further processing. RNA extraction was performed using Qiagen's RNeasy Plus Mini Kit, following the manufacturer's protocol. The quality and integrity of the extracted RNA were assessed using the Agilent TapeStation system. Only samples with an RNA Integrity Number (RIN) >7.0 were selected for library preparation to ensure high-quality sequencing results. Library preparation was conducted using the Illumina Stranded mRNA Prep Kit, which selectively enriches polyadenylated (poly-A) transcripts to capture mature mRNA. The prepared libraries were quantified and assessed for quality using Qubit 3.0 and TapeStation respectively before sequencing. Sequencing was performed on the Illumina NovaSeq X Plus platform using the 10B kit, generating paired-end 100 bp (PE100) reads with standard Illumina primers. Post-sequencing quality control, including adapter trimming and read filtering, was carried out using Rosalind, a cloud-based bioinformatics platform designed for RNA sequencing analysis. Subsequent differential gene expression analysis was also performed in Rosalind, comparing expression profiles across the experimental groups. The analysis applied the following cutoffs to identify significantly differentially expressed genes: a log$_2$ fold change (log$_2$FC) threshold of ≥ 0.6 for upregulated genes and ≤ -0.6 for downregulated genes, with a statistical significance threshold of $P < 0.05$. For differential gene expression analysis, we used the Wald test as implemented in the DESeq2 R package via the Rosalind® analysis platform. A two-sided test was performed to identify both upregulated and downregulated genes. P-values were adjusted for multiple comparisons using the Benjamini–Hochberg false discovery rate (FDR) correction method. Degrees of freedom are handled internally by DESeq2 and are not explicitly reported. Effect sizes are expressed as log$_2$ fold changes. For the volcano plot shown in Fig. 7d, confidence intervals were set at 0.05.

To determine the pathways were affected in the most significant way during optical stimulation of graphene-interfaced brain cortical organoids, we performed the Gene Ontology (GO) analysis (http://geneontology.org/docs/ontology-documentation/). The GO analysis compares the genes in a given group to sets of genes in a general category. It then gives a P-value for the associated pathways based on how many genes you detected that were in the set compared what would be expected for a random sample. FDR, or false discovery rate, is a method of adjusting p-values for large datasets. The enrichment score is the -log$_{10}$ of the FDR adjusted p-value. The results were sorted by their FDR adjusted p-values to get the top most significant pathways.

## Controlling a bionic robot using light-controlled graphene-interfaced brain organoids

Our platform consisted of a quadruped bionic robot dog (PuppyPi4, LewanSoul), a Maestro Pro™ MEA system (Axion Biosystems), and a computer for closed-loop Wi-Fi communication between the Maestro Pro™ and the robot. PuppyPi4 was powered by a Raspberry Pi 4B/5 system-on-chip with the robot Operating System (OS) installed. The robot was equipped with coreless servos that enable precise motor control and rapid rotation. A built-in TOF LiDAR laser system provided Simultaneous Localization and Mapping (SLAM), allowing real-time environmental mapping and dynamic path planning. The experimental setup consisted of an obstacle course with barriers and variable pathways, requiring the robot to navigate based on neural input from the brain organoids. Obstacle information detected by the robot was transmitted to the Axion Maestro Pro™ MEA system via a mobile Wi-Fi connectivity module, supporting bidirectional communications at speeds of up to 1.8 Gb/s.

The experiment was supported by a Linux-based software, which includes the neural MEA recording system, custom data processing software, and a robotic control platform. The software interface integrates a Python API with custom C++ and OpenCV libraries for efficient data processing. All three subsystems seamlessly interacted with the robot OS. A custom graphical user interface (GUI) facilitated real-time communication between the Maestro Pro™ MEA system and the robot, displaying the array well structure, spike raster plots, and a live video feed of the robot's motion. The GUI enabled near real-time analysis of neural activity, ensuring low-latency (millisecond range) communication between the Maestro Pro™ MEA system and the robot OS. The mapping strategy linked distinct spike pattern responses to specific robot motor actions (e.g., forward, right, or left). Sustained neural activity across multiple wells corresponded to forward movement, while significantly reduced neural activity results in a stationary state. For obstacle detection, the PuppyPi robot utilized the TOF LiDAR module and a wide-angle camera, continuously transmitting the spatial data to the Maestro Pro™ MEA system.

### Statistical analysis

Comparison of the data sets was performed using a two-tailed Student's $t$-test, or one-way ANOVA with a Bonferroni post-hoc test, when appropriate. Statistical analysis was performed in in OriginPro 2025b (OriginLab Corporation), Microsoft Excel 2019, and MATLAB 2019b (Mathworks). All experiments were performed using independent biological replicates. No technical replicates were used for statistical analysis. Data were tested for normality prior to statistical inference. Data are represented as means ± s.e.m. The significance between data sets after paired or two-sided $t$-test or ANOVA, with Bonferroni correction for multiple comparisons, is given as a $p$-value (*$p \leq 0.05$; **$p \leq 0.01$; ***$p \leq 0.001$; ****$p \leq 0.0001$).

### Reporting summary

Further information on research design is available in the Nature Portfolio Reporting Summary linked to this article.

## Data availability

All data supporting the findings of this study are available within the article and its supplementary files. Any additional reasonable requests for information can be directed to, and will be fulfilled by, the corresponding authors. Source data are provided with this paper. The transcriptional data generated in this study have been deposited in the NIH Gene Expression Omnibus (GEO) under accession code GSE301685. The data generated in this study are provided in the Supplementary Information/Source Data file. Source data are provided with this paper.

## Code availability

Custom codes and instructions for reproducing the key analyses of this study are available at GitHub (https://github.com/MarianaSac/Graphene; https://github.com/asavtchenko21/GraMOS; https://github.com/Omowuyi/Organoid-Robot-Control/ tree/main) under the MIT License.

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

## Acknowledgements

We would like to thank the Stem Cell Genomics Core at the Sanford Stem Cell Institute for providing access to and maintenance of the Dragonfly 600 microscope, preparing RNA samples for Illumina short-read sequencing, and for providing sequencing services. This publication includes data generated at the UC San Diego IGM Genomics Center utilizing an Illumina X Plus that was purchased with funding from a National Institutes of Health SIG grant (#S10 OD026929). We would like to thank Dr. Xiwei Shan for sharing brain cortical organoids for the initial stage of this project. This project provided a valuable platform for engaging numerous undergraduate and graduate students at UC San Diego in nanotechnology research, and while the individual contributions of some students did not meet the threshold for substantial contribution and authorship, we gratefully acknowledge their participation in various aspects of this project: Jay Chen, Hope T. Leng, Jason W. Adams, Malcom X. Lockett, Erin LaMontagne, Yuhui Li, Nicolas Amelinez,

Tianyao Xu, and Aayush Somani. We thank Meenakshi Patne for assistance with culturing brain cortical organoids. The authors acknowledge funding support from the National Institutes of Health (A.S.: 1R43MH124563; A.R.M: 1R01MH128365, R01NS123642, 1R01ES033636, MH123828, MH127077, NS105969; EM: 1R43NS122666, 1R43AG076088, 5R44DA050393), Department of Defense W81XWH2110306 (to A.R.M.), the California Institute of Regenerative Medicine (DISC2-13866 to A.S.), and the Long-term program of support of the Ukrainian research teams at the Polish Academy of Sciences carried out in collaboration with the U.S. National Academy of Sciences with the financial support of external partners to V.C.

## Author contributions

E.M. and A.S. conceived the project, designed the experiments, fabricated graphene biointerfaces, performed the experiments, analyzed the data, and wrote the manuscript. A.R.M. conceived the project, and provided experimental guidance and funding. T.Z., P.V., G.C., F.P., H.H., A. A.-Q., F.D., C.T.P., P.M., F.D., S.P., and J.S.S. performed experiments, and provided technical assistance and experimental guidance. V.C. performed experiments, wrote automated image analysis software, and analyzed the data. P.V and M.R. analyzed calcium imaging data. M.S.A.F. analyzed the MEA data. P.N. and D.K. performed graphene characterization and field effect transistor studies. O.O. worked on software for robot control using electrical signals from brain organoids. All authors participated in the manuscript revision.

## Competing interests

A.S. and E.M. are co-founders of Nanotools Bioscience, a company focused on exploring the optoelectronic properties of graphene for biomedical applications and developing the tools and technologies for enabling these applications. E.M. is the inventor and applicant on patents (US10137150B2, US10688127B2, CN106458601B, JP6635383B2, EP3157866B1) related to the graphene-based optical stimulation technology, and its applications, including its use for driving activity-dependent maturation of stem cell-derived cells. A.R.M is a co-founder and has an equity interest in TISMOO, a company dedicated to genetic analysis and brain organoid modeling focusing on therapeutic applications customized for autism spectrum disorder and other neurological disorders with genetic origins. The terms of this arrangement have been reviewed and approved by the University of California San Diego in accordance with its conflict-of-interest policies. E.M., A.S., and A.R.M. declare no other competing interests. The remaining authors declare that the research was conducted in the absence of any commercial or financial relationships that could be construed as a potential conflict of interest.

## Inclusion & Ethics Statement

This study does not involve experiment involving animals, human participants, or clinical samples. All research presented in this manuscript was conducted in accordance with ethical guidelines and best practices for scientific integrity and responsible collaboration. No part of the study involved research in resource-limited settings or populations from low- or middle-income countries. All contributors to this work were properly credited for their efforts and intellectual input.

## Additional information

[1]Nanotools Bioscience, La Jolla, CA 92037, USA. [2]NeurANO Bioscience, La Jolla, CA 92037, USA. [3]Shu Chien-Gene Lay Department of Bioengineering, School of Engineering, University of California San Diego, La Jolla, CA 92093, USA. [4]Institute of Bioorganic Chemistry, Polish Academy of Sciences, Poznan, Poland. [5]Bogomoletz Institute of Physiology, Kyiv, Ukraine. [6]Department of Biomedical Engineering, University of Massachusetts, Amherst, Amherst, MA 01003, USA. [7]Department of Pediatrics, School of Medicine, University of California San Diego, La Jolla, CA 92093, USA. [8]Neurogenetics Laboratory, Universidade Federal do ABC, São Bernardo do Campo, SP 09606-045, Brazil. [9]The Salk Institute for Biological Studies, La Jolla, CA 92027, USA. [10]Department of Cellular and Molecular Medicine, School of Medicine, University of California San Diego, La Jolla, CA 92093, USA. [11]Sanford Consortium for Regenerative Medicine, La Jolla, CA 92037, USA. [12]Present address: Axiom Space, Houston, TX 77058, USA. ✉e-mail: emolokanova@neuranobio.com; muotri@ucsd.edu; asavtchenko@nanotoolsbio.com

