## [Transparent Peer Review file · Nature Communications]

Non-Genetic Neuromodulation with Graphene Optoelectronic Actuators for Disease Models, Stem Cell Maturation, and Biohybrid Robotics

Corresponding Author: Dr Alex Savchenko

Version 0:

Reviewer comments:

Reviewer #1

(Remarks to the Author)

In this manuscript, Savchenko and coauthors presented a technological platform called GraMOS, which leverages the unique optoelectronic properties of graphene for optical neuromodulation. This platform addresses limitations associated with traditional optical neuromodulation techniques, particularly those involving genetic or structural modifications of neurons. GraMOS's use of graphene to convert light into electricity without altering the neurons' natural state, offering modulation of neural activity. The demonstration of GraMOS in both 2D neuronal cultures and 3D brain organoids highlights its potential applications.

The authors characterized the graphene materials used in GraMOS through techniques like UV-Vis spectroscopy, Raman spectroscopy, and electrophysiological studies. The biocompatibility of the graphene-coated surfaces was validated using human-induced pluripotent stem cells (hiPSC)-derived neurons. Additionally, the paper demonstrates GraMOS's applications by controlling robotic movement. Comments:

1. There is no characterization of the photo-electric properties of this device or platform. The authors did not report current or voltage generated by the graphene device without neurons present. There are only characterizations of material properties such as Raman or FTIR, but those are not related to the photo-electric properties of the device/platform presented in the manuscript. Please demonstrate characterizations of this graphene material's ability to convert light to electricity by directly measuring evoked current and voltage.
2. The relationship between the power of light and generated voltage/current should be characterized and reported.
3. What is the size of the laser beam used for modulation of graphene to generate electrons? Do all the neurons and axons within the boundary of beam get evoked or only the part that surpass certain energy bars can be evoked?
4. Looks like the graphene materials are deposited on a cover slip. Are they monolayer or multiple layers or graphene? Will it affect the results for monolayers vs multiple layers?
5. Are graphene deposited uniformly across the cover slip? How did the authors characterize the uniformity? Will it affect the required minimum laser power for different regions if graphene was not deposited uniformly across the platform.
6. Can the authors report photos of the device and platform? Throughout the manuscript there are mostly AI-generated schematics and microscopic images of cells. It will be helpful for readers to know what the device/platform looks like.
7. Figure 6 presents illustrations and controlling mechanism flowcharts, but the reviewer thinks that further data support within the figure is necessary to clarify how the robot's movements in Supplementary Video 3 are controlled.
8. Supplementary Video 3 demonstrated a moving robot but it did not demonstrate any controlling of the robot. If the authors really want to demonstrate applying this platform for robot controlling by taking advantage of the brain organoid, some more demonstration of controlling could be included. For example, can demonstrate that this platform can at least be used to control the robot speed, turning left, right, backward, and so on, by optically manipulating different neuron cells within the organoid.

Reviewer #2

(Remarks to the Author)

In this manuscript, the authors develop a graphene based substrate to optically modulate neural activities. The advantages

of graphene layers are their ultrathin structure, optical transparency and biocompatibility. My major concern is the novelty of this submission. Non-genetic, optical-based neural modulations have been extensively studied in various publications, using materials like silicon, graphene, metal nanoparticles, etc. The modulation effects are evaluated in the cellular and cultured tissue levels, as well as peripheral and central nervous systems in vivo. In addition, the authors reported graphene based optical modulation in a Science Advances paper in 2018 [Ref. 41], which has already covered most of the key concepts presented in this submission. The authors claim some promising applications, like "... supporting the studies of neural connectivity, brain development, disease modeling, personalized medicine, drug testing and development, neurotoxicity testing, and brain-computer interface research ..." (Line 268). However, they lack solid demonstrations for these claims, and current results in Fig. 4, 5 and 6 are mundane and not significant. Therefore, the current manuscript is not critical to be considered as a significant contribution to the field.

To improve the manuscript, I have the following suggestions:

1. Demonstrate some new effects based on results in Figs. 4 and 5. The currently presented photo modulation effects on the organoids are trivial and not very significant. Additional drug testing experiments can be very beneficial.
2. Demonstrations in Fig. 6 and Video 3 are too simple as well. More complicated experiments should be presented. There are so many variables and degrees of freedoms in this system, including an organoid with thousands of neurons, spatially resolved and tunable light spots, and multichannel electrodes. Unfortunately, the current demonstration is simply a forward movement of the robot. This makes people doubt about the capability and utility for such a system.

Other minor comments:

3. Words like "excellent", "pioneering", "disrupt the conventional paradigm", ... are used a lot in the paper. These expressions are objective and not appropriate.
4. Optical transmission or absorption spectra should be measured for the graphene layer.
5. The paper claims a so-called "hot electron effect". If there are electrons, how about holes? Where do holes go? A scheme of band diagram is suggested to illustrate the carrier generation and transport process in the graphene, and at the graphene/water interface.
6. Fig. 2c, this prolong effect is due to photocapactive or photoelectrochemical effect?
7. Fig. 5e discusses two different scenarios, "inside" and "outside". Can you provide two figures to illustrate these two configurations?

Reviewer #3

(Remarks to the Author)

This work is primarily aiming at demonstrating that GraMOS can achieve photo-to-electric activation of neurons without genetic modification of them. The concept of using non-genetic regulation methods to achieve neural photostimulation through optoelectronic actuators is an interesting but not new direction. Data supporting the advantages of GraMOS as a non-genetic regulation method compared to other photo-electrical converters are limited, such as the lower light intensity requirements and reduced heating due to conversion efficiency mentioned in the discussion section of the article. The long-term stability and repeatability of light stimulation mentioned in the introduction section of the article are not supported by data. Therefore, the innovativeness of the article in comparison to other methods in the same field is not well demonstrated. Overall, I believe this manuscript is not suitable for publication in Nature Communications.

Major Comments:

1. From Figure 2 to Figure 6, the core content is essentially about demonstrating that graphene can achieve photo-to-electric activation of neurons, where the main difference is roughly only given by different signal recording methods and different neural tissues. The overall information presented in these figures are limited which might not be able to give a whole characterization of the system. It might be considered to compact this part of content and instead increase the multidimensional characterization of GraMOS's such as light intensity, temperature, long term usage, or light spectrum to enrich the article.
2. The usage of figures is too arbitrary; for example, none of the figures clearly illustrate a schematic diagram of the seamless integration of GraMOS flakes with organoids, together with all minor issues mentioned above. Particularly, Figure 6 only provides a rough experimental scheme and method descriptions in the supplementary materials, instead of experimental results.
3. The majority of the schematic diagrams in the article (i.e., Fig. 2a, 3a, 3f, 4e, 5a, 6a) are generated directly using AI software, and the authors may need to check the accuracy of the details to ensure they do not mislead the readers.
4. For information in all figures alone, it might be difficult for readers to immediately understand that there is another interaction strategy of free-standing graphene interface with organoid, in addition to the substrate-based graphene interfaces. Is it necessary to emphasize these two interaction strategies with a schematic diagram?
5. Fig 6 as a whole seems to have limited significance in the experiment, and the entire figure mainly describes the signal processing method without providing specific actual result data.

Minor Comments:

1. Would it be more intuitive to include one or two actual photos of the fabricated graphene in Fig 1? The absence of a direct fabrication result image of graphene throughout the article may confuse readers who are not in the field.
2. In Fig. 1a, the schematic diagram of the morphology of neurons and the lattice situation of graphene seems a bit too unreal which might mislead readers. Could it be refined?
3. It is suggested to increase the number of samples in Fig. 1f and optionally provide specific P-values, especially for the resting potential subfigure, as it does not visually appear to have no significant difference.
4. Why is the action potential waveform in Fig. 2c so long? It has reached the scale of tens of milliseconds.
5. Is there actual experimental data to support the non-interference of light bands described in Fig. 3c? This point is an important support for the valid neural signals in the subsequent Figs 3d-3h.
6. What are the lighting parameters in Fig 3e? There is a discrepancy between the figure caption (638 nm; 3.9 mW/mm², 5 ms) and the main text (line 248s (2 ms, 638 nm, 3.6 mW/mm²)).
7. What do the different colors in Fig 3d, 3e, 4g represent? Do the same colors indicate the same neurons?
8. Is there actual experimental data to support the non-interference of light bands described in Fig. 3c? This point is an important support for the valid neural signals in the subsequent Figs 3d-3h.
9. Is there direct data to support the mention of 10% neurons and 70% neurons in the text from line 247 to line 250?
10. Since Fig 3e has changed to a 2.5D experimental paradigm compared to Fig 3d, should this key change be reflected in the figure caption?
11. In Figs 3f-3h and Supplementary Video 2, it can be seen that the neural impulse propagation within hundreds of micrometers takes nearly ten seconds, while the description in Fig 5c and the text (lines 311-313) states that the neural impulse travels through the entire organoid in only 100 ms. Is this a contradiction, or what is the reason for this?
12. Is there literature support for the claim that the excitation-inhibition ratio represents the maturity of the organoid in the text? If so, could it be provided in main text?
13. Is Fig 5a also an AI-generated graph? If so, would it better be indicated in the figure caption?
14. The 4x4 MEA array distribution shown in Fig 5b contradicts the 8x8 MEA array distribution described in Fig 5d and its figure caption.
15. Would it be more appropriate to change the y-axis label in Fig 5e from "spike increase" to "spike number" to better match the actual meaning?
16. In the data analysis process described in Fig 6b, why is the electrophysiological signal collected from the MEA first converted into an image and then through OCR to a voltage value matrix? Is not the data directly obtained from the MEA already a voltage value matrix?

Reviewer #4

(Remarks to the Author)

This article reports the application of graphene materials for optical neuromodulation through the development of a platform called GraMOS (Graphene-Mediated Optical Stimulation).

It has been previously reported that graphene can be optically excited to trigger single action potentials or induce bursting activity via light pulses of varying durations. The main progress of this work is about the use of three-dimensional brain organoids and the light-triggered electrical signals generated by these organoids, which are collected by a multi-electrode array (MEA) and used to drive a robot to perform pre-set actions.

My Concerns include:

1. There are quite a few missing details, and in several instances, the text and corresponding figures do not align well. The current structure of the manuscript makes it difficult to follow, necessitating significant effort to reorganize and clarify.
2. Through capacitive coupling, GraMOS alters the membrane potential of neurons, triggering action potentials in a femtosecond time scale, providing precise and controllable regulation of neuronal activity. The recent development of nanoelectrodes for non-Faradaic capacitive, rapid, and reversible optical stimulation of neurons has been cited in reference #63. Although this work was only published a few months ago, as a parallel development, a discussion comparing this report with reference #63 is needed.
3. The key advance of combining a graphene interface with an organoid to control the movement of a Freenove robot via custom Python software is interesting. However, the engineering details are lacking, making it difficult for others to replicate the experiments. More information on this aspect are essentially needed.
4. The manuscript mentions that GraMOS is located extracellularly and serves as an optical actuator to stimulate neurons in their natural state without modifications. How is this extracellular positioning characterized? Could it be located on the cell

membrane? Evidence needed here. Additionally, the "graphene outside" and "inside" configurations seem to exhibit similar increases in electrical activity in response to light. This section is presented superficially and needs further elaboration.

5. The authors used a thermocouple probe to measure the graphene surface temperature after illumination at a wavelength of 452 nm, with a light intensity of 3.9 mW/mm² for 30 minutes at a 5 Hz pulse frequency, and no temperature changes were detected. However, the article does not present the corresponding setup and data, which should be included.

Other minor issues include:

The characterization techniques used for rGO do not mention XRD in line 135 (page 4), but it is mentioned as a technique used on page 5. Additionally, SEM images of GO and rGO should be included to observe the morphology and structure for comparison, which is common in the literature (line 164)

In Figure S1, the disappearance of the shoulder peak at 300 nm in rGO is mentioned but not discussed in the text when referring to the UV-Vis of rGO. The Raman spectra of rGO are shown in blue, but it would be helpful to add the spectra of GO in red for comparison. This would support the text (line 148), which compares the Raman spectra of GO and rGO. The XRD x-axis should start at 0° to show the full range, as GO has a peak at 10° in the literature. Including the XRD of GO and comparing it to rGO supports the formation of rGO, as mentioned in line 156.

In Figure 1c and 1d, for the cell viability assay, three replicates and five fields of view were used, with $n \geq 100$ cells per condition. This allows for selective viewing of more viable cells. It would be better to perform MTS or MTT cell viability assays to avoid exclusion bias.

Reference #43 used WST, which is similar to MTS. In Figure 1f, one variable has $n=9$, while the other has $n=12$. When comparing these, the repeat numbers should be consistent. Additionally, is the $n=9$ and $n=12$ representing mean \pm s.e.m. from independent studies or from the same study?

In Figure 2c, where it says "1-s duration for the left trace," do you mean the right trace, as the left trace is already mentioned? Figure 3b should include a control image of neuron cells grown on glass coverslips with Fluo-4 (no graphene).

Lines 374-379 in the discussion need to address how GraMOS provides non-Faradaic capacitive effects and why electrochemical effects cannot be ruled out. It also needs to explain how the lack of pH change after continuous 30-minute exposure plays a role. These points are currently listed but not discussed.

Additionally, the paper is missing a conclusion, which would help summarize the key findings and the significance of the research.

Reviewer #5

(Remarks to the Author)

Version 1:

Reviewer comments:

Reviewer #1

(Remarks to the Author)

My comments from the previous round have been addressed by the authors. I have no additional comments.

Reviewer #2

(Remarks to the Author)

The authors have revised the manuscript according to all the Reviewers' comments, and new experimental results have been provided. Although the manuscript quality is greatly improved, I still have the following suggestions:

1. Line 38, "... particularly, its ability to efficiently convert light into electricity on a femtosecond timescale ...". For biological modulation, I do not see any advantage of a "femtosecond" response. Microsecond or even millisecond responses are sufficient for stimulating cells.
2. Fig. 1e shows the photos of different samples. Please provide the absorption spectra for these samples (e.g., from 400 to 800 nm). Which sample is used for cell culture and subsequent cell and organoid experiments?
3. Figs. 2a and 2d show the photocurrent of graphene. Please add the results of current versus time, under illumination. This will reveal the portion of photocapacitive and photofaradic currents. Please refer to: Y. Jiang, Nat. Biomed. Eng. 2, 508–521 (2018).
4. Fig. 2e and Line 847, this thermal characterization is inaccurate. First, the local temperature at the cell/graphene interface on the laser spot is much higher than the solution in general; second, the laser spot can create artefacts on the thermocouple. A more accurate thermal measurement can be performed using the patch system. Please refer to: Y. Jiang, Nat. Biomed. Eng. 2, 508–521 (2018).
5. Line 201, "... did not result in significant changes in the pH ...". No measurement results are shown here. In addition, more experiments should be performed to evaluate the reactive oxygen species (ROS) level in the solution.
6. Fig. 2l and 2m, the authors should carefully check to raw data for these fluorescence signals. Those narrow spikes should not be calcium spikes, and most likely are the artefacts created by the laser spot. They are dramatically different from normal calcium signals like those in Fig. 3c and 4k. To preclude the possibility of artefacts, control experiments should be

performed: (1) on cells grown on pure glass; (2) on graphene samples without any cells.

7. Fig. 3h, unit is missing.

8. Fig. 3j-3m, there is no control group here (without light / without graphene).

9. Line 325, it is not "Fig. 3n", but "Fig. 3m".

10. Fig. 4, captions are messed up, no 4g-4k.

11. Fig. 6 in the original draft is completely deleted. Some images may be valuable and should be included in the main figure or supporting figures.

12. Non-academic words like "fortunately", "well-being", ... are used a lot in the paper.

Reviewer #3

(Remarks to the Author)

After carefully reviewing the authors' rebuttal letter, I would like to stand by my previous evaluation.

The key concern regarding the acceptance of this manuscript lies in its level of novelty. The authors claim that GraMOS represents the first graphene-based optoelectrical stimulation approach to activate neurons. However, in this field, similar work has already been demonstrated: (1) non-graphene materials have been used for photoelectrical neural stimulation (e.g., from Prof. Bozhi Tian at UChicago and Prof. Xing Sheng at Tsinghua), and (2) graphene has been employed for photothermal stimulation (e.g., from Prof. Tzahi Cohen-Karni at CMU). Although the authors correctly point out that temperature rise may be a concern in photothermal methods, the rationale for choosing graphene in their system is not clearly articulated.

Given the extensive prior research on non-genetic, light-induced neural stimulation—see, for example, the Science essay in 2019 (DOI: 10.1126/science.aay4351)—I would characterize this manuscript as an "A+B" or "me-too" type of study. As such, I believe it falls below the threshold of novelty expected for Nature Communications.

Reviewer #4

(Remarks to the Author)

The authors have significantly enhanced the revised manuscript by incorporating extensive experimental data and expanded discussions, which effectively address the majority of prior concerns. However, critical limitations persist in demonstrating how light-evoked electrophysiological signals from graphene-interfaced cerebral organoids translate into robotic locomotion control. The current reliance on a qualitative demonstration video, devoid of quantitative metrics or methodological rigor, remains insufficient for robust scientific validation.

Major Comments:

1. In the revised manuscript introduction, the authors mention developing graphene-based optical actuators but do not highlight how the newly added data demonstrate the advantages of this development. The authors should highlight the key innovations and novelty of this work in the paragraph (line 86-91).
2. While the revised manuscript includes new data on G-coverslips with optical transmittance, the authors should have described the total graphene concentration without establishing the relationship between concentration/mass and optical transmittance.
3. The temperature measurements shown in Figure 2e lack critical details. The manuscript does not specify whether the detected temperature corresponds to the illuminated area or the entire graphene substrate. The full details of the temperature sensing setup and measurement location are needed.
4. The manuscript states a ~20% success rate for activating individual neurons via optical stimulation in G-interfaced organoids. The methodology for calculating this value requires clarification. Additionally, details about light penetration depth through the entire organoid should be included and discussed.
5. A discrepancy exists in the wavelength selection: UV light was used for assessing rGO flake photosensitive, while 452 nm light was employed for G-interfaced neuron stimulation. The rationale for this inconsistency needs explanation, particularly considering the known cellular damage caused by UV radiation.
6. The revised manuscript mentions 2 Hz optical training of neurons and organoids, but does not quantify the illuminated area or the light-power density. These parameters must be provided.
7. The observation in Figure 4g showing a decreased VGLUT1/GAD65/67 ratio (from ~20% in controls to ~8% in graphene groups) suggests a shift toward inhibitory balance in excitation/inhibition equilibrium. The authors should discuss the potential mechanisms underlying this phenomenon.

Other minor issues include:

1. Figure citation errors require correction: The schematic of the light illumination module stimulating neurons is incorrectly labeled as Figure 3k (should be Figure 3j). References to non-existent Figure 3n should be removed. The Figure 3 description appears disorganized and needs thorough verification.
2. Identical sentences appear at lines 536 and 542. Please remove duplicate text.
3. Some word choices (e.g., "multi-faceted") may be imprecise.
4. In Fig. S2b–c and Fig. S4a, the rGO thickness is reported as 2.9 ± 0.3 nm. Does this correspond to multilayer flakes?
5. The manuscript states that graphene's electronic band structure enables broadband absorption from ultraviolet to infrared wavelengths. Please specify the exact wavelength range tested or expected.

Version 2:

Reviewer comments:

Reviewer #2

(Remarks to the Author)

The authors have revised the manuscript according to all the Reviewers' comments, and new experimental results have been provided.

In my previous review report, there are some comments that have not been addressed:

My previous comment #3: "Figs. 2a and 2d show the photocurrent of graphene. Please add the results of current versus time, under illumination. This will reveal the portion of photocapacitive and photofaradic currents. Please refer to: Y. Jiang, Nat. Biomed. Eng. 2, 508–521 (2018)."

The authors provide a long and obscure response to this comment. I do not understand why they could not simply provide the experimental results of the photocurrent versus time. It is super easy to measure using a patch-clamp system. I insist that these results should be added.

My previous comment #6: "Fig. 2l and 2m, the authors should carefully check to raw data for these fluorescence signals. Those narrow spikes should not be calcium spikes, and most likely are the artefacts created by the laser spot. They are dramatically different from normal calcium signals like those in Fig. 3c and 4k. To preclude the possibility of artefacts, control experiments should be performed: (1) on cells grown on pure glass; (2) on graphene samples without any cells."

I still have doubts about these data (Fig. 2o and 2p in the revised paper). Please provide the raw data (excel sheet) for these plots.

Reviewer #4

(Remarks to the Author)

The authors have provided a detailed response to the question raised and have offered a clear explanation of the underlying mechanism(s). my major comment is still around 'biohybrid robotics"! at this stage of demonstration, the role of light-evoked electrophysiological signals from graphene-interfaced cerebral organoids in robotic locomotion control remains very basic, so the claim of biohybrid robotics is over claime, suggest to detone here. Some additional minor corrections need further clarification.

1. The "n-doped" in Fig. 1a is not an accurate way to describe the transport mechanism. No dopant is applied in this work. I assume that the author would like to say it is "n-type", can the authors explain the "n-doped" instead of "n-type" in details?
2. Line 683, the author describes the fabrication of the rGO-FET device in Figure 5b. But Figure 5b does not provide the information. And please illustrate the IDE material in this device.
3. The authors have updated Fig. 4 to demonstrate the excitation/inhibition equilibrium of the neuronal network. Indeed, the complex of the neuronal network will affect the excitations and inhibition, but the authors' explanation is still ambiguous, please find more specific explanation in the reference (<https://doi.org/10.1038/s41467-020-17521-w>).
4. In Fig. 3b, the scale bar is μm rather than μM .
5. In Fig. 4e and 4g, it is better to align the scale bars well. In the caption, h is wrongly labeled as j.
6. In Supplementary Figure 1d, the peak around 3000 cm^{-1} is $\text{D}+\text{D}'$ instead of $\text{D}+\text{D}$.

Version 3:

Reviewer comments:

Reviewer #2

(Remarks to the Author)

The authors have provided experimental data to respond to my comments. I think the paper can be accepted, although I still reserve my judgement on their explanations about the photocurrent mechanisms provided in the inset of Fig. 2b.

(Remarks on code availability)

Reviewers' Comments and the Authors' Responses

Reviewer #1 (Remarks to the Author): In this manuscript, Savchenko and coauthors presented a technological platform called GraMOS, which leverages the unique optoelectronic properties of graphene for optical neuromodulation. This platform addresses limitations associated with traditional optical neuromodulation techniques, particularly those involving genetic or structural modifications of neurons. GraMOS's use of graphene to convert light into electricity without altering the neurons' natural state, offering modulation of neural activity. The demonstration of GraMOS in both 2D neuronal cultures and 3D brain organoids highlights its potential applications. The authors characterized the graphene materials used in GraMOS through techniques like UV-Vis spectroscopy, Raman spectroscopy, and electrophysiological studies. The biocompatibility of the graphene-coated surfaces was validated using human-induced pluripotent stem cells (hiPSC)-derived neurons. Additionally, the paper demonstrates GraMOS's applications by controlling robotic movement.

Comments:

1. There is no characterization of the photo-electric properties of this device or platform. The authors did not report current or voltage generated by the graphene device without neurons present. There are only characterizations of material properties such as Raman or FTIR, but those are not related to the photo-electric properties of the device/platform presented in the manuscript. Please demonstrate characterizations of this graphene material's ability to convert light to electricity by directly measuring evoked current and voltage.

We appreciate the reviewer's suggestions. We conducted extensive additional characterization of our materials, including their photoresponsiveness (please, see Fig. 1b-e, Fig. 2 a-e, Fig. S4, Fig. S5, and the text in the main text on p. 6, and the Supplementary Information).

2. The relationship between the power of light and generated voltage/current should be characterized and reported.

Thank you for the reviewer's recommendations. We conducted additional experiments as requested by the reviewer. The results of these experiments are now presented in Fig. 2a, 2b, and 2g.

3. What is the size of the laser beam used for modulation of graphene to generate electrons? Do all the neurons and axons within the boundary of beam get evoked or only the part that surpass certain energy bars can be evoked?

In our experiments, the laser spot was approximately 2 μm in diameter, approximately matching a single-neuron activation spot. The light intensity of a laser spot must indeed exceed a certain threshold to activate a neuron.

4. Looks like the graphene materials are deposited on a cover slip. Are they monolayer or multiple layers or graphene? Will it affect the results for monolayers vs multiple layers?

In our experiments, graphene materials deposited on the surface are multilayer structures. We now show that the relation between the coating thickness and photocurrent amplitudes in new Fig. 2a.

5. Are graphene deposited uniformly across the cover slip? How did the authors characterize the uniformity? Will it affect the required minimum laser power for different regions if graphene was not deposited uniformly across the platform.

We characterized the coating uniformity using optical microscopy by quantifying the optical density of the coated substrates and confirmed that, in the case of spray coating deposition, the uniformity was $\pm 5\%$. If less than desired uniformity is encountered, this issue can be addressed by optimizing the experimental setup. Since neurons respond to activating signals in an all-or-none manner, a slightly higher light intensity can be chosen to activate any neuron on a given substrate while still remaining safe.

6. Can the authors report photos of the device and platform? Throughout the manuscript there are mostly AI-generated schematics and microscopic images of cells. It will be helpful for readers to know what the device/platform looks like.

We appreciate the reviewer's suggestions. We now included images of our graphene materials and optical stimulation platform components in Fig. 1b, 1e, Fig. 3j, Fig. 5m, Fig. S2a, and Fig. S4a.

7. Figure 6 presents illustrations and controlling mechanism flowcharts, but the reviewer thinks that further data support within the figure is necessary to clarify how the robot's movements in Supplementary Video 3 are controlled.

We are thankful for the reviewer's valuable feedback. We removed the flowchart, and took a completely different technological route in our experiments involving the organoid-robot system. This new approach demonstrates close-loop dynamic interaction between the organoid and the robot, enhancing the system's ability to adapt and respond to environmental challenges (p. 13-14, Video S4).

8. Supplementary Video 3 demonstrated a moving robot but it did not demonstrate any controlling of the robot. If the authors really want to demonstrate applying this platform for robot controlling by taking advantage of the brain organoid, some more demonstration of controlling could be included. For example, can demonstrate that this platform can at least be used to control the robot speed, turning left, right, backward, and so on, by optically manipulating different neuron cells within the organoid.

We appreciate the reviewer's feedback. In response, the section describing the organoid-robot system, along with Supplementary Video 4, has been thoroughly revised and updated to reflect a more advanced technological approach.

Reviewer #2 (Remarks to the Author): In this manuscript, the authors develop a graphene-based substrate to optically modulate neural activities. The advantages of graphene layers are their ultrathin structure, optical transparency and biocompatibility. My major concern is the novelty of this submission. Non-genetic, optical-based neural modulations have been extensively studied in various publications, using materials like silicon, graphene, metal nanoparticles, etc. The modulation effects are evaluated in the cellular and cultured tissue levels, as well as peripheral and central nervous systems in vivo. In addition, the authors reported graphene based optical modulation in a *Science Advances* paper in 2018 [Ref. 41], which has already covered most of the key concepts presented in this submission. The authors claim some promising applications, like "... supporting the studies of neural connectivity, brain development, disease modeling, personalized medicine, drug testing and development, neurotoxicity testing, and brain-computer interface research ..." (Line 268). However, they lack solid demonstrations for these claims, and current results in Fig. 4, 5 and 6 are mundane and not significant. Therefore, the current manuscript is not critical to be considered as a significant contribution to the field.

A. We appreciate the reviewer's perspective regarding the current landscape of non-genetic light-controlled neural modulators. We would like to take this opportunity to clarify our position in light of these comments. Several non-genetic, optical-based neural modulators cited by the reviewer employ photothermal processes to activate neurons, including silicon-based neuromodulators (Fig. 10a in PMID 30980031), gold nanoparticles (Fig. 2F in PMID 25772189), and fuzzy graphene (Fig. 2b in PMID: 32482882). Considering that neuronal physiology is extremely temperature-sensitive, raising the temperature by several degrees and affecting numerous side processes in neurons just to trigger an action potential is a suboptimal way to operate. For example, a temperature increase of 0.2-2°C on its own leads to changes in neuronal spiking in multiple brain regions (PMID: 31209378; PMID: 36126646). Photoelectrochemical processes that are present in some non-genetic neuromodulators (PMID: 29459654) are also suboptimal because redox reactions accompanying these processes can lead to oxidative stress, which damages cellular components and disrupts normal neuronal function. These limitations apply to all the platforms mentioned by the reviewer.

We briefly reviewed the platforms mentioned by the reviewer in Introduction and Discussion (p. 5, p. 16) and explained how our platform is fundamentally different from the platforms cited by the reviewer.

B. This is the first time ever that we submitted a manuscript about neuromodulation using a graphene optoelectronic platform. Although neurons are also excitable cells, they differ significantly from cardiomyocytes (the focus of our earlier publication in *Science Advances*, 2018) in terms of their structure, excitability, biophysical properties, modes of intercellular communication, and the relative importance of individual versus population-level responses in the brain compared to the heart. As a result, neuromodulation platforms must differ greatly from cardiac optical pacing platforms.

To Improve the manuscript, I have the **following suggestions**:

1. Demonstrate some new effects based on results in Figs. 4 and 5. The currently presented photo modulation effects on the organoids are trivial and not very significant.

To our knowledge, there are no published studies describing any non-genetic optical stimulation platforms for long-term interfacing and neuron activation in 3D brain organoids. Generating and culturing brain cortical organoids is not considered to be trivial even on its own. Our new protocol that allows to intersperse graphene materials into developing organoids without affecting their viability and organization, while remaining functionally active and acquiring light sensitivity, was a tall order.

To strengthen our work in response to the reviewer's suggestion, we conducted a series of additional experiments demonstrating the long-term use of GraMOS. These include studies on functional phenotyping and neural connectivity in hiPSC-derived models of Alzheimer's disease, as well as GraMOS-driven development and maturation of hiPSC-derived neurons and cortical brain organoids.

2. Demonstrations in Fig. 6 and Video 3 are too simple as well. More complicated experiments should be presented. There are so many variables and degrees of freedoms in this system, including an organoid with thousands of neurons, spatially resolved and tunable light spots, and multichannel electrodes. Unfortunately, the current demonstration is simply a forward movement of the robot. This makes people doubt about the capability and utility for such a system.

We appreciate the reviewer's suggestions. We have redesigned our organoid-robot system both technically and conceptually, creating a dynamic closed-loop interaction between the organoid and the robot that enhances the system's ability to adapt and respond to environmental challenges. This new system is described on pp. 13–14 of the main text and on p. 28 of the Methods section. It is also demonstrated in the new Supplementary Video 4.

Other minor comments:

3. Words like “excellent”, “pioneering”, “disrupt the conventional paradigm”, ... are used a lot in the paper. These expressions are objective and not appropriate.

We are grateful for this constructive criticism, and revised the text to maintain a more neutral tone.

4. Optical transmission or absorption spectra should be measured for the graphene layer.

We appreciate the reviewer's suggestions. These results can be found in Fig. 1e, 2a, 2c, and S1b.

5. The paper claims a so-called “hot electron effect”. If there are electrons, how about holes? Where do holes go? A scheme of band diagram is suggested to illustrate the carrier generation and transport process in the graphene, and at the graphene/water interface.

We appreciate the reviewer's suggestions. We added Dirac cones to illustrate the transitions of photogenerated charge carriers in graphene (Fig. 1a), and provided an explanation regarding the localization of electrons and holes, as well as the breakdown in symmetry between their behaviors on multilayer flake-based graphene surfaces in the presence of external electric fields (p. 4). In general, an asymmetric response to light in graphene comes from the fact that photogenerated holes tend to become trapped in defects within the graphene lattice, while electrons, which occupy delocalized π orbitals extending above and below the lattice plane, are less influenced by these localized imperfections.

6. Fig. 2c, this prolong effect is due to optocapacitive or photoelectrochemical effect?

We thank the reviewer for this insightful question. The duration of light illumination does not change the mechanism of GraMOS. All GraMOS effects are optocapacitive because (a) the lifetime of photogenerated charge carriers is on the femtosecond time scale, and (b) no changes in pH were observed during 30 minutes of continuous light exposure at graphene interfaces immersed in an electrolytic solution.

7. Fig. 5e discusses two different scenarios, “inside” and “outside”. Can you provide two figures to illustrate these two configurations?

To strengthen our work in response to the reviewer's suggestion, we added an explanatory diagram (Fig. 5b) for two interfacing configurations. We also conducted additional experiments and expanded the analysis of the results in internal vs. external interfacing configurations in brain organoids (Fig. 5d-l).

Reviewer #3 (Remarks to the Author):

This work is primarily aiming at demonstrating that GraMOS can achieve photo-to-electric activation of neurons without genetic modification of them. The concept of using non-genetic regulation methods to achieve neural photostimulation through optoelectronic actuators is an interesting but not new direction. Data supporting the advantages of GraMOS as a non-genetic regulation method compared to other photo-electrical converters are limited, such as the lower light intensity requirements and reduced heating

due to conversion efficiency mentioned in the discussion section of the article. The long-term stability and repeatability of light stimulation mentioned in the introduction section of the article are not supported by data. Therefore, the innovativeness of the article in comparison to other methods in the same field is not well demonstrated. Overall, I believe this manuscript is not suitable for publication in Nature Communications.

We respectfully thank the reviewer for their feedback and the opportunity to clarify the significance and novelty of our work. However, we would like to express our concern that the assessment may underappreciate the substantial advances presented in our study.

While the broader concept of non-genetic optical neuromodulation has indeed been explored in the field, the existence of other approaches does not diminish the innovation of GraMOS. Rather, the diversity of methods reflects the richness of ongoing scientific progress, not its redundancy. Each approach must be evaluated on its specific capabilities, mechanisms, and advantages.

The central goal of any optical neuromodulation platform is to reliably depolarize neurons without compromising their function or viability. The majority of non-genetic optical neuromodulators are based on photothermal and photoelectrochemical effects. High light intensity and heating are especially problematic for this task. At minimum, they can: (a) alter ion channel gating kinetics and firing thresholds; (b) affect synaptic transmission and calcium buffering; (c) disrupt network synchrony and excitability. At worst, they can (a) compromise cellular health by inducing thermal stress, structural damage, or cell death; (b) trigger inflammatory pathways, altering experimental conditions and confounding results; and (c) generate reactive oxygen species, causing oxidative stress and damage to essential cellular components such as DNA, proteins, and membranes.

To suggest that GraMOS offers only limited advantages—despite its ability to effectively activate neurons while avoiding the significant issues outlined above—unduly minimizes its contribution and impact.

Regarding the reviewer's comment on long-term stability and repeatability: while our original manuscript included demonstrations of these properties, we have now performed additional experiments that further support the long-term use of GraMOS. Specifically, we show that repeated photoactivation over several weeks promotes the maturation of hiPSC-derived neurons and brain organoids. To the best of our knowledge, no other non-genetic optical neuromodulation platforms have demonstrated effective use for durations exceeding a few hours, making this a distinguishing feature of our approach.

Major Comments:

1. From Figure 2 to Figure 6, the core content is essentially about demonstrating that graphene can achieve photo-to-electric activation of neurons, where the main difference is roughly only given by different signal recording methods and different neural tissues. The overall information presented in these figures are limited which might not be able to give a whole characterization of the system. It might be considered to compact this part of content and instead increase the multidimensional characterization of GraMOS's such as light intensity, temperature, long term usage, or light spectrum to enrich the article.

The manuscript is the first ever demonstration that graphene-mediated optical stimulation can efficiently activate neurons, and can do it in different types of neurons (primary, hiPSC-derived), in different spatial configurations (2D and 3D), and in two configurations (internal and external) in 3D. Our goal was to use orthogonal methods for this demonstration to convince the reviewer that this phenomenon is real. We are glad to see that, per the reviewer's comments, that goal was achieved.

To enrich the article, we conducted more thorough characterization of our materials (Fig. 1c-d, Fig. 2 a-e, Fig. S1, Fig. S2, Fig. S4, Suppl. Fig. S5), including the effects of such parameters as light intensity, temperature, long term usage, or light spectrum.

2. The usage of figures is too arbitrary; for example, none of the figures clearly illustrate a schematic diagram of the seamless integration of GraMOS flakes with organoids, together with all minor issues mentioned above. Particularly, Figure 6 only provides a rough experimental scheme and method descriptions in the supplementary materials, instead of experimental results.

In response to the reviewers' comments, and due to a substantial increase in experiments (approximately doubling their number), all figures have been expanded with a strong focus on clarity.

3. The majority of the schematic diagrams in the article (i.e., Fig. 2a, 3a, 3f, 4e, 5a, 6a) are generated directly using AI software, and the authors may need to check the accuracy of the details to ensure they do not mislead the readers.

Schematic diagrams are intended as simple visual aids to help readers understand the differences between experimental configurations. These diagrams serve this purpose effectively without misleading the audience. Although there was no intention to mislead in the first place, we have modified some diagrams and expanded certain figure legends to address potential sources of confusion.

4. For information in all figures alone, it might be difficult for readers to immediately understand that there is another interaction strategy of free-standing graphene interface with organoid, in addition to the substrate-based graphene interfaces.

We appreciate the reviewer's suggestion. We added an explanatory diagram (Fig. 5b) for two interfacing configurations. We also conducted more experiments and expanded the analysis of the results in internal versus external configurations (Fig. 5d-l).

5. Fig 6 as a whole seems to have limited significance in the experiment, and the entire figure mainly describes the signal processing method without providing specific actual result data.

In response to the reviewer's suggestion, Fig. 6 is removed from the manuscript.

Minor Comments:

1. Would it be more intuitive to include one or two actual photos of the fabricated graphene in Fig 1? The absence of a direct fabrication result image of graphene throughout the article may confuse readers who are not in the field.

In response to the reviewer's comments, we added panels Fig. 1b and 1e to the manuscript.

2. In Fig. 1a, the schematic diagram of the morphology of neurons and the lattice situation of graphene seems a bit too unreal which might mislead readers. Could it be refined?

We have carefully considered the reviewer's recommendations. The area of the area of one hexagonal unit cell in the graphene lattice is $\sim 5.24 \text{ \AA}^2$ ($\sim 0.0524 \text{ nm}^2$), while a neuronal soma is $\sim 10 \text{ }\mu\text{m}$ (10,000 nm). Due to a $\sim 40,000\times$ size difference between a neuronal cell body and a single graphene hexagon, it is not possible to scale in one image. This information is now added to the legends of Figure 1.

3. It is suggested to increase the number of samples in Fig. 1f and optionally provide specific P-values, especially for the resting potential subfigure, as it does not visually appear to have no significant difference.

We appreciate the reviewer's comment and the opportunity to clarify our results. As standard practice, statistical analysis is usually conducted using appropriate statistical tests rather via visual inspection of the bar graphs. The results described in the manuscript were obtained using statistical tests, and they show that the parameters in the 2024 graphs were not statistically significant. The corresponding p-values were as follows: 0.12194 for cell membrane potential, 0.92947 for the action potential (AP) amplitude, and 0.39887 for the amplitude of after-hyperpolarization (AHP).

Nevertheless, as requested, we increased the sample size. The outcome of statistical tests remained the same. The electrical activity parameters in neurons cultured on glass vs. G-substrates are still not statistically different. The updated graphs yielded the following p-values: 0.18961 for cell membrane potential, 0.86958 for the AP amplitude, and 0.46391 for the AHP amplitude.

4. Why is the action potential waveform in Fig. 2c so long? It has reached the scale of tens of milliseconds.

There are no action potentials in Fig. 2c that last tens of milliseconds. If this question is about the right panel of Fig. 2c, then we would like to point out that it shows a train of action potentials during sustained depolarization of the cell membrane.

5. Is there actual experimental data to support the non-interference of light bands described in Fig. 3c? This point is an important support for the valid neural signals in the subsequent Figs 3d-3h.

The existence of the threshold light intensity needed for GraMOS inevitably results to the sub-threshold light intensities regardless of their wavelengths being inefficient for neuronal activation, as shown in Fig. 3c. Yes, there is the actual experimental data supporting the non-interference of light bands: e.g., in Suppl. Video 1. For clarity, we now added a new panel in Fig. 2l, which shows no changes in fluorescence intensity in the background signal (i.e., the area without Fluo4-stained neurons).

6. What are the lighting parameters in Fig 3e? There is a discrepancy between the figure caption (638 nm; 3.9 mW/mm², 5 ms) and the main text (line 248s (2 ms, 638 nm, 3.6 mW/mm²)).

Thank you for pointing out this oversight. We have now corrected it.

7. What do the different colors in Fig 3d, 3e, 4g represent? Do the same colors indicate the same neurons?

The same colors in different figures obviously cannot show the same neurons, because these Figures show different experiments with different biological systems as described in the text of the manuscript.

8. Is there actual experimental data to support the non-interference of light bands described in Fig. 3c? This point is an important support for the valid neural signals in the subsequent Figs 3d-3h.

This is a duplicate of Comment 5. It is answered above.

9. Is there direct data to support the mention of 10% neurons and 70% neurons in the text from line 247 to line 250?

Yes, these numbers are the direct result of analysis of calcium imaging experiments. Since not all numbers are required to be graphed, we chose to provide this information in the text.

10. Since Fig 3e has changed to a 2.5D experimental paradigm compared to Fig 3d, should this key change be reflected in the figure caption?

The information about the experimental paradigm is already provided in the text of the manuscript.

11. In Figs 3f-3h and Supplementary Video 2, it can be seen that the neural impulse propagation within hundreds of micrometers takes nearly ten seconds, while the description in Fig 5c and the text (lines 311-313) states that the neural impulse travels through the entire organoid in only 100 ms. Is this a contradiction, or what is the reason for this?

We thank the reviewer for this thoughtful observation and the opportunity to clarify this point.

(a) Fig. 3 and Suppl. Video 2 show changes in intracellular calcium levels in 2D neurons, while Fig. 5 shows changes in electrical activity in 3D brain organoids. The slower changes in intracellular calcium concentration compared to action potential (AP) propagation in immature hiPSC-derived neurons stem from several biological and biophysical differences between calcium signaling and electrical excitability—especially during early stages of neuronal development.

APs are electrical signals that travel along the neuron's membrane due to the rapid opening and closing of voltage-gated Na⁺ and K⁺ channels.

Changes in intracellular Ca²⁺ levels reflect biochemical signaling, not just membrane voltage. They may involve: calcium influx through voltage-gated Ca²⁺ channels (VGCCs) or NMDA receptors, buffering by intracellular proteins (e.g., calmodulin), sequestration into organelles like mitochondria and the ER, pumping out via Ca²⁺-ATPases or exchangers. These processes are much slower—calcium transients can take hundreds of milliseconds to several seconds to rise and fall.

(b) As is well known, age is a critical factor in development and maturation. The neurons shown in Fig. 3 and Supplementary Video 2 are approximately 3 weeks old, while the brain organoids in Fig. 5a are 4 months old, contributing to differences in the kinetics of their responses.

(c) Finally, the maturity of 2D hiPSC-derived neurons is always significantly delayed compared to 3D hiPSC-derived brain organoids, which also contributes to differences in the kinetics of their responses.

12. Is there literature support for the claim that the excitation-inhibition ratio represents the maturity of the organoid in the text? If so, could it be provided in main text?

We thank the reviewer for bringing up this important point. It is well established that the excitation-inhibition (E/I) ratio typically decreases over the course of neuronal maturation (PMID: 11371348, 31474560, 33714681). This phenomenon arises because, during early development, organoids exhibit a predominance of excitatory activity, driven by the earlier differentiation and integration of excitatory glutamatergic neurons. In contrast, inhibitory GABAergic interneurons mature more slowly and are incorporated into functional circuits at later stages.

13. Is Fig 5a also an AI-generated graph? If so, would it better be indicated in the figure caption?

Information about the use of regenerative AI for cartoons can now be found in Acknowledgements

14. The 4x4 MEA array distribution shown in Fig 5b contradicts the 8x8 MEA array distribution described in Fig 5d and its figure caption.

We appreciate the reviewer's interest in the technical setup.

In principle, MEA plates of various sizes can be used to monitor the electrical activity of brain organoids, both with and without GramOS. In our experiments, we employed both 6-well MEA plates

featuring 8×8 electrode arrays and 48-well MEA plates with 4×4 electrode arrays. Details regarding the MEA configurations used are provided in the Methods section.

15. Would it be more appropriate to change the y-axis label in Fig 5e from "spike increase" to "spike number" to better match the actual meaning?

We thank the reviewer for their attention to detail. The previous version of Fig. 5e has since been removed; however, we would like to clarify that the Y-axis in the former Fig. 5e was correctly labeled as 'Spike Increase'.

16. In the data analysis process described in Fig 6b, why is the electrophysiological signal collected from the MEA first converted into an image and then through OCR to a voltage value matrix? Is not the data directly obtained from the MEA already a voltage value matrix?

In the present manuscript, we no longer use the multistep conversion involving OCR step. We have taken a completely different technical route, when, among other things, we are using the data directly obtained from the MEA.

Reviewer #4 (Remarks to the Author):

This article reports the application of graphene materials for optical neuromodulation through the development of a platform called GraMOS (Graphene-Mediated Optical Stimulation). It has been previously reported that graphene can be optically excited to trigger single action potentials or induce bursting activity via light pulses of varying durations. The main progress of this work is about the use of three-dimensional brain organoids and the light-triggered electrical signals generated by these organoids, which are collected by a multi-electrode array (MEA) and used to drive a robot to perform pre-set actions.

We thank the reviewer for their thoughtful summary.

We would like to clarify that, prior to this submission, we had not reported any results involving the use of the GraMOS platform for optical stimulation of neurons in any form. This manuscript presents the first demonstration of GraMOS-mediated neuromodulation in both 2D and 3D human neural systems.

My Concerns include:

1. There are quite a few missing details, and in several instances, the text and corresponding figures do not align well. The current structure of the manuscript makes it difficult to follow, necessitating significant effort to reorganize and clarify.

We thank the reviewer for their comments. We understand the concern regarding the clarity and structure of the manuscript, and we have made every effort to address the points raised. Although the specific missing details and discrepancies between the text and figures were not fully clear from the reviewer's comments, we have worked diligently to improve the manuscript by incorporating the following changes: we performed additional studies (a) to characterize GraMOS, (b) to demonstrate GraMOS's utility for short-term phenotyping studies, (c) to characterize and analyze two types of interfacing (internal and external) between graphene materials and brain organoids, (d) to demonstrate GraMOS's utility for long-term maturation studies both in 2D and 3D format, and (e) to engineer a completely redesigned experimental platform for the organoid-controlled robotic system.

2. Through capacitive coupling, GraMOS alters the membrane potential of neurons, triggering action potentials in a femtosecond time scale, providing precise and controllable regulation of neuronal activity. The recent development of nanoelectrodes for non-Faradaic capacitive, rapid, and reversible optical stimulation of neurons has been cited in reference #63. Although this work was only published a few months ago, as a parallel development, a discussion comparing this report with reference #63 is needed.

We thank the reviewer for this suggestion. We added a brief discussion of the former Ref #63 (now Ref # 55, PMID: 38195782), as well as another publication from the same group (PMID: 39592569).

3. The key advance of combining a graphene interface with an organoid to control the movement of a Freenove robot via custom Python software is interesting. However, the engineering details are lacking, making it difficult for others to replicate the experiments. More information on this aspect is needed.

In response to the reviewer's valuable suggestions, we have completely redesigned the organoid-robotic engineering system, and provided detailed technical information in the Methods along with a new video of this system in action (Suppl. Video 4).

4. The manuscript mentions that GraMOS is located extracellularly and serves as an optical actuator to stimulate neurons in their natural state without modifications. How is this extracellular positioning characterized? Could it be located on the cell membrane? Evidence needed here. Additionally, the "graphene outside" and "inside" configurations seem to exhibit similar increases in electrical activity in response to light. This section is presented superficially and needs further elaboration.

We thank the reviewer for this valuable comment. To clarify, the extracellular positioning of GraMOS is indeed a critical aspect of its function. Photogenerated electrons can only depolarize the neuronal cell membrane and trigger action potentials when GraMOS is positioned outside the cell membrane (as illustrated in the new Fig. 1b). When neurons are cultured on graphene-coated substrates, the extracellular localization is inherent, allowing GraMOS to induce depolarization, open voltage-gated channels, and trigger action potentials. The same effects are observed when GraMOS is used in free-standing interfaces.

In response to your suggestion, we have added cartoons to further illustrate the external and internal configurations of graphene interfaces with brain cortical organoids (Fig. 5b). Additionally, we have substantially expanded the analysis of the MEA data from both externally and internally interfaced organoids, providing more detailed results in Fig. 5d-l.

5. The authors used a thermocouple probe to measure the graphene surface temperature after illumination at a wavelength of 452 nm, with a light intensity of 3.9 mW/mm² for 30 minutes at a 5 Hz pulse frequency, and no temperature changes were detected. However, the article does not present the corresponding setup and data, which should be included.

In response to the reviewer's request, we have now included temperature data (Fig. 2e) demonstrating that no temperature changes occur during light illumination.

Other minor issues include:

The characterization techniques used for rGO do not mention XRD in line 135 (page 4), but it is mentioned as a technique used on page 5. Additionally, SEM images of GO and rGO should be included to observe the morphology and structure for comparison, which is common in the literature (line 164)

We appreciate the reviewer's suggestion. In response, we expanded the characterization section by including additional SEM images and their analysis, AFM images and their analysis, Kelvin Probe images and their analysis, and GFET experiments.

In Figure S1, the disappearance of the shoulder peak at 300 nm in rGO is mentioned but not discussed in the text when referring to the UV-Vis of rGO. The Raman spectra of rGO are shown in blue, but it would be helpful to add the spectra of GO in red for comparison. This would support the text (line 148), which compares the Raman spectra of GO and rGO. The XRD x-axis should start at 0° to show the full range, as GO has a peak at 10° in the literature. Including the XRD of GO and comparing it to rGO supports the formation of rGO, as mentioned in line 156.

We appreciate the reviewer's suggestion. Our goal was to use characterization methods to show that we were able to convert GO into rGO based on several criteria well established in the field. We believe that we unequivocally established that fact. We do not think that the mechanism behind the disappearance of the shoulder peak at 300 nm is not pertinent to this project, and thus, we have not added this discussion to the text.

Per the reviewer's request, we added the Raman spectra for GO for comparison and extended the XRD axis to clearly show a peak at 10°. Note that due to (a) poor resolution and high background noise below 5° in lab diffractometers, (b) geometrical issues / beam divergence problems at very low angles, and (c) the lack of no valuable XRD signals below 5°, the XRD graphs for graphene often start at around 5° on the X-axis.

In Figure 1c and 1d, for the cell viability assay, three replicates and five fields of view were used, with $n \geq 100$ cells per condition. This allows for selective viewing of more viable cells. It would be better to perform MTS or MTT cell viability assays to avoid exclusion bias. Reference #43 used WST, which is similar to MTS.

We sincerely thank the reviewer for this thoughtful suggestion regarding cell viability assessment methods. We appreciate the opportunity to clarify our rationale for choosing the Live/Dead assay over the MTT assay. A brief explanation has now been included in the Methods section (p. 22). In short, we selected the Live/Dead assay due to several key advantages, including:

(a) MTT assays are destructive and indirect as they rely on metabolic activity (mitochondrial reduction of MTT to formazan). Live/Dead stains are non-destructive and more direct — live cells enzymatically activate green dyes, while dead cells take up red dye due to membrane compromise.

(b) The MTT readout can be significantly affected by electrically conductive cell substrates, because conductive materials (e.g., graphene) can chemically reduce MTT non-enzymatically (i.e., abiotic reduction), resulting in false-positive viability signals — you'll detect high MTT reduction even if the cells are dead or absent.

Importantly, neuroscientists often need to analyze the complex multi-color morphological images for multiple biometers from different experimental groups, which led to the development of standardized image analysis protocols that eliminate bias during analysis. These protocols include such steps as blinded analysis (e.g., randomization and anonymization of image files before analysis; using one person for imaging, and another one for analysis) and automation (no humans) of quantitative image analysis.

In Figure 1f, one variable has $n=9$, while the other has $n=12$. When comparing these, the repeat numbers should be consistent. Additionally, is the $n=9$ and $n=12$ representing mean \pm s.e.m. from independent studies or from the same study?

We sincerely appreciate the reviewer's suggestion and the attention to statistical rigor. While equal sample sizes can be ideal, it is not necessary to have identical numbers of samples in each group to perform valid statistical comparisons. In our current dataset, the sample sizes are $n = 12$ and $n = 14$, which are generally considered very similar for statistical purposes. Furthermore, to minimize variability, neurons used in these experiments were derived from the same preparations and cultured in parallel on different substrates (glass vs. graphene). This approach ensures that the groups are well-matched and suitable for direct comparison of their electrophysiological parameters.

In Figure 2c, where it says "1-s duration for the left trace," do you mean the right trace, as the left trace is already mentioned?

Thank you for pointing out this oversight. We have now corrected it.

Figure 3b should include a control image of neuron cells grown on glass coverslips with Fluo-4 (no graphene).

We appreciate the reviewer's suggestion. The purpose of this panel (now presented as Fig. 2j) is to demonstrate that the optical transparency of the G-substrates allows for conducting calcium imaging studies in live neuronal cultures. This goal is achieved in the revised figure. Regarding the request for a control image of neurons grown on glass coverslips with Fluo-4 (without graphene), we believe this comparison is not necessary. The optical transparency of the G-substrates is clearly shown to support calcium imaging, and including an additional control with glass coverslips would not provide additional insights, as it is well-established that glass coverslips are compatible with calcium imaging. Therefore, we feel that the data presented sufficiently addresses the experimental goal without redundancy.

Lines 374-379 in the discussion need to address how GraMOS provides non-Faradaic capacitive effects and why electrochemical effects cannot be ruled out. It also needs to explain how the lack of pH change after continuous 30-minute exposure plays a role. These points are currently listed but not discussed. Additionally, the paper is missing a conclusion, which would help summarize the key findings and the significance of the research.

We appreciate the reviewer's suggestions, and added a larger conclusion paragraph at the end of the Discussion section. We also added the text (p. 4) to describe and Fig. 1a to illustrate non-Faradaic capacitive effects GraMOS. Although the kinetics of photogenerated electrons from graphene is in the femtosecond range, we cannot rule out that these electrons still may interact with the components of electrolytic cell culture media. If their interactions were noticeable, it would have led to the transfer of electrons between species – in other words, to redox reactions. As electrons are transferred, ions (such as H^+ , OH^-) can move in or out of solution to balance the charge. The movement of these ions directly affects the local pH of the media. The lack of pH changes after continuous 30-minute exposure indicates that these redox processes are practically non-existent. This information is present on lines 522-526.

Reviewer #5 (Remarks to the Author): I co-reviewed this manuscript with one of the reviewers who provided the listed reports. This is part of the Nature Communications initiative to facilitate training in peer review and to provide appropriate recognition for Early Career Researchers who co-review manuscripts.

RESPONSE TO REVIEWER COMMENTS

Reviewer #1 (Remarks to the Author): My comments from the previous round have been addressed by the authors. I have no additional comments.

We sincerely thank the reviewer for their positive feedback and for recognizing the revisions made in response to the previous comments.

Reviewer #2 (Remarks to the Author):

The authors have revised the manuscript according to all the Reviewers' comments, and new experimental results have been provided. Although the manuscript quality is greatly improved, I still have the following suggestions.

We are grateful to the reviewer for their thoughtful feedback and acknowledgment of the manuscript's improvements. Our detailed responses are provided below.

1. Line 38, "... particularly, its ability to efficiently convert light into electricity on a femtosecond timescale ..." For biological modulation, I do not see any advantage of a "femtosecond" response. Microsecond or even millisecond responses are sufficient for stimulating cells.

We thank the reviewer for this question. We agree that in neuroscience, femtosecond timescales may not directly apply to typical neuronal firing or synaptic events, given the physical limitations of molecular interactions and signaling cascades. On the other hand, femtosecond timescales of photogenerated electrons in graphene preclude their potential engagement in redox reactions. For these reasons, and to provide a quantitative description of graphene's optoelectronic properties, we thought it is appropriate to highlight the ultrafast kinetics of photogenerated electrons in graphene.

2. Fig. 1e shows the photos of different samples. Please provide the absorption spectra for these samples (e.g., from 400 to 800 nm). Which sample is used for cell culture and subsequent cell and organoid experiments?

We thank the reviewer for the opportunity to clarify the absorption properties of G-coverslips. In these G-coverslips, the absorption is dictated by the rGO coating, since the underlying glass is optically transparent. As a result, the optical absorption characteristics of the G-coverslips (Fig. 1e) closely match the intrinsic absorption spectrum of rGO shown in Fig. 2c, and differences in optical transmittance arise solely from variations in the thickness of the rGO coating. To illustrate this point, we show absorption spectra for G-coverslips of different optical transmittance (**new Suppl. Figure 4**). In biological experiments, we primarily used G-coverslips with optical transmittance in the 70–80% range. We have revised the manuscript to make this point explicit.

4. Fig. 2e and Line 847, this thermal characterization is inaccurate. First, the local temperature at the cell/graphene interface on the laser spot is much higher than the solution in general; second, the laser spot can create artefacts on the thermocouple. A more accurate thermal measurement can be performed using the patch system. Please refer to: Y. Jiang, *Nat. Biomed. Eng.* 2, 508–521 (2018).

We thank the reviewer for this important comment. We have performed additional experiments and revised the manuscript to indicate that we measured temperature values using three independent methods, including a patch-clamp-based approach that leverages the temperature dependence of electrolyte conductivity. We have now added a **Fig. 2f**, which presents the results of temperature measurements obtained using the patch-clamp method. Importantly, all methods consistently showed no detectable temperature changes during light illumination at the intensity used in GraMOS experiments (e.g., $\sim 3.7 \text{ mW/mm}^2$). For context, Jiang et al., *Nat. Biomed. Eng.* 2, 508–521 (2018), used significantly higher light intensities—e.g., 240 kW/cm^2 for 10 ms—to induce a measurable temperature increase of $\sim 5.4 \text{ K}$ (Fig. 2b in Yang et al., 2018).

5. Line 201, "*did not result in significant changes in the pH*". No measurement results are shown here. In addition, more experiments should be performed to evaluate the reactive oxygen species (ROS) level in the solution.

In response to the reviewer's request, we have now added a graph describing the results of our pH experiments (**new Fig. 2g**) to the revised manuscript and conducted additional experiments to evaluate ROS levels (**new Fig. 2h**) as a function of graphene presence and light exposure.

3. Figs. 2a and 2d show the photocurrent of graphene. Please add the results of current versus time, under illumination. This will reveal the portion of photocapacitive and photofaradic currents. Please refer to: Y. Jiang, *Nat. Biomed. Eng.* 2, 508–521 (2018).

We thank the reviewer for this important question and appreciate the opportunity to clarify why we are confident that GraMOS operates via capacitive rather than faradaic mechanisms.

Faradaic effects involve electron transfer resulting in oxidation or reduction of chemical species, and subsequent current generation from these redox reactions. The occurrence of such redox processes can be reliably assessed by monitoring the levels of reactive oxygen species (ROS) and changes in local pH values. In the revised manuscript, we have now included experimental data and new **Fig. 2f-g** demonstrating that light exposure of G-interfaces does not induce changes either in ROS levels or pH values, strongly indicating the absence of faradaic activity during GraMOS.

In further support of this conclusion, we would like to highlight certain graphene's properties that likely contribute to the lack of faradaic effects in GraMOS. In graphene, photogenerated electrons undergo ultrafast relaxation on the femtosecond timescale due to strong electron-electron interactions and phonon scattering. Because faradaic redox reactions typically require nanosecond to millisecond lifetimes to proceed, the ~1,000,000-fold shorter lifetime of graphene's photoelectrons makes their participation in classical redox processes highly improbable.

While current-versus-time traces can be useful in some contexts, the shape of photocurrents recorded in cell-free conditions and while using electrodes that are not required for optical stimulation does not provide meaningful insight into the mechanisms of light-induced neuronal activation by any nanoactuator. As for graphene, its properties make this situation even more complex due to photogating effects. Graphene, being a zero-bandgap semimetal with high carrier mobility and low density of states near the Dirac point, is extremely sensitive to its electrostatic environment. When light is turned on, photogenerated electrons in graphene induce a rapid displacement of nearby ions, forming an electric double layer. These processes modulate the local electrostatic environment of graphene and create a quasi-static local gating field, resulting in a persistent shift in the Dirac point and Fermi level. Steady-state photocurrents observed throughout light illumination are the results of photogating effects arising from the displacement of ions near the graphene–electrolyte interface.

Moreover, in biological settings, there are additional compounding effects due to the ion displacement near the cell membrane in the presence of the transmembrane electric field. Therefore, the indicators of the occurrence of faradaic redox process at least for graphene-based actuators should be not the photocurrent shapes, but rather the indicators of changes in pH and ROS levels. We hope this explanation clarifies our position, and we thank the reviewer again for raising this important point.

6. Fig. 2l and 2m, the authors should carefully check to raw data for these fluorescence signals. Those narrow spikes should not be calcium spikes, and most likely are the artefacts created by the laser spot. They are dramatically different from normal calcium signals like those in Fig. 3c and 4k. To preclude the possibility of artefacts, control experiments should be performed: (1) on cells grown on pure glass; (2) on graphene samples without any cells.

We thank the reviewer for the opportunity to elaborate further on our calcium imaging experiments. The experiments previously presented in Fig. 2l and 2m (now Fig. 2o and 2p) were conducted using wide-field LED-based optical stimulation, rather than a confined laser spot, which allows us to simultaneously compare multiple experimental outcomes, thereby streamlining interpretation and strengthening our conclusions. We would like to clarify that, although the calcium transients observed in older hiPSC-derived neurons in Fig. 2l may appear relatively narrow, we are confident that these signals represent genuine calcium events for the following reasons:

(a) The duration of neuronal calcium spikes in these experiments is on the order of ~300 ms, which is almost two orders of magnitude longer than the 5-ms light stimulation pulses. This mismatch in timescales indicates that the observed calcium signals cannot be artifacts of the light pulse itself. This 300-ms duration of neuronal calcium spikes in these experiments is consistent with previously published data on neuronal calcium dynamics.

(b) All panels in Fig. 2l represent different regions of interest (ROIs) from the same field of view, all of which were simultaneously exposed to pulsed light stimulation. If the signals were artifacts caused by the light pulses, these artifacts would be expected across all ROIs (whether neuron-, astrocyte, or cell-free regions); however, this is not observed, reinforcing the conclusion that the recorded calcium responses reflect true neuronal activity rather than light-induced artifacts.

(c) The bottom panel in Fig. 2l, titled “Background,” serves as the requested control and corresponds to a graphene area without cells. This provides a reliable control, as we monitored fluorescence signals simultaneously in the same field of view in both cell-containing (the top four panels) and cell-free (the bottom panel) regions.

Finally, Fig. 2m shows well-resolved individual calcium spikes occurring within bursts in response to prolonged, continuous, step-like light stimulation. These calcium responses do not match the temporal profile of the light pulses. Instead, their pattern closely resembles the light-evoked action potentials seen in the bursting responses shown in Fig. 2h, further supporting the physiological origin of the recorded calcium signals.

7. Fig. 3h, unit is missing.

In response to this helpful comment, we have now added the appropriate unit to the Fig. 3h legend.

8. Fig. 3j-3m, there is no control group here (without light / without graphene).

We thank the reviewer for this important question. We would like to clarify that hiPSC-derived neurons used in Fig. 3j–3m are very immature in their electrophysiological development and do not exhibit spontaneous activity at this stage. Moreover, only neurons interfaced with graphene respond to light stimulation. As such, control groups without light or without graphene are not expected to show any activity. We have added this clarification to the revised manuscript to explain why separate control groups are not required in this case.

9. Line 325, it is not “Fig. 3n”, but “Fig. 3m”.

We thank the reviewer for their careful proofreading and attention to detail. We have corrected the figure reference from “Fig. 3n” to “Fig. 3m” in the revised manuscript.

10. Fig. 4, captions are messed up, no 4g-4k.

We sincerely thank the reviewer for catching our mistake. We have corrected the figure caption for Fig. 4 and ensured all labeling is now accurate and consistent across the revised manuscript.

11. Fig. 6 in the original draft is completely deleted. Some images may be valuable and should be included in the main figure or supporting figures.

We appreciate the reviewer’s suggestion and understand the importance of providing clear visual documentation of the experiment. We would like to note that Figure 6 was removed from the original draft at the request of other reviewers, who found it to be uninformative. We agree with this recommendation, because, in this experiment, the robot movement controlled by light-activated, G-interfaced brain cortical organoids is best represented dynamically. To that end, we have provided a high-resolution video (Suppl. Video 4) that simultaneously displays both organoid activity and robot behavior in a split-screen format. This format offers a more comprehensive and intuitive representation of the experiment than static images could convey.

In addition, we included detailed descriptions of the setup and results in the main text and provide the custom software used for communication between the organoid and robot as supplementary material. As there are no static images that offer further meaningful insight beyond what is already conveyed through the video and description, we believe including still frames would be redundant. However, if the reviewer believes it is essential to include a representative image, we would be happy to extract and include a still frame from the Suppl. Video.

12. Non-academic words like “fortunately”, “well-being”, ... are used a lot in the paper.

In response to the reviewer’s feedback, we have carefully revised the manuscript to replace informal expressions with more precise and objective language that aligns with academic writing standards.

Reviewer #3 (Remarks to the Author):

After carefully reviewing the authors’ rebuttal letter, I would like to stand by my previous evaluation. The key concern regarding the acceptance of this manuscript lies in its level of novelty. The authors claim that GraMOS represents the first graphene-based optoelectrical stimulation approach to activate neurons. However, in this field, similar work has already been demonstrated: (1) non-graphene materials have been used for photoelectrical neural stimulation (e.g., from Prof. Bozhi Tian at UChicago and Prof. Xing Sheng at Tsinghua), and (2) graphene has been employed for photothermal stimulation (e.g., from Prof. Tzahi Cohen-Karni at CMU). Although the authors correctly point out that temperature rise may be a concern in photothermal methods, the rationale for choosing graphene in their system is not clearly articulated. Given the extensive prior research on non-genetic, light-induced neural stimulation—see, for example, the Science essay in 2019 (DOI: 10.1126/science.aay4351)—I would characterize this manuscript as an “A+B” or “me-too” type of study. As such, I believe it falls below the threshold of novelty expected for Nature Communications.

We appreciate Reviewer #3’s engagement with our work and the opportunity to clarify the unique contributions of our study. While we understand the intention behind the A+B argument, we respectfully believe there may be a misunderstanding regarding the novelty of our approach.

Specifically, our work presents the first demonstration of graphene-based optoelectrical neuromodulation, as none of the studies cited by Reviewer #3 harness the optoelectronic properties of graphene to optically activate neurons. As outlined in our manuscript, these unique (not present in other materials) optoelectronic properties of graphene were pivotal in enabling our thin, transparent graphene interfaces to efficiently and reliably activate neurons by light.

In support of their argument, Reviewer #3 brought up silicon nanowires with randomly grown out-of-plane graphene flakes (NT-3DFG) developed by Prof. Cohen-Karni at CMU, solely based on its inclusion of the word "graphene" in its name. However, NT-3DFG operates by absorbing nearly all incident light (thus not transparent), due to light trapping caused by densely packed out-of-plane graphene flakes growing in all directions on silicon nanowires. It requires extremely high-intensity light to generate significant local temperature increases, thereby achieving photothermal neuron activation. Therefore, neither the structure nor the function of NT-3DFG shares commonalities with the GraMOS platform.

Reviewer #3 also cited the existence of several non-graphene materials with photothermal and photoelectrochemical properties (e.g., from Prof. Tian at UChicago). Similar to NT-3DFG, neither structure nor function of these materials has commonalities with the GraMOS platform.

We fail to see how arbitrarily extracting the term "graphene" from NT-3DFG (Component B) and merging it with the photoelectrochemical properties of other unrelated materials (Component A) may result in our graphene-based optoelectronic platform as "A + B", either structurally or functionally.

We would also like to emphasize that this graphene-based optoelectronic technology has been granted several patents across four jurisdictions, which strongly supports its novelty.

Finally, we would like to highlight that the GraMOS platform offers a unique combination of distinct material and functional properties that make it exceptionally well-suited for optical neuromodulation:

1. Its transparency ensures full compatibility with optical monitoring techniques, allowing simultaneous stimulation and imaging - an essential capability for studying real-time neuronal dynamics. In contrast, the NT-3DFG platform is completely optically impenetrable.
2. The broad light absorption spectrum provides unparalleled flexibility in the choice of stimulation wavelengths, enabling a much broad range of practical applications. In contrast, the existing nanoactuators have only specific light wavelengths for activation.
3. GraMOS achieves highly efficient light-to-electricity conversion due carrier multiplication effects in graphene, which allows the use of light intensities in the range of 1- 5 mW/mm². In contrast, as highlighted in our manuscript, many existing nanoactuators need 100 - 10,000-fold higher light intensities to operate.
4. The absence of thermal effects, enabled by graphene's high thermal conductivity, ensures that neuronal activation occurs without heating the surrounding tissue. This is particularly critical in neural systems, where even slight temperature increases can disrupt normal brain function, alter cellular behavior, or induce damage during prolonged stimulation. In contrast, many existing nanoactuators (NT-3DFG platform, gold nanoparticles, silicon-based nanoactuators) either rely on or inadvertently produce photothermal effects.
5. Unlike many other nanotechnology-based actuators, G-interfaces can be fabricated not only in free-standing, but also in a stable, substrate-supported format, expending the technical capabilities of probing the functionality of neuronal networks.
6. A very short (femtosecond) lifetime of photogenerated electrons in graphene contributes to the lack of Faradaic effects in GraMOS due to insufficient (by 10⁶-fold) time for their engagement in electron transfer with other chemical species. At the same time, G-interfaces can act as ROS scavengers, because this process is not time-limited due to ROS kinetics. This unique combination of properties is not observed in other nanoactuators, which means they are more likely to induce unwanted redox reactions or oxidative stress under similar conditions.
7. Because GraMOS operates via a capacitive mechanism, the platform is exceptionally biocompatible and can be used repeatedly over extended periods to support month-long neurodevelopmental, phenotyping, and therapeutic studies. To our knowledge, no other nanoactuators have demonstrated this level of sustained, non-invasive functionality in long-term biological applications.

Reviewer #4 (Remarks to the Author): The authors have significantly enhanced the revised manuscript by incorporating extensive experimental data and expanded discussions, which effectively address the majority of prior concerns. However, critical limitations persist in demonstrating how light-evoked electrophysiological signals from graphene-interfaced cerebral organoids translate into robotic locomotion control. The current

reliance on a qualitative demonstration video, devoid of quantitative metrics or methodological rigor, remains insufficient for robust scientific validation.

We thank the reviewer for their thoughtful assessment and for recognizing the substantial improvements made to the manuscript through additional experimental data and expanded discussions. In response to the remaining concerns, we have carefully addressed all points raised in both the revised manuscript and the detailed responses below. We hope these clarifications and additions satisfactorily resolve the reviewer's concerns.

Major Comments:

1. In the revised manuscript introduction, the authors mention developing graphene-based optical actuators but do not highlight how the newly added data demonstrate the advantages of this development. The authors should highlight the key innovations and novelty of this work in the paragraph (line 86-91).

We thank the reviewer for this valuable feedback. We have expanded the paragraph above to briefly summarize the key innovations of our work. We hope this revision more effectively communicates its significance.

2. While the revised manuscript includes new data on G-coverslips with optical transmittance, the authors should have described the total graphene concentration without establishing the relationship between concentration/mass and optical transmittance.

We thank the reviewer for this constructive comment. In the revised manuscript, we have now clarified the relationship between rGO concentrations, deposition methods, and the resulting optical transmittance of G-coverslips. Specifically, we have revised the Results and Methods sections to explicitly describe how both drop-casting (0.1–0.5 mg/mL) and spray-coating (0.5 mg/mL applied in 5 to 25 discrete deposition steps) were used to achieve target transmittance values in the range of 50% to 90%.

3. The temperature measurements shown in Figure 2e lack critical details. The manuscript does not specify whether the detected temperature corresponds to the illuminated area or the entire graphene substrate. The full details of the temperature sensing setup and measurement location are needed.

We appreciate the reviewer for pointing out this important oversight. The temperature values shown in Fig. 2e were recorded only from the illuminated area of the G-coverslips. We have revised the Methods section to clarify how these measurements were conducted. Additionally, to obtain more precise local temperature data, we performed complementary experiments by relying on the temperature dependence of electrolyte conductivity, whereby changes in local temperature were inferred from patch pipette resistance during electrophysiological recordings. Details of this approach and the corresponding results are included in the revised manuscript, along with **new Fig. 2f**.

4. The manuscript states a ~20% success rate for activating individual neurons via optical stimulation in G-interfaced organoids. The methodology for calculating this value requires clarification. Additionally, details about light penetration depth through the entire organoid should be included and discussed.

We thank the reviewer for raising this important point. The ~20% success rate for activating individual neurons via GraMOS in G-interfaced organoids was calculated by assessing the ratio of neurons responding to a single-cell laser pulse to the total number of neurons tested by the pulse. This information has now been added to the revised Methods section. Since only clearly visible fluorescent Fluo-4-labeled neurons were subjected to GraMOS in our experiments, we are confident that light penetration was not a limiting factor in evaluating neuronal responsiveness to GraMOS.

5. A discrepancy exists in the wavelength selection: UV light was used for assessing rGO flake photosensitive, while 452 nm light was employed for G-interfaced neuron stimulation. The rationale for this inconsistency needs explanation, particularly considering the known cellular damage caused by UV radiation.

We thank the reviewer for the opportunity to clarify the technical characteristics of GraMOS. We fully agree that UV light is not suitable for biological experiments due to its detrimental effects on cells, and we do not intend to use it for such purposes. The UV-based GFET experiments were intended only to provide orthogonal evidence of the photosensitivity of the graphene materials used in our study and to complement our experiments conducted with visible light (Fig. 2a–e). Since the absorption spectrum of graphene is broadband and relatively flat across the UV–visible–NIR range (Fig. 2c) with well-characterized features, by knowing the absorption spectrum of our materials, we can reasonably infer its photoresponse characteristics across the visible spectrum, including at 452 nm. This allows us to predict the presence of photocurrents under visible light illumination without requiring measurements at every specific wavelength. The photocurrent responses we observe under 452 nm

light during neuromodulation experiments are consistent with these optical absorption properties and confirm effective light-graphene interaction in the biologically safe visible range.

6. The revised manuscript mentions 2-Hz optical training of neurons and organoids, but does not quantify the illuminated area or the light-power density. These parameters must be provided.

We thank the reviewer for this important point. We have now included the relevant parameters in the revised manuscript. During the light training, wide-field illumination was applied to the entire area of graphene-interfaced brain cortical organoids. The light intensity used in GraMOS-driven maturation studies was 1.9 mW/mm². We have updated the Methods section in the revised manuscript to include this information.

7. The observation in Figure 4g showing a decreased VGLUT1/GAD65/67 ratio (from ~20% in controls to ~8% in graphene groups) suggests a shift toward inhibitory balance in excitation/inhibition equilibrium. The authors should discuss the potential mechanisms underlying this phenomenon.

We thank the reviewer for this insightful observation. One likely explanation to the changes in the VGLUT1/GAD65/67 ratio relates to developmental changes during neuronal maturation. In early stages of cortical development, excitatory glutamatergic neurons typically emerge and mature earlier than inhibitory GABAergic interneurons. However, as the network matures, GABAergic circuits progressively increase in number and functional influence, contributing to the fine-tuning of excitation/inhibition (E/I) balance. Graphene interfacing may influence this trajectory by promoting network maturation or altering local microenvironments, potentially accelerating the functional integration of GABAergic neurons or modulating their synaptic activity. We have now included this discussion in the revised manuscript to address the potential mechanisms underlying the observed shift in the VGLUT1/GAD65/67 ratio.

Other minor issues include:

1. Figure citation errors require correction: The schematic of the light illumination module stimulating neurons is incorrectly labeled as Figure 3k (should be Figure 3j). References to non-existent Figure 3n should be removed. The Figure 3 description appears disorganized and needs thorough verification.

We sincerely apologize for this oversight. We have carefully reviewed and corrected the labeling errors, including the misattribution to Figure 3k (now accurately cited as Figure 3j), and non-existent Figure 3n (now accurately cited as Figure 3m).

2. Identical sentences appear at lines 536 and 542. Please remove duplicate text.

We rephrased the above-mentioned paragraph to remove duplicate phrases.

3. Some word choices (e.g., “multi-faceted”) may be imprecise.

We removed the term “multi-faceted” from the revised manuscript to ensure greater precision in our language.

4. In Fig. S2b–c and Fig. S4a, the rGO thickness is reported as 2.9 ± 0.3 nm. Does this correspond to multilayer flakes?

A single graphene layer is ~ 0.34 nm thick. However, due to substrate interactions, adsorbates, or tip convolution in AFM measurements, the apparent thickness of a single graphene layer in these measurements may appear closer to 0.5–1 nm. The thickness of the rGO flakes shown in Fig. S2b–c (2.9 ± 0.3 nm) and Fig. S4a (1.44 – 1.81 nm) suggests that our rGO flakes is a mixed population few-layer and multilayer flakes.

5. The manuscript states that graphene’s electronic band structure enables broadband absorption from ultraviolet to infrared wavelengths. Please specify the exact wavelength range tested or expected.

We thank the reviewer for this important question. Graphene’s broadband optical absorption (Fig. 2c) arises from its gapless linear electronic band structure, which enables interband transitions across a wide spectral range. While our biological experiments primarily utilized visible light (e.g., 452 nm, 561 nm), it is well established in the literature that monolayer and few-layer graphene absorb light from the ultraviolet (~300 nm) through the visible spectrum and into the mid-infrared (~2500 nm) (PMID: 18388259). This broad absorption range is intrinsic to graphene and has been extensively characterized in both theoretical and experimental studies. Due to graphene’s broadband absorption, any light wavelength can be used for GraMOS, and the specific choice of particular wavelength can be guided by secondary considerations, such as the need for enhanced light penetration depth or the avoidance of optical crosstalk in all-optical studies.

RESPONSE TO REVIEWERS' COMMENTS

Reviewer #2 (Remarks to the Author): The authors have revised the manuscript according to all the Reviewers' comments, and new experimental results have been provided.

We thank the reviewer for kindly acknowledging the improvements in our revised manuscript and the addition of new experimental results.

In my previous review report, there are some comments that have not been addressed:

My previous comment #3: "Figs. 2a and 2d show the photocurrent of graphene. Please add the results of current versus time, under illumination. This will reveal the portion of photocapacitive and photofaradic currents. Please refer to: Y. Jiang, *Nat. Biomed. Eng.* 2, 508–521 (2018)." The authors provide a long and obscure response to this comment. I do not understand why they could not simply provide the experimental results of the photocurrent versus time.

Per the reviewer's request, we have added a photocurrent trace from light-illuminated G-interfaces in Fig. 2b.

We would like to clarify that, when writing our previous response to comment #3, we understood this question as aiming to assess the GraMOS mechanism based on the shape of the photocurrent. Accordingly, we conducted additional experiments and responded that, based on our experimental results (Fig. 2e, 2f), GraMOS photofaradaic and photothermal effects are negligible, if not absent altogether. Further, the photocurrent shapes in p/i/n type semiconductors (e.g., Jiang *et al.*, *Nat. Biomed. Eng.* 2, 508–521 (2018)) and in graphene differ fundamentally due to their underlying mechanisms: in a p-i-n silicon-based materials, light generates transient capacitive photocurrents with a faradaic component due to carrier separation and interfacial charge transfer. In contrast, in graphene, photogenerated electrons displace nearby ions at the graphene–electrolyte interface, forming an electric double layer. This leads to a quasi-static local electrostatic field that shifts the Dirac point and Fermi level, resulting in steady-state photocurrents. Therefore, the shape of the photocurrent in graphene does not carry information about the relative contributions of photocapacitive and photofaradic currents.

My previous comment #6: "Fig. 2l and 2m, the authors should carefully check to raw data for these fluorescence signals. Those narrow spikes should not be calcium spikes, and most likely are the artefacts created by the laser spot. They are dramatically different from normal calcium signals like those in Fig. 3c and 4k. To preclude the possibility of artefacts, control experiments should be performed: (1) on cells grown on pure glass; (2) on graphene samples without any cells." I still have doubts about these data (Fig. 2o and 2p in the revised paper). Please provide the raw data (excel sheet) for these plots.

Per the reviewer's request, we have attached the Excel file (see below) with the data presented in Fig. 2o and Fig. 2p. We would also like to note that the original video for Fig. 2o showing the lack of optical crosstalk was previously included with the manuscript as Supplementary Video 1.

The key evidence supporting that the traces in Fig. 2o represent calcium spikes rather than light artifacts lies in their distinct temporal characteristics: the duration of calcium transients (~300 ms – see the Zoom-in on the 1st light-triggered calcium spike in the Excel file below) is significantly longer than the 5-ms light pulses, making it highly unlikely they are caused by optical artifacts. In Fig. 2p, the light pulse lasted 2 seconds, during which neurons exhibited a burst of several calcium transients—similar to the activity shown in Fig. 2k. This pattern of activity is consistent with light-evoked neuronal firing and not with an artifact, and we respectfully believe that any interpretation to the contrary is not supported by the data.

Reviewer #4 (Remarks to the Author): The authors have provided a detailed response to the question raised and have offered a clear explanation of the underlying mechanism(s). My major comment is still around "biohybrid robotics" at this stage of demonstration, the role of light-evoked electrophysiological signals from graphene-interfaced cerebral organoids in robotic locomotion control remains very basic, so the claim of biohybrid robotics is over claim, suggest to detone here.

We are very grateful to the reviewer for the thoughtful feedback and for recognizing the clarity of our response and mechanistic explanation. We appreciate the suggestion regarding the biohybrid robotics claim and revised our manuscript to appropriately, emphasizing that it is presented solely as a proof-of-concept at this stage.

Some additional minor corrections need further clarification.

1. The "n-doped" in Fig. 1a is not an accurate way to describe the transport mechanism. No dopant is applied in this work. I assume that the author would like to say it is "n-type", can the authors explain the "n-doped" instead of "n-type" in details?

We sincerely thank the reviewer for pointing out this important detail. We indeed intended to describe the material as “n-type”, and have now corrected Fig. 1a.

2. Line 683, the author describes the fabrication of the rGO-FET device in Figure 5b. But Figure 5b does not provide the information. And please illustrate the IDE material in this device.

We apologize for this oversight. We have made the correction in the revised manuscript to state that the IDE used in the GFET device is illustrated in Supplementary Figure 6b, not in Figure 5b.

3. The authors have updated Fig. 4 to demonstrate the excitation/inhibition equilibrium of the neuronal network. Indeed, the complex of the neuronal network will affect the excitations and inhibition, but the authors' explanation is still ambiguous, please find more specific explanation in the reference (<https://doi.org/10.1038/s41467-020-17521-w>).

We appreciate the reviewer's suggestion to mention the reference describing the GABA polarity switch in brain organoids. However, we would like to clarify that the observed reduction in the VGLUT1/GAD65/67 ratio in our study and the GABA polarity switch are two distinct developmental phenomena that reflect different aspects of neurotransmission maturation.

The GABA polarity switch refers to the intracellular chloride gradient change during early brain development, whereby GABA receptor-mediated responses transition from depolarizing (excitatory) to hyperpolarizing (inhibitory). This switch is primarily regulated by chloride transporters such as NKCC1 and KCC2, and reflects the functional state of GABAergic signaling at the postsynaptic level. In contrast, the VGLUT1/GAD65/67 ratio reflects presynaptic neurotransmitter phenotype expression, specifically the balance between excitatory (glutamatergic) and inhibitory (GABAergic) terminal markers. Therefore, while both processes are part of overall circuit maturation, they occur at different molecular levels and represent non-overlapping mechanisms.

4. In Fig. 3b, the scale bar is μm rather than μM .

We appreciate the reviewer pointing out this oversight – we have corrected the scale bar in Fig. 3b.

5. In Fig. 4e and 4g, it is better to align the scale bars well. In the caption, h is wrongly labeled as j.

We thank the reviewer for bringing this to our attention—we have aligned the scale bars in Fig. 4e and 4g and corrected the labeling error in the caption.

6. In Supplementary Figure 1d, the peak around 3000 cm^{-1} is D+D' instead of D+D.

We have corrected the label to D+D' in Supplementary Figure 1d.

FIGURE 2o

Time (ms)	Neuron 1	Neuron 2	Neuron 3	Astrocyte	Background
0	600.767	548.676	513.592	522.190	634.609
30	597.150	552.857	510.804	519.483	638.043
60	593.033	547.019	510.706	516.690	632.348
90	590.367	549.857	512.883	519.000	630.652
120	594.067	547.562	511.141	517.741	634.957
150	596.317	547.800	509.255	517.897	631.913
180	598.100	548.010	512.491	518.690	636.609
210	593.033	548.629	509.500	515.345	636.000
240	596.383	549.762	511.037	519.000	633.130
270	598.233	550.891	509.089	517.793	635.957
300	596.167	550.095	511.356	518.517	630.565
330	595.883	549.100	510.607	514.983	637.522
360	598.117	550.219	510.862	515.276	638.696
390	595.767	548.819	510.914	516.897	628.913
420	593.333	548.067	510.543	515.379	632.913
450	592.017	548.529	510.610	515.552	642.826
480	592.817	550.014	511.113	517.276	630.957
510	599.433	548.743	509.721	518.466	635.913
540	602.350	550.181	511.653	518.724	631.348
570	596.133	548.176	508.684	517.914	635.826
600	593.767	546.690	510.580	517.569	630.522
630	597.917	549.076	512.107	516.828	638.522
660	596.000	549.000	510.601	511.845	625.391
690	592.217	546.790	510.080	515.897	633.870
720	597.350	547.076	510.055	516.707	628.870
750	600.467	549.452	510.199	516.431	636.348
780	598.167	547.629	509.160	518.086	635.391
810	592.267	545.343	509.776	514.948	629.217
840	592.750	547.795	509.819	518.483	635.391
870	595.467	547.962	510.586	515.517	641.435
900	596.267	545.381	510.264	513.397	630.000
930	591.450	546.781	511.261	515.121	627.957
960	594.500	547.495	512.040	519.138	628.261
990	595.917	548.229	510.957	516.724	632.130
1020	592.367	549.100	510.368	520.793	629.870
1050	593.700	550.329	510.420	515.793	635.087
1080	592.067	549.895	511.405	516.810	629.217
1110	593.350	548.329	508.936	516.517	633.739
1140	593.667	548.862	510.080	519.707	625.739
1170	597.433	549.905	511.067	517.086	629.652
1200	595.367	547.776	511.037	517.724	621.435
1230	599.300	548.933	510.147	512.431	634.826
1260	591.217	549.481	510.613	516.310	627.913
1290	599.133	548.248	510.193	518.293	636.565
1320	595.317	548.095	510.482	521.000	631.174
1350	592.483	548.538	510.926	516.155	628.304
1380	593.450	545.657	508.644	516.155	635.652
1410	592.200	547.386	511.340	520.431	635.609
1440	594.367	545.410	509.982	515.414	631.174
1470	594.350	549.105	509.077	521.017	625.826
1500	589.050	547.805	508.000	515.534	633.522
1530	600.033	548.714	508.798	513.879	633.087
1560	595.767	549.762	509.463	517.138	635.000
1590	594.867	548.614	511.552	514.793	633.609
1620	596.517	548.062	509.850	517.379	636.696
1650	591.350	548.910	509.031	514.138	624.957
1680	599.033	548.771	510.791	518.086	626.478
1710	590.617	547.781	508.890	514.655	635.174
1740	590.183	547.948	508.724	518.293	633.696
1770	594.700	546.005	509.571	516.172	635.478
1800	591.550	548.624	507.819	512.707	629.522
1830	590.683	548.662	510.939	517.207	637.870
1860	595.800	547.010	508.859	514.759	633.652
1890	599.733	547.095	509.494	514.293	633.522
1920	602.033	548.100	510.031	513.086	630.739
1950	595.517	548.229	508.212	517.552	637.043
1980	592.867	549.157	512.120	514.776	625.696
2010	596.483	546.781	509.770	517.017	631.130
2040	592.467	548.490	508.939	514.914	631.261
2070	591.283	547.333	510.534	513.534	639.000
2100	591.500	545.500	511.110	514.603	636.478
2130	595.967	548.695	510.586	520.328	633.696
2160	594.150	544.176	508.948	514.500	632.435
2190	594.017	547.690	510.859	516.207	636.000
2220	593.333	545.738	509.350	511.552	629.826
2250	593.917	548.190	512.518	511.310	624.391
2280	591.700	548.348	513.043	513.379	630.435
2310	596.733	546.790	510.009	517.724	634.478
2340	592.133	548.481	511.975	512.603	629.565
2370	593.733	548.757	509.408	513.500	631.348
2400	590.217	547.886	508.472	517.241	634.696
2430	594.667	550.686	510.748	516.500	630.217
2460	594.867	548.038	508.681	514.362	628.130
2490	595.817	547.600	510.844	517.690	633.957
2520	593.783	548.338	509.218	517.655	624.696
2550	593.533	548.676	509.175	517.345	629.087
2580	591.600	550.371	508.656	518.138	631.522
2610	593.683	546.833	510.696	516.155	626.391
2640	592.483	543.986	509.939	517.052	638.130
2670	593.817	548.762	508.883	515.552	628.957
2700	591.033	548.867	508.669	517.534	625.304
2730	600.500	547.481	507.635	516.879	632.391
2760	593.317	549.938	510.356	514.345	629.739
2790	593.550	547.505	509.206	514.276	636.348
2820	594.267	545.652	509.816	514.845	628.652
2850	589.483	546.614	509.371	515.155	633.087
2880	595.083	548.962	507.850	513.879	623.435
2910	598.500	547.152	510.963	517.379	622.522

2940	590.367	548.605	510.362	514.914	630.043
2970	592.900	547.881	509.371	517.103	630.217
3000	596.617	547.757	510.718	515.207	630.043
3030	592.833	550.662	510.006	514.483	625.435
3060	593.150	548.686	511.166	512.224	627.913
3090	596.700	547.676	511.420	514.983	630.696
3120	597.550	547.971	509.678	516.500	629.696
3150	593.917	548.129	510.509	514.310	630.609
3180	594.500	546.876	511.221	513.207	629.522
3210	594.550	546.071	510.230	515.621	632.043
3240	591.433	548.790	508.718	511.034	631.261
3270	589.650	547.933	508.653	514.500	635.783
3300	595.200	548.867	510.451	517.397	629.130
3330	591.917	548.595	510.380	514.448	630.435
3360	590.300	547.219	508.883	516.052	633.130
3390	593.033	547.719	508.966	516.190	632.261
3420	594.817	547.771	509.589	516.190	632.652
3450	595.800	548.586	509.733	514.207	626.696
3480	594.350	548.043	509.494	514.448	631.130
3510	592.917	548.700	509.463	516.724	624.609
3540	591.600	545.676	510.439	515.276	627.913
3570	591.983	549.138	508.969	513.500	633.130
3600	591.033	547.590	512.482	511.293	628.043
3630	591.900	544.667	508.776	514.000	635.174
3660	589.650	544.719	510.061	516.241	630.870
3690	596.650	546.614	509.396	515.603	628.565
3720	591.867	550.810	509.617	512.931	628.826
3750	591.433	547.243	509.457	514.259	624.522
3780	592.967	545.743	509.113	512.293	637.696
3810	592.917	548.762	509.405	513.328	636.217
3840	588.600	544.952	509.150	518.345	628.870
3870	590.983	548.090	507.994	518.052	622.391
3900	592.250	545.310	508.500	514.724	628.957
3930	589.767	544.952	511.463	516.707	632.130
3960	595.267	546.757	509.294	511.948	629.348
3990	589.200	546.067	509.031	515.690	629.783
4020	594.250	548.010	509.160	516.034	630.652
4050	593.317	547.738	509.963	510.948	627.261
4080	591.483	545.724	509.417	512.069	632.609
4110	590.733	548.186	508.877	512.034	625.522
4140	596.267	547.419	507.672	511.052	622.478
4170	592.233	546.109	507.387	513.948	637.348
4200	591.767	545.967	510.948	511.638	632.609
4230	591.267	548.519	508.791	514.034	627.348
4260	593.717	545.248	507.058	515.483	620.130
4290	592.700	546.100	512.034	514.345	622.043
4320	595.017	545.538	509.745	514.655	632.000
4350	595.300	549.148	508.242	513.793	624.304
4380	594.550	548.310	509.905	512.500	629.087
4410	592.217	547.281	509.663	513.569	631.870
4440	592.450	549.143	509.877	506.224	632.087
4470	598.400	547.490	508.451	517.672	625.478
4500	590.800	549.524	510.120	512.448	630.043
4530	595.283	547.767	510.267	514.707	634.043
4560	592.500	546.933	511.482	513.897	627.043
4590	588.900	546.410	509.650	515.328	627.565
4620	589.433	545.914	509.310	515.466	627.609
4650	590.250	544.410	509.469	512.379	631.174
4680	588.733	545.581	509.402	513.224	628.261
4710	591.450	547.729	509.844	515.121	627.565
4740	588.267	546.438	510.212	514.828	628.957
4770	589.883	546.838	508.770	513.966	623.435
4800	594.300	546.843	508.856	513.431	627.565
4830	599.783	547.771	509.549	510.017	632.043
4860	600.933	546.891	508.337	514.914	625.609
4890	589.383	544.881	507.638	519.707	637.913
4920	594.750	548.433	508.653	510.000	623.087
4950	595.567	545.929	509.472	514.155	628.217
4980	589.500	547.090	509.276	513.293	627.826
5010	592.633	546.329	509.313	513.121	630.087
5040	594.250	548.324	508.258	513.603	626.609
5070	589.433	547.024	509.739	515.948	626.609
5100	596.733	546.790	508.733	514.966	631.348
5130	593.217	547.624	510.147	515.724	629.043
5160	589.667	547.071	510.423	512.017	627.217
5190	594.733	547.605	509.380	514.483	625.130
5220	591.167	547.957	508.675	516.086	625.696
5250	594.517	546.467	510.242	511.345	628.217
5280	591.317	550.676	508.583	517.328	630.957
5310	595.867	547.095	508.908	509.655	627.739
5340	596.283	547.271	508.727	513.862	639.739
5370	594.683	545.448	510.954	512.466	625.304
5400	591.000	548.000	506.715	515.845	623.087
5430	592.400	549.229	510.972	516.190	631.696
5460	590.683	547.805	508.730	510.569	633.000
5490	592.367	546.157	508.914	513.948	634.217
5520	590.817	544.733	510.104	515.931	626.348
5550	592.900	548.381	510.610	511.638	621.609
5580	591.467	547.210	508.669	514.828	630.217
5610	594.533	546.548	508.518	514.241	632.739
5640	594.583	546.419	508.252	518.310	623.826
5670	594.083	545.971	508.368	510.224	626.652
5700	592.767	547.819	508.482	513.655	620.652
5730	595.150	545.910	508.742	508.931	626.174
5760	593.317	545.814	509.460	512.241	632.522
5790	589.350	545.795	508.794	512.810	629.522
5820	592.133	546.690	510.463	518.345	623.739
5850	591.950	544.914	507.837	516.500	627.522
5880	586.700	548.171	510.218	511.966	627.217
5910	594.433	546.838	507.896	514.517	630.348
5940	593.617	545.614	507.798	511.914	623.174

5970	591.367	546.705	510.147	511.948	630.913
6000	590.767	546.695	509.242	511.121	624.870
6030	592.867	546.329	509.126	514.983	624.739
6060	593.617	548.062	509.337	512.362	636.522
6090	589.383	546.062	508.963	511.810	625.391
6120	590.617	548.286	507.230	513.828	628.043
6150	590.050	546.010	509.644	512.172	625.043
6180	591.200	549.690	510.325	511.586	628.348
6210	590.633	547.729	508.617	513.862	636.087
6240	597.150	544.881	508.518	516.517	630.696
6270	591.717	547.338	508.936	515.259	627.913
6300	593.900	547.424	510.267	511.828	639.957
6330	588.100	544.805	510.117	514.586	638.304
6360	596.417	547.843	508.583	518.241	633.870
6390	589.017	546.186	509.064	517.500	624.913
6420	593.217	548.424	508.193	520.310	636.652
6450	592.250	546.257	509.770	523.707	626.870
6480	589.567	547.933	507.985	527.397	625.783
6510	589.983	545.848	508.463	530.776	624.348
6540	589.283	547.829	508.592	530.483	633.348
6570	596.467	546.562	507.218	529.017	626.957
6600	592.350	548.224	508.923	529.759	620.565
6630	598.133	544.648	511.242	532.155	636.435
6660	587.717	544.095	509.834	529.431	632.826
6690	590.383	548.391	509.074	535.397	634.609
6720	590.200	546.514	506.954	531.569	632.609
6750	592.333	547.152	509.037	531.328	633.826
6780	594.150	546.048	510.175	534.121	632.391
6810	593.217	547.095	509.485	537.483	628.304
6840	597.217	547.990	509.853	533.155	630.348
6870	589.783	547.395	510.334	530.603	633.000
6900	593.983	547.843	508.077	530.759	639.565
6930	588.283	547.067	507.859	533.466	631.261
6960	594.833	545.857	508.647	533.431	631.174
6990	590.683	547.357	508.798	531.517	636.609
7020	591.650	546.171	507.209	530.276	631.000
7050	593.650	547.671	508.178	533.121	629.348
7080	593.783	547.062	509.985	530.362	626.826
7110	587.950	544.905	508.896	530.517	621.261
7140	592.517	548.652	508.936	532.034	630.217
7170	590.450	547.848	508.822	533.672	629.957
7200	594.333	546.124	510.408	536.759	631.522
7230	592.317	547.386	507.779	532.190	631.826
7260	596.000	547.438	508.887	535.862	628.391
7290	588.683	545.910	509.123	535.172	626.391
7320	591.150	545.395	507.310	529.948	628.087
7350	593.500	547.152	510.230	532.155	627.565
7380	593.200	547.805	508.567	533.207	629.174
7410	590.283	545.162	507.172	529.966	629.130
7440	593.017	548.514	512.083	529.483	620.217
7470	597.750	563.129	530.844	534.879	628.826
7500	610.098	564.790	529.957	535.966	630.478
7530	608.812	567.774	528.831	534.397	625.565
7560	608.251	565.114	530.831	533.690	629.565
7590	606.367	561.038	527.095	534.741	634.957
7620	603.935	563.533	525.383	537.655	635.739
7650	598.243	555.333	521.463	534.534	627.217
7680	595.033	552.252	516.911	535.086	622.174
7710	591.558	548.891	508.663	532.690	624.043
7740	592.250	546.200	508.813	524.534	621.957
7770	590.167	548.633	507.641	532.155	632.826
7800	590.783	544.676	508.874	532.017	628.821
7830	592.967	547.181	507.653	532.276	621.217
7860	594.100	547.100	509.954	531.603	623.870
7890	592.467	548.105	509.840	531.552	628.957
7920	591.617	547.548	508.202	532.069	626.522
7950	598.817	545.105	508.613	531.224	618.391
7980	591.383	546.319	510.586	530.931	622.261
8010	588.833	546.243	507.592	527.121	632.043
8040	586.067	545.005	509.755	529.672	620.217
8070	592.200	547.081	508.969	528.155	633.217
8100	590.983	546.724	508.310	530.500	621.478
8130	590.583	548.319	509.107	531.017	625.522
8160	597.200	547.686	508.807	528.948	622.826
8190	591.817	548.971	507.006	529.534	626.783
8220	591.550	545.410	508.994	530.362	624.174
8250	590.667	547.190	507.160	532.138	630.130
8280	592.900	549.300	510.325	531.172	630.913
8310	589.700	547.471	508.494	530.414	635.087
8340	589.600	546.109	509.558	529.793	624.261
8370	588.233	547.010	509.365	522.155	624.391
8400	588.917	546.986	506.933	529.776	637.043
8430	590.183	548.057	510.117	533.138	627.739
8460	601.500	558.800	526.209	533.155	634.870
8490	614.070	563.638	540.825	532.759	636.565
8520	611.957	563.781	535.893	529.207	625.130
8550	609.143	568.200	535.718	532.828	636.783
8580	610.967	565.338	536.994	535.241	637.522
8610	607.317	564.048	533.432	533.310	628.000
8640	599.452	560.867	525.531	535.897	631.261
8670	593.250	545.533	515.184	526.845	622.043
8700	590.917	544.514	508.040	531.759	623.261
8730	595.267	546.905	508.798	532.810	631.217
8760	588.850	545.352	508.755	529.138	632.652
8790	588.200	546.600	508.969	526.759	631.478
8820	592.083	543.176	507.021	532.655	620.174
8850	594.600	545.410	508.150	530.086	629.696
8880	591.033	546.976	507.331	532.397	630.739
8910	598.350	546.495	509.221	534.207	621.217
8940	586.817	547.790	508.325	524.879	629.435
8970	592.083	543.881	507.660	533.310	630.826

9000	591.583	547.824	509.132	531.172	626.261
9030	594.117	544.533	509.791	535.483	625.391
9060	591.450	547.929	508.862	537.017	624.043
9090	591.483	546.376	507.798	536.517	637.435
9120	589.917	542.967	508.160	538.328	626.435
9150	591.583	547.057	509.285	535.672	627.391
9180	589.733	547.795	509.426	541.810	624.957
9210	587.850	545.067	507.764	540.052	628.609
9240	591.483	548.195	509.337	538.103	635.739
9270	592.883	545.143	509.647	539.948	630.826
9300	591.083	548.900	509.436	537.448	631.261
9330	586.067	547.671	506.834	539.241	630.304
9360	594.033	547.848	507.156	539.483	624.348
9390	592.250	546.576	508.687	543.397	625.261
9420	586.967	548.562	509.252	546.207	625.261
9450	596.033	555.762	518.156	546.172	629.957
9480	610.543	563.762	538.893	548.569	637.130
9510	609.945	564.571	538.301	541.155	623.913
9540	608.391	565.914	537.540	549.034	631.348
9570	607.716	566.933	534.172	544.638	630.652
9600	605.144	563.533	536.110	546.638	631.739
9630	601.235	561.829	528.951	546.293	630.870
9660	595.722	550.652	514.966	541.310	629.130
9690	591.354	545.905	507.500	538.897	630.348
9720	589.300	545.486	507.825	540.603	622.783
9750	588.167	544.833	508.911	541.586	619.000
9780	589.950	543.871	508.144	537.690	623.043
9810	592.817	545.905	508.997	535.414	628.174
9840	592.433	545.676	507.285	542.534	632.826
9870	593.933	545.771	509.267	541.293	626.043
9900	592.817	544.095	507.347	543.190	633.174
9930	592.133	545.924	505.865	537.759	633.000
9960	589.600	544.500	508.905	542.034	626.304
9990	592.667	545.738	508.138	540.586	626.478
10020	590.167	545.076	507.242	535.517	628.304
10050	592.283	545.595	510.383	540.828	630.130
10080	595.200	545.295	508.788	539.776	636.696
10110	589.600	548.152	509.442	535.483	636.739
10140	588.267	545.676	507.190	537.741	619.696
10170	590.933	547.348	507.408	539.466	630.826
10200	594.250	545.771	507.899	532.069	627.087
10230	592.250	545.609	507.745	537.741	624.391
10260	593.033	546.219	509.377	539.655	628.522
10290	587.300	544.433	506.561	535.362	629.870
10320	590.750	545.276	507.049	536.086	622.087
10350	589.033	543.300	507.644	536.276	619.739
10380	591.567	546.871	508.270	535.310	633.957
10410	591.383	545.738	508.463	537.310	634.870
10440	590.167	546.790	514.359	538.845	622.565
10470	602.133	562.109	530.978	541.397	632.174
10500	608.913	563.638	530.158	538.603	628.087
10530	609.703	561.486	530.242	541.190	630.391
10560	609.257	565.732	527.261	541.034	633.435
10590	606.290	561.681	529.068	542.224	629.783
10620	603.917	563.410	526.678	535.776	633.174
10650	598.510	553.705	520.791	540.259	636.391
10680	592.767	545.214	512.408	539.810	629.174
10710	594.400	545.443	508.095	533.017	623.783
10740	593.533	545.414	509.135	536.017	630.174
10770	593.117	546.438	507.169	531.707	633.348
10800	589.233	546.614	508.537	538.069	626.000
10830	595.067	546.867	508.641	533.241	624.783
10860	593.083	545.871	507.623	532.897	623.043
10890	593.683	546.086	507.761	534.466	629.696
10920	593.250	545.629	508.426	535.897	629.957
10950	589.067	544.833	508.604	537.966	630.826
10980	590.200	545.714	510.248	538.741	625.565
11010	587.900	545.686	508.095	536.621	630.174
11040	587.950	543.229	508.426	539.207	634.478
11070	592.333	547.767	506.043	536.966	622.478
11100	591.717	546.710	508.227	533.310	631.217
11130	594.167	549.443	509.880	531.586	624.652
11160	594.683	545.281	507.512	532.948	635.130
11190	588.883	546.438	508.316	534.379	631.261
11220	590.683	547.562	506.773	532.879	627.783
11250	592.200	546.414	507.350	532.362	625.130
11280	591.683	545.924	508.101	534.414	626.391
11310	587.650	546.314	505.788	530.276	625.913
11340	591.083	546.562	506.853	533.017	622.739
11370	592.450	543.619	509.316	534.259	627.783
11400	589.767	544.752	509.371	536.328	630.870
11430	591.867	544.876	507.359	532.948	627.870
11460	597.050	558.505	522.647	534.552	621.739
11490	608.162	562.900	529.537	541.586	628.130
11520	608.175	565.557	527.028	537.879	629.522
11550	604.900	560.938	528.911	539.690	637.696
11580	602.801	561.362	527.383	537.603	627.652
11610	600.853	562.033	528.199	539.448	624.652
11640	601.592	560.348	524.414	535.707	637.130
11670	593.442	543.824	514.209	536.345	636.870
11700	595.133	545.210	511.150	536.224	631.130
11730	589.017	547.695	508.509	537.138	629.957
11760	592.267	548.576	507.613	538.621	626.478
11790	591.117	545.690	508.850	537.552	630.000
11820	587.500	550.948	508.129	541.293	621.652
11850	589.267	545.862	508.457	544.345	627.304
11880	591.517	547.557	508.494	547.759	633.913
11910	594.867	548.495	508.475	541.362	624.348
11940	592.750	543.295	509.025	542.603	621.652
11970	592.550	548.305	509.515	542.672	623.435
12000	590.083	547.576	509.129	543.328	637.652

12030	594.800	547.238	506.994	548.155	627.565
12060	589.733	544.948	507.834	546.138	629.870
12090	591.750	545.948	508.791	548.276	629.826
12120	590.283	545.990	510.276	545.034	625.261
12150	594.083	545.329	507.353	546.052	619.870
12180	587.217	546.167	508.558	545.793	622.652
12210	598.067	546.310	509.917	542.759	623.957
12240	593.450	547.148	507.460	542.293	630.565
12270	593.717	544.752	508.156	546.897	632.870
12300	591.250	546.638	509.077	541.690	629.391
12330	587.767	547.176	507.899	543.328	624.609
12360	591.217	545.657	507.245	547.517	624.435
12390	590.167	542.810	508.807	544.207	623.174
12420	595.500	544.495	507.718	543.086	619.609
12450	591.250	552.548	515.718	548.241	632.261
12480	602.100	562.743	529.880	548.828	625.565
12510	608.773	564.329	533.899	538.293	634.304
12540	604.867	567.388	533.135	548.190	631.522
12570	603.667	564.467	531.896	546.793	626.609
12600	602.381	563.752	530.518	544.621	636.043
12630	598.727	563.962	525.518	542.241	628.174
12660	593.621	550.795	513.985	543.776	630.565
12690	592.258	547.048	507.426	547.379	633.348
12720	588.117	543.724	508.972	543.638	624.739
12750	583.625	545.543	508.678	541.276	623.696
12780	591.317	547.181	507.187	543.345	628.783
12810	590.933	548.219	506.497	540.362	625.391
12840	596.867	546.452	507.739	540.069	635.739
12870	590.233	545.386	509.571	540.552	624.000
12900	590.867	547.805	508.245	542.845	627.783
12930	588.583	545.814	507.745	542.879	624.609
12960	593.967	544.133	507.758	539.586	629.478
12990	589.433	545.781	505.798	544.414	627.957
13020	587.550	544.262	509.175	542.552	624.435
13050	597.167	545.986	508.525	541.241	633.087
13080	592.900	544.524	507.635	546.293	627.652
13110	592.533	547.824	508.150	543.345	620.522
13140	586.550	543.519	508.098	541.845	624.826
13170	592.200	544.790	507.285	538.362	627.435
13200	594.483	545.781	508.592	541.069	621.261
13230	593.717	547.781	507.868	542.466	626.957
13260	588.600	544.919	507.316	537.466	629.261
13290	592.217	544.871	509.166	538.379	630.913
13320	588.767	545.891	507.031	539.379	625.174
13350	592.283	546.590	505.423	539.138	627.000
13380	592.317	545.367	506.687	540.310	622.435
13410	589.267	544.643	508.043	537.155	623.304
13440	591.817	548.752	509.979	539.379	629.000
13470	604.100	562.362	531.844	546.034	625.435
13500	608.621	563.581	536.831	542.379	628.870
13530	607.691	564.124	538.279	539.776	627.000
13560	606.633	565.471	533.003	545.483	632.043
13590	605.317	562.557	534.098	538.259	630.609
13620	605.700	562.367	532.218	541.241	629.043
13650	602.333	555.210	521.908	537.241	625.217
13680	593.659	544.657	513.362	541.138	629.783
13710	592.350	544.652	508.356	537.276	629.087
13740	594.233	545.400	507.712	542.534	621.304
13770	593.650	544.567	506.724	536.862	628.739
13800	590.517	547.257	509.475	538.897	629.174
13830	589.633	544.443	509.166	536.241	627.087
13860	594.683	545.224	507.917	538.345	625.087
13890	587.733	545.510	509.080	538.293	625.261
13920	594.150	549.443	506.975	536.362	628.130
13950	593.367	543.681	506.491	540.948	631.609
13980	591.483	545.695	507.475	536.621	630.217
14010	588.150	544.895	507.706	537.500	627.043
14040	593.983	544.395	507.681	534.414	628.913
14070	593.600	545.162	508.819	535.828	627.826
14100	592.050	544.533	506.994	539.466	630.043
14130	589.550	544.871	508.387	534.293	632.087
14160	586.617	547.014	507.055	539.069	623.522
14190	597.833	546.238	507.390	534.793	624.913
14220	593.033	543.943	506.966	530.810	625.130
14250	592.383	544.114	506.997	540.828	620.696
14280	592.267	546.010	507.129	538.690	628.696
14310	589.933	545.048	507.761	537.500	618.435
14340	589.533	544.076	507.172	534.052	624.826
14370	594.067	545.971	508.436	538.431	625.174
14400	589.067	542.276	506.736	533.121	624.174
14430	596.783	544.024	509.135	539.690	624.522
14460	602.050	561.038	526.644	537.224	633.348
14490	611.486	563.667	535.316	537.448	625.000
14520	609.550	564.900	534.301	540.638	627.652
14550	606.950	562.290	534.917	536.552	629.565
14580	605.300	563.624	532.798	539.000	625.174
14610	606.443	563.038	531.699	537.397	633.522
14640	597.772	560.419	527.187	538.190	626.652
14670	593.550	546.181	513.975	535.362	630.000
14700	593.250	544.238	507.172	530.690	633.304
14730	594.033	544.929	508.356	533.345	628.652
14760	591.883	545.819	507.451	535.069	625.043
14790	593.250	545.529	506.828	536.638	627.348
14820	587.500	544.376	507.607	533.207	629.391
14850	593.233	542.914	508.675	536.810	620.043
14880	584.517	545.371	506.117	531.810	636.565
14910	593.700	544.300	507.699	533.241	626.087
14940	596.283	543.595	509.025	530.310	636.000
14970	594.650	544.190	508.390	534.707	633.174
15000	587.350	546.586	506.515	532.845	625.087
15030	590.550	547.262	506.227	534.569	626.870

15060	589.500	544.510	506.629	529.931	633.783
15090	591.667	546.386	508.755	528.466	622.130
15120	592.700	548.214	507.046	535.259	624.913
15150	592.200	547.029	507.224	538.448	630.000
15180	597.483	544.681	508.641	532.293	625.478
15210	589.717	544.171	507.773	538.466	628.348
15240	587.517	546.057	506.055	535.828	626.087
15270	589.683	546.095	507.767	529.448	624.696
15300	591.533	545.114	507.571	529.517	620.391
15330	588.667	542.938	507.107	529.690	631.478
15360	591.950	547.376	508.571	533.862	626.870
15390	593.417	545.871	508.138	533.086	628.652
15420	592.117	544.976	509.202	528.310	624.261
15450	594.533	552.629	519.537	532.724	628.696
15480	605.017	562.248	540.074	533.879	622.565
15510	608.952	567.129	538.006	536.052	629.000
15540	607.538	565.529	539.752	535.897	623.348
15570	605.233	563.400	539.150	536.621	634.696
15600	604.177	563.138	537.868	536.448	627.913
15630	602.381	562.605	532.583	535.052	628.391
15660	600.267	544.814	517.267	528.000	623.565
15690	595.500	544.795	511.187	527.897	630.826
15720	594.617	547.400	508.856	535.517	622.652
15750	589.417	545.471	508.034	528.362	625.609
15780	591.850	544.552	506.948	531.293	630.739
15810	590.800	543.576	509.221	528.103	633.348
15840	594.317	544.000	509.699	530.931	624.652
15870	591.300	545.976	508.052	531.897	624.522
15900	593.933	543.410	507.494	530.759	632.652
15930	588.983	544.095	507.862	527.586	625.087
15960	588.150	543.895	507.356	528.983	625.087
15990	591.500	544.729	508.417	532.155	622.043
16020	594.617	545.438	506.675	528.776	625.261
16050	594.567	544.200	508.153	535.345	628.826
16080	592.767	546.071	507.175	530.621	629.217
16110	592.017	546.986	506.310	528.431	620.217
16140	589.933	546.238	504.632	529.448	625.435
16170	591.167	544.276	508.604	527.483	629.478
16200	591.483	545.157	508.571	527.034	625.304
16230	592.283	546.790	507.853	529.810	625.913
16260	596.233	544.614	508.012	529.172	624.087
16290	593.133	544.314	506.745	527.655	628.957
16320	584.733	542.895	507.439	527.103	631.609
16350	593.983	544.562	507.807	531.621	623.435
16380	590.374	545.905	509.669	527.862	621.087
16410	589.037	543.767	507.908	529.431	625.565
16440	595.750	548.224	511.101	526.190	630.087
16470	605.733	563.362	535.071	531.397	634.043
16500	610.047	567.557	539.794	534.534	626.783
16530	607.933	563.014	538.052	534.655	634.478
16560	607.600	564.667	537.003	531.276	629.087
16590	605.221	565.152	535.841	534.276	625.565
16620	605.700	563.962	530.037	532.500	631.217
16650	598.317	556.633	522.301	536.276	624.478
16680	594.028	545.810	513.006	527.103	629.261
16710	591.150	546.552	508.344	526.155	635.130
16740	592.850	545.833	508.273	528.655	635.435
16770	591.100	547.529	508.141	523.793	625.522
16800	594.833	543.600	505.586	525.966	635.391
16830	591.333	546.752	507.423	528.672	634.217
16860	592.033	547.033	506.782	525.362	628.913
16890	592.833	543.710	507.632	523.431	622.217
16920	594.217	547.081	509.163	524.776	626.043
16950	594.633	547.757	506.647	526.328	627.739
16980	591.867	545.186	508.788	526.741	629.130
17010	592.533	544.919	506.681	526.845	629.348
17040	590.033	544.738	509.147	525.914	621.957
17070	589.717	546.829	507.696	529.552	629.783
17100	591.683	546.095	508.583	526.586	630.435
17130	596.567	544.781	508.233	528.190	621.696
17160	591.233	546.391	506.509	525.741	625.348
17190	590.250	546.967	508.089	528.293	632.478
17220	590.450	542.271	507.709	530.448	628.609
17250	592.300	546.071	507.485	525.241	637.261
17280	590.700	545.505	508.880	527.638	626.000
17310	589.767	545.505	507.997	530.793	626.565
17340	590.467	545.824	507.620	522.707	627.391
17370	589.483	544.619	508.233	524.241	629.435
17400	591.267	547.181	507.580	526.672	635.130
17430	587.433	545.190	509.985	528.517	626.826
17460	603.200	559.981	529.690	531.207	636.739
17490	610.658	566.662	541.736	530.241	626.609
17520	614.134	562.471	536.767	529.466	631.391
17550	613.523	565.967	538.334	531.707	624.217
17580	608.467	568.824	539.948	529.172	632.783
17610	607.462	566.648	536.641	536.000	631.522
17640	606.239	563.167	531.794	534.948	636.609
17670	597.150	546.576	514.482	529.017	636.435
17700	594.588	544.738	508.598	529.655	636.565
17730	593.833	546.181	508.865	528.362	629.652
17760	592.083	547.533	509.460	521.966	625.174
17790	590.983	542.876	507.755	521.862	631.174
17820	592.217	546.538	508.755	533.776	626.696
17850	596.183	545.810	507.914	521.621	626.609
17880	590.000	546.109	507.491	527.862	622.174
17910	593.000	544.619	507.724	520.155	626.435
17940	595.800	544.819	508.233	527.414	627.304
17970	595.650	544.186	508.969	525.414	633.000
18000	589.883	545.957	508.337	524.569	623.609
18030	591.733	546.486	508.242	523.534	631.304
18060	588.533	545.724	508.000	529.741	623.304

18090	587.817	545.095	509.939	522.483	623.043
18120	594.450	545.824	507.997	523.966	622.391
18150	585.783	543.471	507.979	521.328	627.565
18180	593.700	547.286	508.270	526.086	630.609
18210	591.483	545.524	508.712	522.293	631.087
18240	589.767	544.657	506.460	528.000	622.913
18270	590.300	546.052	506.012	524.466	630.043
18300	594.950	544.967	506.730	521.914	624.130
18330	592.450	545.105	509.304	520.121	628.696
18360	591.583	545.490	506.129	522.138	630.130
18390	595.100	544.895	509.537	522.155	635.261
18420	591.233	545.838	507.411	526.086	634.087
18450	595.750	557.033	523.196	523.069	626.913
18480	609.617	568.000	538.083	530.138	637.913
18510	618.795	563.805	539.362	528.948	626.217
18540	615.038	568.090	540.390	526.155	636.957
18570	613.625	565.357	538.702	526.328	629.783
18600	611.473	564.157	535.736	528.138	629.174
18630	607.367	565.238	529.236	530.862	633.348
18660	598.367	547.729	514.175	527.207	629.957
18690	593.582	547.057	509.380	520.966	625.826
18720	591.787	543.795	507.181	522.121	623.739
18750	590.383	546.614	507.451	521.241	631.696
18780	595.633	545.019	507.037	523.828	629.043
18810	588.209	549.181	511.021	523.103	630.739
18840	591.000	541.690	507.739	524.345	623.043
18870	591.617	544.214	506.871	522.569	621.217
18900	590.717	545.129	506.825	522.862	621.304
18930	594.817	546.738	506.865	526.621	621.957
18960	594.500	543.857	507.463	521.983	627.696
18990	594.200	545.005	507.426	524.086	632.043
19020	590.433	544.186	507.012	522.966	624.739
19050	594.333	544.614	507.997	518.534	629.783
19080	595.233	543.705	508.196	522.707	627.435
19110	588.183	544.776	508.985	520.052	632.696
19140	590.433	547.200	510.107	522.293	627.348
19170	591.150	544.181	508.233	522.776	634.696
19200	593.283	544.662	507.985	521.086	623.478
19230	593.850	543.338	507.801	519.828	625.957
19260	590.733	544.376	508.555	517.466	632.130
19290	588.767	544.786	507.706	522.190	626.217
19320	595.833	545.005	508.110	522.655	628.391
19350	592.417	545.424	509.724	521.241	622.174
19380	596.483	544.757	505.926	523.190	626.000
19410	589.483	547.119	506.678	520.672	625.913
19440	591.850	543.805	506.558	522.362	621.522
19470	592.783	545.043	508.107	522.172	632.522
19500	588.533	544.119	508.147	517.862	629.174
19530	592.883	548.148	508.138	522.828	629.739
19560	591.633	547.633	507.877	517.534	623.652
19590	590.683	546.952	506.693	518.966	620.957
19620	592.783	543.457	507.991	521.931	634.261
19650	590.533	546.190	509.015	521.259	626.652
19680	586.717	546.543	510.101	521.948	628.087
19710	589.783	545.643	509.549	524.569	623.826
19740	590.400	545.733	507.469	522.862	629.130
19770	591.950	549.476	507.571	527.293	618.348
19800	587.117	545.824	508.074	526.190	623.957
19830	590.900	544.100	509.423	521.793	623.913
19860	591.550	544.895	508.653	518.310	623.435
19890	593.067	545.190	506.601	522.276	620.609
19920	593.067	543.119	508.767	521.603	618.478
19950	588.467	547.257	507.770	516.638	630.652
19980	592.167	545.100	506.905	519.483	622.826
20010	589.467	545.976	507.982	522.534	631.087
20040	591.817	545.771	507.166	518.138	622.087
20070	591.400	543.562	509.000	513.914	627.435
20100	596.217	547.314	506.887	520.879	623.000
20130	588.350	548.686	507.353	524.138	633.522
20160	593.733	545.286	507.693	524.621	621.043
20190	591.883	544.276	507.788	519.793	619.435
20220	589.083	543.633	507.212	515.897	628.696
20250	591.117	544.924	506.494	514.914	627.652
20280	590.533	546.381	509.647	518.845	626.783
20310	594.900	547.410	508.380	522.138	631.304
20340	596.417	545.262	507.000	524.017	620.826
20370	588.800	542.343	509.215	525.776	624.957
20400	592.917	544.181	507.353	520.241	629.826
20430	588.833	546.148	505.755	515.259	631.435
20460	589.300	546.652	508.138	520.362	635.870
20490	593.400	542.462	508.104	521.690	634.783
20520	586.067	546.886	508.693	519.345	625.696
20550	584.317	544.367	505.874	520.379	627.739
20580	590.383	545.905	506.975	522.017	622.174
20610	592.533	548.043	506.472	516.345	621.826
20640	589.533	546.119	506.442	518.414	628.435
20670	590.967	543.109	506.853	519.828	628.130
20700	595.367	545.681	505.184	519.121	627.826
20730	592.800	545.810	507.988	518.052	623.087
20760	592.250	546.600	506.856	518.121	631.913
20790	590.167	543.848	506.979	518.776	620.565
20820	592.000	544.571	507.583	521.138	629.087
20850	589.367	545.690	507.460	518.276	629.261
20880	587.250	544.995	507.340	520.414	628.739
20910	592.900	547.243	509.331	518.948	627.000
20940	589.433	541.676	508.472	521.103	633.130
20970	586.617	545.490	506.975	523.345	620.000
21000	591.117	546.229	507.402	516.414	635.957
21030	589.867	547.648	506.859	516.828	626.652
21060	586.950	543.138	507.951	518.948	636.304
21090	587.567	545.519	507.169	520.466	626.783

21120	589.717	543.714	507.709	523.310	629.217
21150	588.150	544.662	506.862	513.931	626.087
21180	591.700	542.586	508.215	524.328	635.000
21210	590.050	546.924	508.120	518.879	635.609
21240	587.633	544.624	506.816	519.293	622.087
21270	591.417	547.271	506.586	519.966	624.000
21300	591.050	545.567	508.402	522.586	633.435
21330	595.450	544.043	506.816	519.224	622.522
21360	591.367	546.029	506.718	518.138	624.870
21390	589.600	546.343	507.525	517.690	633.391
21420	589.650	544.195	508.276	518.397	630.870
21450	587.833	543.681	506.672	517.914	625.174
21480	586.350	543.514	506.491	515.931	624.391
21510	595.783	542.505	505.699	520.776	630.391
21540	592.850	544.562	507.282	521.966	620.000
21570	589.500	544.790	507.095	516.897	624.739
21600	594.367	545.310	508.092	516.121	625.087
21630	588.517	545.495	507.620	517.690	622.609
21660	593.150	543.314	507.647	517.966	632.957
21690	594.000	545.881	507.699	516.241	631.783
21720	591.200	545.714	507.175	515.914	630.217
21750	590.300	547.490	507.015	513.672	635.913
21780	586.567	542.205	507.859	517.000	622.217
21810	593.050	545.524	508.058	515.069	628.391
21840	590.033	544.557	506.356	515.621	627.391
21870	588.417	546.471	509.058	517.086	629.304
21900	586.433	542.538	506.595	516.155	631.217
21930	591.950	543.871	506.859	516.966	628.826
21960	588.583	543.819	505.920	516.810	624.913
21990	590.750	545.043	508.166	513.517	625.783
22020	594.167	547.162	506.261	517.534	622.783
22050	590.400	544.205	507.040	518.293	629.130
22080	593.800	543.376	507.110	513.483	624.870
22110	588.433	544.262	507.018	515.138	627.870
22140	595.033	545.219	505.813	519.741	632.348
22170	591.250	545.805	506.261	515.000	632.696
22200	590.583	545.495	508.193	515.672	634.435
22230	596.700	546.629	506.574	520.276	629.957
22260	589.883	544.795	506.242	516.362	624.696
22290	586.417	543.614	505.969	515.345	628.826
22320	593.517	546.148	508.531	517.379	626.870
22350	587.317	543.805	506.834	517.724	628.522
22380	589.983	543.800	507.798	508.845	626.870
22410	591.983	544.819	507.037	515.534	632.826
22440	589.800	544.290	506.696	517.155	624.652
22470	589.900	547.981	507.837	518.500	625.609
22500	589.583	543.905	507.313	513.983	627.826
22530	590.317	543.595	508.074	515.638	629.261
22560	587.833	544.267	507.368	515.603	626.826
22590	588.500	545.571	508.288	513.517	632.435
22620	592.683	543.681	505.782	513.276	629.957
22650	585.283	544.795	506.368	513.224	624.957
22680	586.100	543.700	507.847	512.862	628.087
22710	592.300	545.576	506.350	516.897	632.870
22740	587.000	547.862	508.209	514.690	630.391
22770	587.350	543.762	506.644	518.362	632.739
22800	586.767	542.714	507.469	517.207	630.783
22830	591.300	544.948	507.856	514.621	632.478
22860	590.933	544.014	508.862	513.552	633.217
22890	591.667	543.876	506.687	516.086	631.652
22920	590.017	546.495	506.767	511.828	624.000
22950	591.517	543.238	505.788	512.345	620.522
22980	589.783	543.638	506.936	517.155	625.000
23010	585.783	542.933	507.224	517.052	628.739
23040	586.883	544.095	506.730	520.500	632.000
23070	587.717	544.595	506.583	513.759	627.870
23100	588.067	545.295	506.819	514.828	628.913
23130	587.133	544.757	508.678	512.931	627.217
23160	588.283	543.881	509.028	515.397	625.478
23190	591.300	545.090	506.328	515.741	630.087
23220	587.583	542.500	506.086	515.052	622.652
23250	588.033	544.310	507.595	511.690	633.870
23280	588.567	545.019	507.742	513.121	627.957
23310	587.567	545.371	506.417	517.414	626.174
23340	588.100	545.433	505.436	511.103	626.348
23370	589.200	548.395	508.669	512.500	634.565
23400	592.533	542.686	506.399	515.138	626.217
23430	592.217	548.019	507.147	514.948	625.565
23460	588.633	545.062	505.788	513.741	627.783
23490	592.350	541.810	508.061	515.672	630.304
23520	588.383	543.667	508.528	515.293	619.435
23550	587.733	544.257	505.564	510.569	625.565
23580	587.850	544.405	506.172	516.397	634.652
23610	589.667	543.186	507.844	514.655	630.913
23640	595.617	544.943	506.561	513.914	638.435
23670	590.967	545.310	505.853	516.707	630.696
23700	595.367	544.176	505.592	514.862	632.043
23730	593.600	541.010	507.233	517.138	624.348
23760	590.817	544.548	509.390	515.034	631.000
23790	588.800	546.629	506.123	515.983	627.826
23820	591.683	545.671	505.994	517.052	631.217
23850	590.567	545.590	506.156	514.638	634.435
23880	590.883	544.271	508.709	515.207	625.043
23910	584.067	544.219	507.199	515.672	631.000
23940	594.683	547.057	507.356	514.172	627.739
23970	593.950	545.190	506.733	514.724	621.490
24000	592.167	544.990	506.221	514.121	631.000
24030	588.000	545.248	507.779	514.414	623.000
24060	593.317	546.024	507.362	515.483	628.826
24090	589.667	544.062	505.531	515.345	627.565
24120	590.683	543.324	506.095	515.810	624.957

24150	590.117	545.891	508.083	516.052	633.348
24180	589.733	545.776	506.107	515.500	634.783
24210	585.450	546.119	506.426	506.517	624.217
24240	585.250	545.386	504.583	513.414	629.261
24270	588.883	547.238	507.445	515.483	635.739
24300	589.867	543.833	507.613	516.879	626.217
24330	584.000	543.962	507.604	514.586	633.261
24360	592.350	544.324	507.420	516.741	622.609
24390	591.000	544.810	509.132	515.759	634.696
24420	589.150	542.043	505.549	515.603	623.043
24450	592.833	543.552	505.405	512.500	632.609
24480	588.033	546.929	506.702	515.069	630.000
24510	596.900	543.857	506.150	512.690	620.913
24540	592.350	546.514	506.975	515.466	624.261
24570	590.300	545.367	506.206	517.328	628.435
24600	588.783	545.133	507.058	515.052	626.696
24630	592.167	544.048	506.893	514.379	631.174
24660	590.033	545.786	505.506	513.879	631.609
24690	588.783	543.419	506.755	509.466	634.087
24720	591.550	544.743	506.380	512.707	624.261
24750	588.017	543.609	508.853	512.345	626.870
24780	591.483	546.248	507.279	513.948	625.304
24810	591.800	546.410	506.353	516.707	626.478
24840	589.333	545.152	509.058	514.069	624.130
24870	592.883	546.200	509.353	513.207	635.000
24900	586.517	546.195	505.794	515.741	626.609
24930	587.900	544.543	508.371	513.224	633.391
24960	587.650	544.219	506.282	512.034	626.957
24990	592.867	544.024	508.402	514.172	627.870
25020	591.650	544.324	506.021	516.793	627.913
25050	586.033	544.690	507.393	514.586	624.652
25080	588.217	543.843	506.396	511.690	629.522
25110	592.733	544.448	508.291	513.948	625.870
25140	589.883	546.376	506.325	519.241	633.391
25170	592.450	543.105	506.601	513.672	634.130
25200	592.867	543.733	507.261	517.517	638.783
25230	593.950	545.248	505.727	512.638	623.652
25260	583.800	543.029	505.589	514.586	633.391
25290	586.733	546.095	507.104	513.707	622.783
25320	587.583	543.081	507.052	515.517	638.000
25350	586.267	545.014	507.567	510.741	633.391
25380	586.700	542.300	506.408	519.793	629.391
25410	591.817	544.857	505.979	513.397	632.391
25440	586.650	542.133	507.064	514.879	625.913
25470	592.433	543.829	507.779	516.207	634.696
25500	584.767	544.824	506.758	511.724	626.130
25530	589.850	545.233	505.399	513.017	626.696
25560	592.617	541.943	508.871	511.638	623.957
25590	590.417	543.838	507.718	511.741	631.826
25620	586.917	546.300	507.393	515.759	628.261
25650	590.283	544.505	507.166	513.310	630.565
25680	587.100	548.219	505.558	517.810	626.826
25710	595.050	543.362	506.433	513.310	623.652
25740	586.383	544.586	508.402	513.707	624.696
25770	587.617	544.433	508.025	516.224	622.522
25800	586.883	544.005	508.880	518.172	626.261
25830	593.333	545.314	506.160	521.483	626.304
25860	594.950	546.976	506.156	517.155	626.000
25890	591.600	543.914	507.227	520.103	631.261
25920	588.850	545.952	505.564	518.724	624.565
25950	583.717	545.500	506.899	517.931	627.304
25980	591.150	545.205	507.463	519.534	626.522
26010	587.417	544.005	507.061	519.586	625.478
26040	588.933	545.762	506.945	521.241	620.435
26070	585.383	547.129	506.466	528.138	626.348
26100	589.200	545.162	504.294	527.741	621.217
26130	589.767	548.376	507.347	529.397	625.435
26160	589.667	548.181	507.506	527.517	626.696
26190	598.767	544.929	508.025	531.207	624.957
26220	590.517	545.609	506.144	535.414	626.522
26250	593.500	545.214	508.365	532.569	632.826
26280	591.033	545.138	506.923	535.155	626.435
26310	590.050	547.200	507.012	538.259	626.217
26340	590.733	541.362	505.426	532.121	634.130
26370	591.133	546.843	507.564	527.655	633.913
26400	592.333	546.781	505.549	535.103	624.304
26430	588.850	546.481	505.865	533.879	625.652
26460	591.700	545.705	508.957	532.483	623.870
26490	590.250	546.186	508.736	529.914	634.130
26520	588.167	545.438	507.337	532.155	630.000
26550	588.583	544.448	507.344	537.121	632.739
26580	587.983	545.681	508.684	531.517	634.000
26610	590.967	547.395	507.230	537.672	623.739
26640	588.050	545.352	506.877	537.259	633.913
26670	595.300	544.391	506.684	532.862	630.652
26700	586.283	545.148	505.997	533.207	623.217
26730	587.350	544.500	507.058	535.052	630.435
26760	588.400	541.471	507.294	532.121	628.609
26790	590.417	545.733	506.482	536.966	633.217
26820	592.433	545.714	507.215	534.293	632.043
26850	593.783	543.348	505.702	531.224	628.391
26880	586.333	546.910	505.856	534.845	629.565
26910	589.450	546.057	505.890	531.724	622.478
26940	592.833	543.248	506.644	527.914	630.870
26970	587.617	544.852	507.132	533.983	634.087
27000	586.283	543.833	507.156	531.448	632.217
27030	588.117	544.738	506.236	535.431	633.870
27060	589.017	544.967	506.528	530.517	622.652
27090	588.233	545.095	507.966	533.776	627.783
27120	592.317	544.805	506.957	533.397	621.826
27150	593.467	546.252	506.169	530.621	630.913

27180	591.533	545.529	506.135	528.103	621.522
27210	588.017	542.600	506.592	532.517	631.043
27240	588.650	544.262	507.310	530.724	635.304
27270	590.067	547.295	507.874	530.534	633.391
27300	586.833	546.167	506.663	534.897	625.870
27330	594.417	544.643	508.301	529.931	625.565
27360	590.300	543.771	507.117	533.500	620.435
27390	584.917	545.443	505.834	529.172	629.087
27420	585.733	543.757	505.890	528.897	636.435
27450	592.400	545.219	506.264	532.293	633.957
27480	587.717	545.348	504.902	526.000	634.217
27510	595.800	545.448	508.258	526.517	628.565
27540	591.167	544.029	506.371	529.966	633.261
27570	590.417	545.900	506.074	532.379	632.826
27600	589.783	547.114	505.423	529.138	628.043
27630	586.717	547.567	506.644	530.345	626.348
27660	590.783	543.710	506.742	532.483	635.739
27690	587.983	546.938	507.086	530.603	623.826
27720	588.033	546.133	507.414	530.759	629.739
27750	588.717	545.529	508.120	531.259	635.870
27780	586.417	546.205	507.555	530.638	629.696
27810	593.000	545.033	506.979	531.897	627.348
27840	589.267	547.010	506.847	531.552	628.522
27870	596.067	543.767	505.451	524.724	633.435
27900	584.867	546.109	507.702	529.948	639.522
27930	584.867	546.619	505.463	529.672	627.043
27960	590.683	542.552	506.224	530.172	628.870
27990	589.417	546.571	507.454	529.983	625.435
28020	585.850	545.871	507.825	531.672	629.565
28050	587.767	543.971	507.267	533.448	622.696
28080	592.717	546.686	508.135	528.466	630.130
28110	593.600	545.095	507.736	531.328	636.826
28140	588.850	546.924	507.399	529.155	620.043
28170	586.183	543.567	505.785	532.345	635.478
28200	585.900	544.981	506.696	527.534	624.870
28230	588.467	543.438	508.377	527.707	625.000
28260	587.600	546.790	507.012	528.259	629.174
28290	587.883	547.305	507.408	529.776	633.783
28320	588.750	543.238	507.752	531.586	637.739
28350	589.233	545.281	506.770	531.000	621.043
28380	590.217	545.386	508.457	533.707	619.000
28410	591.883	543.924	508.426	529.293	626.043
28440	586.433	544.881	508.804	527.190	621.000
28470	584.450	545.257	508.801	525.638	621.783
28500	593.500	543.776	507.031	528.793	630.739
28530	591.233	544.438	507.810	532.241	621.565
28560	583.983	543.376	505.813	528.052	630.043
28590	591.867	544.729	506.804	526.155	632.087
28620	584.133	542.543	506.607	529.552	627.870
28650	587.733	544.891	508.485	524.569	620.391
28680	587.167	545.171	506.724	524.948	631.174
28710	588.283	547.629	507.656	524.466	629.478
28740	588.400	544.581	508.509	530.931	628.304
28770	589.883	545.257	506.525	530.310	623.435
28800	587.017	544.086	507.420	527.603	626.565
28830	587.683	544.395	507.828	530.086	635.391
28860	591.183	544.771	506.472	530.276	622.130
28890	591.800	544.133	507.518	529.931	621.261
28920	586.517	545.867	505.546	528.448	623.652
28950	583.750	544.395	507.117	526.569	628.435
28980	591.533	545.571	505.908	527.448	630.913
29010	586.717	544.609	505.828	531.638	629.696
29040	589.717	546.900	506.313	522.241	628.739
29070	587.933	543.233	506.506	528.121	621.174
29100	584.517	544.214	508.028	525.672	627.304
29130	583.683	544.143	507.086	525.793	628.522
29160	584.783	547.914	508.190	525.103	628.087
29190	593.517	543.300	507.580	527.190	624.174
29220	586.817	548.514	507.969	524.172	624.000
29250	587.033	546.995	506.620	528.345	620.217
29280	588.083	544.748	508.169	526.121	619.435
29310	588.217	543.414	507.377	527.414	629.957
29340	587.083	544.495	505.902	525.017	635.217
29370	588.967	544.895	505.595	527.931	625.957
29400	589.517	542.862	505.816	526.362	624.478
29430	585.700	544.510	507.663	526.690	630.261
29460	593.733	543.514	506.979	530.724	631.304
29490	587.017	542.510	506.948	529.828	619.696
29520	590.033	544.405	506.595	526.362	624.565
29550	586.333	545.029	507.518	527.069	628.870
29580	582.250	544.490	508.739	525.655	622.522

FIGURE 2p

Time (ms)	Neuron 1	Neuron 2	Neuron 3	Neuron 4
0	91.801	75.609	79.975	69.905
40	92.451	72.588	78.350	67.813
80	93.015	73.426	79.186	68.475
120	91.132	74.669	77.672	69.271
160	93.019	74.275	78.325	66.923
200	92.224	73.173	80.355	67.930
240	94.177	75.542	78.863	68.081
280	93.671	73.975	78.169	68.408
320	93.801	72.507	78.762	68.842
360	93.976	74.039	80.057	68.285
400	91.445	74.500	78.279	69.151
440	93.254	73.261	78.046	69.123
480	92.555	73.412	79.224	65.965
520	93.432	73.859	79.574	68.736
560	92.667	74.863	77.656	69.454
600	92.013	74.535	78.066	67.599
640	91.686	73.634	78.956	67.264
680	91.927	73.190	77.872	68.609
720	93.523	74.299	76.117	68.356
760	92.656	75.496	78.325	67.944
800	92.023	75.102	78.235	69.151
840	103.496	74.359	75.833	72.553
880	120.414	74.785	81.612	80.905
920	110.716	85.880	106.921	78.549
960	102.239	94.704	111.172	75.673
1000	95.171	102.074	107.383	71.275
1040	92.835	102.074	106.811	69.165
1080	91.784	96.687	103.732	69.761
1120	89.669	97.250	99.076	68.292
1160	92.246	95.028	98.137	68.370
1200	89.239	89.359	93.546	66.243
1240	92.714	86.775	89.601	68.370
1280	90.440	87.208	90.301	67.264
1320	91.186	85.553	87.546	67.754
1360	100.994	82.923	90.516	70.757
1400	114.729	89.817	105.970	78.556
1440	107.915	97.313	111.708	75.891
1480	98.141	102.261	109.052	72.771
1520	94.805	97.835	104.260	69.120
1560	92.461	95.261	99.765	68.856
1600	91.855	91.602	95.464	66.736
1640	95.318	90.789	92.377	69.123
1680	95.957	85.982	91.555	67.465
1720	93.479	84.313	90.115	69.750

1760	93.077	82.335	88.213	68.415
1800	91.280	80.377	88.033	68.232
1840	90.665	80.211	88.978	67.923
1880	102.367	80.225	87.074	70.437
1920	115.987	80.356	85.443	78.746
1960	107.327	76.507	85.631	75.986
2000	97.459	78.415	84.186	71.592
2040	93.789	76.539	84.899	69.070
2080	91.665	75.961	87.948	68.025
2120	91.789	76.680	84.566	69.261
2160	91.171	76.496	83.749	68.229
2200	109.641	74.349	81.893	72.208
2240	106.226	76.320	83.735	74.641
2280	99.064	76.384	84.801	72.982
2320	94.427	74.842	81.210	70.278
2360	92.096	74.137	82.899	67.113
2400	90.944	74.042	80.473	67.944
2440	92.021	73.408	80.617	68.574
2480	95.434	75.155	80.533	68.011
2520	113.752	74.754	79.175	72.201
2560	105.295	74.989	83.590	72.806
2600	95.340	75.859	83.697	71.521
2640	91.393	75.271	82.525	69.053
2680	91.848	75.468	81.344	66.866
2720	89.739	74.511	82.287	68.011
2760	91.338	73.602	78.694	66.651
2800	93.336	73.803	80.552	66.711
2840	94.263	72.623	83.167	67.827
2880	95.378	74.690	82.516	69.229
2920	92.145	73.567	79.068	64.208
2960	90.957	72.838	81.902	68.352
3000	90.382	73.662	78.052	66.173
3040	90.120	73.500	79.828	65.870
3080	91.053	71.623	76.798	67.173
3120	92.092	73.134	79.563	66.342
3160	92.026	72.595	80.992	67.327
3200	91.808	75.356	79.440	66.725
3240	91.252	71.782	79.596	67.141
3280	89.759	74.014	80.350	67.715
3320	90.282	73.236	79.320	65.898
3360	91.511	73.335	78.754	68.130
3400	91.509	73.342	77.322	66.197
3440	93.648	73.415	77.167	67.919
3480	91.867	74.511	78.475	67.806
3520	94.750	72.134	81.331	68.303
3560	91.774	75.116	81.683	68.715
3600	90.607	73.711	79.727	67.444

3640	90.081	72.683	79.680	68.092
3680	90.921	72.894	80.372	66.496
3720	91.137	73.246	77.156	67.560
3760	92.438	74.331	78.350	67.923
3800	91.098	73.264	78.153	66.204
3840	93.075	72.792	76.041	67.113
3880	90.265	73.993	77.809	66.570
3920	92.391	75.356	76.760	67.077
3960	91.160	74.197	76.975	67.595
4000	91.201	74.032	76.844	68.162
4040	91.034	72.504	76.590	67.511
4080	91.100	73.697	76.623	66.521
4120	92.192	73.208	76.801	66.113
4160	90.062	73.933	76.620	67.158
4200	92.380	72.398	77.260	67.856
4240	92.254	73.081	76.268	68.088
4280	92.692	71.644	75.926	68.067
4320	92.353	72.627	77.393	67.789
4360	92.284	72.148	76.019	66.919
4400	89.115	71.011	75.180	68.292
4440	90.445	72.060	77.350	66.676
4480	90.395	74.042	74.773	67.701
4520	92.793	73.239	77.339	65.493
4560	90.898	73.053	77.317	67.028
4600	92.780	73.796	75.995	67.620
4640	93.383	71.736	77.497	66.264
4680	93.303	72.338	76.732	67.208
4720	91.835	74.359	77.456	67.359
4760	90.902	73.380	76.317	67.444
4800	91.205	73.937	75.153	66.144
4840	91.754	72.894	77.612	67.320
4880	90.996	74.623	75.967	68.342
4920	91.519	72.648	77.760	68.539
4960	93.626	72.120	75.123	67.616
5000	91.064	73.454	76.033	66.592
5040	89.898	72.732	77.549	68.246
5080	90.829	71.352	75.541	67.222
5120	91.810	73.708	76.194	67.761
5160	91.028	72.782	75.732	67.218
5200	90.429	73.514	76.142	67.243
5240	91.720	72.563	76.607	67.356
5280	90.592	73.648	75.347	66.799
5320	91.111	72.151	76.940	67.613
5360	91.130	71.810	76.607	66.067
5400	92.528	71.870	77.557	67.894
5440	91.699	70.908	76.869	68.701
5480	91.675	71.563	76.180	67.313

5520	90.774	72.704	76.536	68.169
5560	92.622	71.574	77.082	66.648
5600	91.438	73.313	76.022	68.106
5640	90.133	71.317	77.123	67.419
5680	93.265	72.683	76.486	67.345
5720	90.714	73.634	77.850	68.514
5760	91.002	71.327	77.459	66.725
5800	91.316	74.229	76.809	66.729
5840	91.662	71.444	75.656	66.514
5880	91.201	74.046	75.645	67.944
5920	91.549	73.218	76.352	67.049
5960	92.092	74.035	74.546	66.014
6000	91.605	74.493	76.027	66.074
6040	92.066	72.394	75.328	66.405
6080	90.523	73.778	75.344	66.025
6120	92.308	74.081	76.648	67.658
6160	92.088	72.518	77.426	64.648
6200	89.844	72.201	76.306	67.648
6240	90.581	72.870	75.169	66.718
6280	91.128	72.134	76.770	68.359
6320	93.500	73.278	75.459	66.736
6360	91.445	73.426	76.705	66.683
6400	90.105	74.236	74.915	66.979
6440	92.692	74.560	77.260	67.701
6480	91.338	72.982	75.790	66.232
6520	92.073	71.514	76.243	67.292
6560	90.799	72.806	78.161	66.810
6600	91.806	73.352	77.117	65.525
6640	91.882	72.144	74.620	67.489
6680	91.011	74.183	74.743	67.792
6720	89.692	74.437	76.011	66.915
6760	90.697	72.856	76.514	65.444
6800	91.141	73.109	75.178	66.736
6840	91.829	71.592	76.369	67.141
6880	107.883	73.743	77.503	69.377
6920	117.325	75.042	81.016	80.472
6960	105.630	84.764	103.697	76.183
7000	97.776	94.345	103.281	71.482
7040	93.408	98.894	104.107	69.384
7080	90.562	100.067	102.686	69.342
7120	89.137	98.824	98.087	67.602
7160	86.964	93.919	93.219	67.165
7200	90.182	91.560	92.292	66.468
7240	90.028	90.637	89.538	66.113
7280	88.306	87.113	88.620	68.761
7320	91.660	84.965	89.667	68.246
7360	99.060	92.665	106.781	68.539

7400	115.897	101.275	107.101	77.919
7440	107.615	104.004	106.560	73.423
7480	98.940	101.504	100.904	72.687
7520	93.451	96.905	98.568	68.359
7560	89.622	95.514	94.631	69.658
7600	94.387	91.827	99.104	68.173
7640	112.752	98.532	113.686	73.715
7680	104.930	103.549	112.544	72.021
7720	96.387	102.676	106.642	68.982
7760	91.697	98.204	101.724	68.979
7800	90.156	94.085	99.669	67.468
7840	88.895	89.577	90.973	65.965
7880	95.393	86.479	93.093	68.011
7920	112.197	84.363	89.828	74.433
7960	104.902	85.732	92.995	71.514
8000	95.731	85.236	91.279	70.299
8040	90.750	82.813	89.995	68.408
8080	91.765	80.863	86.749	67.831
8120	88.536	77.827	86.574	68.074
8160	89.008	76.915	83.060	66.665
8200	101.389	75.437	82.536	68.560
8240	111.000	77.627	82.546	73.986
8280	102.252	76.958	86.057	71.711
8320	95.630	76.630	84.645	69.049
8360	91.936	73.845	84.538	66.060
8400	90.504	74.585	81.645	65.211
8440	89.974	73.757	80.970	67.060
8480	90.331	74.518	80.850	66.137
8520	101.536	71.926	79.738	67.574
8560	107.727	72.817	78.740	73.053
8600	98.558	75.384	80.795	68.958
8640	94.036	74.123	80.186	69.915
8680	90.876	73.380	78.041	66.173
8720	90.139	73.648	77.016	66.930
8760	88.915	73.704	77.046	65.278
8800	91.801	72.299	77.333	65.754
8840	100.453	73.768	76.779	66.215
8880	105.440	71.915	77.082	69.996
8920	97.549	74.764	77.984	69.236
8960	92.209	72.817	77.145	66.810
9000	90.321	72.665	75.331	66.345
9040	90.062	72.162	76.995	68.060
9080	91.107	71.673	76.415	67.415
9120	90.274	71.528	76.210	66.870
9160	91.195	72.289	74.107	66.673
9200	91.758	72.250	77.180	66.775
9240	89.105	71.799	75.956	66.827

9280	90.132	69.725	75.495	63.725
9320	91.682	73.585	74.839	65.680
9360	90.850	73.613	75.924	65.785
9400	89.679	72.433	75.896	65.324
9440	90.571	70.866	75.948	65.637
9480	88.466	72.845	75.254	64.873
9520	91.476	72.201	76.732	65.271
9560	90.241	72.310	75.333	65.032
9600	90.876	71.683	75.142	65.824
9640	90.070	70.912	75.115	65.757
9680	89.853	70.908	73.770	64.324
9720	90.445	71.669	74.210	65.581
9760	93.564	71.423	73.486	66.366
9800	91.340	71.894	73.708	66.771
9840	91.774	71.722	76.331	64.187
9880	90.455	71.606	74.607	67.113
9920	90.496	72.271	73.303	66.264
9960	91.602	71.278	74.240	65.313
10000	90.842	72.380	74.847	66.183
10040	89.453	72.820	75.230	66.032
10080	90.729	72.468	75.068	65.155
10120	90.611	71.746	75.093	64.570
10160	89.502	71.982	77.014	66.342
10200	91.786	73.014	76.068	65.049
10240	91.353	71.222	75.989	64.095
10280	92.492	73.278	74.484	66.715
10320	92.508	71.996	74.260	66.518
10360	90.009	72.380	74.462	67.690
10400	90.073	71.264	73.779	66.732
10440	90.173	72.423	75.352	65.581
10480	91.256	71.810	75.137	64.169
10520	91.305	70.588	73.139	66.380
10560	91.336	70.514	75.221	64.694
10600	90.509	71.440	75.806	65.560
10640	90.680	71.785	76.454	66.718
10680	91.047	73.201	74.730	67.690
10720	89.679	71.778	74.869	66.102
10760	90.741	72.873	76.456	66.496
10800	91.090	74.018	74.265	67.525
10840	90.523	70.232	74.803	66.556
10880	90.846	72.468	74.839	65.352
10920	92.338	71.222	73.238	66.349
10960	90.492	70.461	73.984	65.254
11000	90.130	71.683	76.219	65.715
11040	90.143	72.989	75.131	66.215
11080	90.844	71.824	74.301	66.539
11120	91.088	71.764	75.238	66.820

11160	91.350	73.042	74.030	65.658
11200	91.064	72.588	75.735	66.306
11240	91.058	72.261	74.615	65.306
11280	91.684	72.567	74.443	63.722
11320	91.323	71.070	76.713	63.683
11360	91.267	72.905	75.809	66.000
11400	90.474	71.806	74.880	66.363
11440	90.825	71.817	73.784	65.620
11480	92.177	70.898	74.984	66.817
11520	92.385	70.063	75.191	65.035
11560	90.720	72.123	76.836	65.025
11600	91.173	72.754	74.156	66.261
11640	91.147	72.592	76.555	64.197
11680	89.564	73.060	75.997	65.954
11720	90.981	73.282	76.852	65.039
11760	91.053	72.264	76.891	64.542
11800	91.769	69.377	74.511	66.581
11840	92.032	71.982	73.951	66.528
11880	90.603	71.859	76.429	65.665
11920	89.970	72.451	74.451	66.165
11960	95.192	71.356	75.227	66.535
12000	118.571	71.380	76.038	76.996
12040	110.626	77.933	92.814	76.299
12080	100.620	88.387	105.464	71.873
12120	94.169	95.380	104.287	69.856
12160	91.154	96.141	101.598	68.528
12200	90.009	96.739	98.620	66.504
12240	88.107	94.176	93.418	66.898
12280	89.947	92.204	91.967	66.085
12320	89.985	89.454	89.694	65.930
12360	90.348	87.806	86.765	64.915
12400	89.250	83.764	86.224	65.665
12440	91.252	82.725	90.150	64.046
12480	105.026	89.680	104.183	70.173
12520	112.825	95.944	105.320	72.789
12560	103.857	101.842	105.186	70.831
12600	96.152	97.137	99.598	69.415
12640	91.329	92.514	95.303	67.968
12680	89.613	92.063	92.104	66.880
12720	92.961	91.975	104.022	67.250
12760	107.355	97.831	106.082	69.771
12800	103.581	99.099	106.645	69.658
12840	95.453	98.377	101.967	69.063
12880	91.171	91.303	96.637	67.518
12920	91.203	88.859	92.344	66.137
12960	88.645	83.792	90.178	66.356
13000	108.735	82.433	89.314	70.877

13040	108.637	82.327	90.145	72.405
13080	98.397	83.165	91.932	69.278
13120	92.355	80.426	90.325	66.320
13160	90.874	77.754	87.915	67.546
13200	91.427	76.915	85.989	66.063
13240	89.961	77.556	82.984	65.750
13280	98.906	75.954	79.801	66.035
13320	112.211	75.514	83.861	72.908
13360	102.831	83.595	97.131	69.947
13400	94.385	86.338	98.669	68.577
13440	90.485	82.669	96.273	67.496
13480	88.940	81.345	92.355	67.018
13520	89.436	78.711	89.645	66.602
13560	91.570	77.743	84.563	64.785
13600	108.684	77.246	86.713	69.739
13640	106.722	76.676	89.025	72.996
13680	95.105	77.447	87.385	67.993
13720	92.686	75.778	83.883	67.553
13760	90.190	74.148	83.407	66.697
13800	89.756	74.296	82.691	66.729
13840	88.352	75.373	80.459	67.504
13880	89.023	72.074	79.429	67.113
13920	90.047	72.824	79.459	65.609
13960	95.650	72.165	77.877	66.613
14000	93.109	72.243	78.385	66.775
14040	88.925	71.239	77.478	64.004
14080	88.323	72.493	77.672	63.116
14120	90.197	71.018	76.014	63.778
14160	90.102	71.458	76.956	65.243
14200	90.122	71.201	75.486	64.278
14240	90.883	71.067	75.421	63.979
14280	89.404	71.933	75.962	64.194
14320	89.611	71.004	76.571	64.535
14360	88.840	70.359	75.060	66.007
14400	88.483	72.342	74.030	63.303
14440	90.383	72.511	74.664	64.225
14480	90.669	70.651	74.768	65.324
14520	91.412	70.320	74.576	65.007
14560	88.173	71.627	75.478	63.592
14600	89.468	71.525	73.090	65.761
14640	89.724	71.151	74.858	65.366
14680	91.062	73.718	73.210	64.757
14720	89.549	71.968	73.410	64.176
14760	91.192	70.004	73.645	65.320
14800	90.774	72.884	74.831	64.359
14840	90.536	69.831	75.470	65.088
14880	89.957	69.627	73.150	64.197

14920	91.633	70.940	73.530	65.211
14960	90.024	70.292	73.743	64.701
15000	90.684	70.053	74.899	64.278
15040	89.429	70.123	74.232	64.292
15080	91.957	71.004	73.260	64.933
15120	90.517	70.053	73.822	65.486
15160	90.346	70.609	74.347	64.754
15200	90.205	70.285	72.563	67.440
15240	90.494	69.996	74.456	64.331
15280	90.026	70.141	71.904	65.514
15320	90.658	71.496	73.776	65.130
15360	90.398	69.275	73.637	66.113
15400	90.724	70.201	71.910	64.665
15440	89.914	71.296	74.675	65.243
15480	90.669	68.711	74.596	64.032
15520	89.803	71.620	73.836	65.687
15560	90.750	70.827	74.205	65.391
15600	90.897	70.056	74.899	63.331
15640	91.291	70.690	74.068	65.342
15680	90.038	71.130	74.735	66.504
15720	91.024	71.275	73.068	66.211
15760	90.660	71.750	76.566	65.680
15800	91.353	70.634	74.385	64.148
15840	91.325	70.461	74.079	65.250
15880	92.058	72.408	74.454	64.613
15920	89.994	70.011	75.249	66.007
15960	90.352	72.056	73.978	65.004
16000	91.094	71.070	74.683	65.937
16040	91.784	71.489	73.989	65.641
16080	90.226	70.884	75.290	63.845
16120	90.962	70.349	76.331	65.391
16160	92.188	70.359	75.150	64.817
16200	92.925	70.560	75.098	65.401
16240	91.350	68.169	74.268	64.581
16280	90.874	71.289	75.915	65.884
16320	91.139	71.437	73.683	65.599
16360	90.630	71.831	75.093	66.574
16400	92.737	69.398	72.495	64.616
16440	91.049	70.088	74.005	65.993
16480	89.216	70.539	73.757	64.475
16520	90.312	70.746	73.702	65.074
16560	91.402	68.764	73.833	66.782
16600	90.289	71.944	74.724	63.021
16640	91.835	70.915	74.680	64.433
16680	90.588	71.803	73.902	65.665
16720	90.776	71.433	73.842	66.475
16760	90.423	71.560	74.981	66.754

16800	91.038	70.803	74.719	64.236
16840	90.793	70.820	74.538	64.887
16880	90.987	71.835	74.484	65.996
16920	90.508	71.602	75.404	65.954
16960	91.291	71.644	74.352	64.947
17000	92.242	71.236	75.000	64.025
17040	91.553	68.838	75.123	64.570
17080	91.498	72.254	73.978	63.384
17120	92.917	70.239	74.131	65.475
17160	91.581	71.401	72.945	65.937
17200	91.117	70.616	75.101	65.567
17240	91.633	70.363	72.874	64.447
17280	91.128	70.933	73.347	65.602
17320	91.115	71.313	74.066	64.898
17360	89.722	71.961	73.847	64.782
17400	91.521	69.581	74.164	66.004
17440	90.459	70.335	74.270	64.458
17480	90.789	68.581	74.546	64.912
17520	91.602	72.194	73.205	65.201
17560	91.387	71.521	73.929	64.201
17600	90.370	70.655	73.934	64.697
17640	91.096	70.275	74.699	64.261
17680	91.107	70.063	73.743	64.014
17720	91.088	71.539	73.459	64.430
17760	91.789	69.387	73.068	65.004
17800	90.630	71.500	75.148	64.461
17840	91.254	70.560	76.806	65.768
17880	90.650	69.824	74.746	63.451
17920	92.468	72.465	74.314	66.141
17960	90.047	70.127	74.087	64.211
18000	91.267	70.014	73.284	64.842
18040	88.744	71.088	73.276	64.627
18080	92.397	69.697	75.055	62.771
18120	91.383	70.866	74.656	65.556
18160	90.068	71.176	74.522	65.225
18200	89.269	72.194	75.710	63.954
18240	100.703	70.725	74.929	67.496
18280	116.902	70.817	78.120	76.085
18320	110.299	78.391	97.495	74.838
18360	99.966	89.771	102.765	71.000
18400	94.831	95.356	103.962	67.349
18440	91.400	98.204	101.374	68.835
18480	91.199	95.032	97.514	65.511
18520	89.483	91.961	91.836	65.465
18560	89.556	91.021	90.396	65.352
18600	89.722	86.665	87.740	65.919
18640	89.305	84.292	85.858	66.208

18680	101.962	85.479	92.475	71.007
18720	114.167	91.701	106.954	72.500
18760	103.361	98.056	108.369	69.465
18800	94.774	98.637	102.117	66.979
18840	91.827	97.120	98.388	65.711
18880	90.611	93.313	94.178	67.035
18920	90.220	89.039	95.866	65.750
18960	103.109	92.532	107.437	68.042
19000	109.921	98.500	107.366	73.000
19040	99.263	97.669	101.399	70.405
19080	93.006	93.754	97.661	67.165
19120	90.449	89.687	93.585	66.415
19160	90.632	87.331	92.369	66.190
19200	88.897	86.190	103.686	63.655
19240	93.778	94.944	105.033	65.433
19280	109.288	93.708	101.011	67.637
19320	106.387	91.187	97.852	70.539
19360	96.605	84.863	93.402	68.053
19400	92.414	82.694	91.656	65.630
19440	90.357	82.722	90.314	64.792
19480	90.090	86.739	103.287	65.320
19520	89.321	89.461	105.421	65.088
19560	97.417	88.887	99.929	65.155
19600	107.094	84.169	94.557	69.356
19640	98.579	80.905	91.199	68.926
19680	92.274	80.514	88.090	66.201
19720	90.017	78.151	86.525	65.063
19760	89.630	80.665	102.044	66.137
19800	90.395	87.585	102.391	65.444
19840	93.355	87.144	98.921	64.067
19880	102.662	85.539	95.079	65.521
19920	99.071	80.511	89.153	67.556
19960	93.530	77.827	87.962	67.282
20000	90.705	76.349	84.658	66.130
20040	90.447	75.053	82.410	63.817
20080	89.648	76.254	85.041	63.873
20120	90.502	75.606	83.918	64.310
20160	91.062	71.926	82.311	64.644
20200	90.961	73.849	81.637	64.972
20240	90.357	71.870	80.169	64.419
20280	90.348	71.585	79.178	65.102
20320	88.709	71.937	78.765	63.750
20360	89.036	71.384	79.131	65.208
20400	89.780	72.095	76.634	63.581
20440	89.566	69.229	77.189	64.317
20480	90.680	70.954	75.973	64.074
20520	89.761	72.173	75.880	64.648

20560	90.414	72.465	74.145	64.391
20600	89.064	69.887	73.754	64.697
20640	90.791	70.542	74.749	65.211
20680	91.205	70.912	75.576	64.134
20720	88.776	70.246	75.738	64.704
20760	91.481	70.504	74.924	64.842
20800	89.788	68.923	72.877	65.373
20840	88.496	70.694	74.505	63.187
20880	90.075	70.518	72.656	63.317
20920	89.897	71.835	74.918	64.866
20960	89.838	70.930	73.213	63.556
21000	90.769	70.141	73.899	64.680
21040	91.130	69.542	73.511	65.852
21080	92.009	70.673	73.743	64.092
21120	90.423	71.310	73.112	65.102
21160	90.970	70.081	74.801	63.493
21200	88.436	70.662	73.514	63.651
21240	90.558	70.148	73.221	64.426
21280	91.429	70.356	71.970	63.743
21320	90.974	71.736	73.369	64.651
21360	90.158	69.694	72.246	66.873
21400	90.070	69.577	74.230	64.796
21440	90.470	69.415	72.934	64.363
21480	90.524	68.637	72.598	63.211
21520	89.731	70.282	70.866	65.296
21560	89.756	71.004	73.036	64.289
21600	89.092	69.975	73.175	64.905
21640	89.498	72.623	72.754	64.729
21680	91.487	70.915	73.085	64.996
21720	90.203	69.687	73.432	65.299
21760	89.929	69.961	75.549	64.947
21800	88.726	69.757	78.546	62.479
21840	90.135	71.905	82.940	63.268
21880	91.291	69.514	80.180	63.419
21920	91.641	70.701	79.109	64.106
21960	90.583	70.613	76.205	64.303
22000	90.543	69.870	79.044	64.567
22040	90.451	69.454	77.262	65.349
22080	90.765	70.028	75.063	64.609
22120	90.716	69.331	75.792	64.320
22160	90.534	70.222	72.833	64.511
22200	89.408	69.842	73.333	64.215
22240	91.117	70.810	73.661	63.553
22280	90.190	71.282	74.243	65.852
22320	90.908	68.803	74.404	63.915
22360	89.269	69.599	72.934	64.074
22400	89.209	69.327	75.538	64.782

22440	92.472	70.873	72.527	64.764
22480	91.720	69.222	73.776	64.792
22520	90.774	72.264	73.415	64.817
22560	89.805	69.870	74.585	63.785
22600	91.427	70.014	72.746	63.768